



# High-temporal-resolution water level and storage change data sets for lakes on the Tibetan Plateau during 2000–2017 using multiple altimetric missions and Landsat-derived lake shoreline positions

**Xingdong Li[1], Di Long[1], Qi Huang[1], Pengfei Han[1], Fanyu Zhao[1], and Yoshihide Wada[2]**

[1]State Key Laboratory of Hydroscience and Engineering, Department of Hydraulic Engineering, Tsinghua University, Beijing, China
[2]International Institute for Applied Systems Analysis (IIASA), 2361 Laxenburg, Austria

**Correspondence:** Di Long (dlong@tsinghua.edu.cn)

**Abstract.** The Tibetan Plateau (TP), known as Asia's water tower, is quite sensitive to climate change, which is reflected by changes in hydrologic state variables such as lake water storage. Given the extremely limited ground observations on the TP due to the harsh environment and complex terrain, we exploited multiple altimetric missions and Landsat satellite data to create high-temporal-resolution lake water level and storage change time series at weekly to monthly timescales for 52 large lakes (50 lakes larger than $150\,km^2$ and 2 lakes larger than $100\,km^2$) on the TP during 2000–2017. The data sets are available online at https://doi.org/10.1594/PANGAEA.898411 (Li et al., 2019). With Landsat archives and altimetry data, we developed water levels from lake shoreline positions (i.e., Landsat-derived water levels) that cover the study period and serve as an ideal reference for merging multisource lake water levels with systematic biases being removed. To validate the Landsat-derived water levels, field experiments were carried out in two typical lakes, and theoretical uncertainty analysis was performed based on high-resolution optical images (0.8 m) as well. The RMSE of the Landsat-derived water levels is 0.11 m compared with the in situ measurements, consistent with the magnitude from theoretical analysis (0.1–0.2 m). The accuracy of the Landsat-derived water levels that can be derived in relatively small lakes is comparable with most altimetry data. The resulting merged Landsat-derived and altimetric lake water levels can provide accurate information on multiyear and short-term monitoring of lake water levels and storage changes on the TP, and critical information on lake overflow flood monitoring and prediction as the expansion of some TP lakes becomes a serious threat to surrounding residents and infrastructure.

## 1 Introduction

The Tibetan Plateau (TP), providing vital water resources for more than a billion population in Asia, is a sensitive region undergoing rapid climate change (Field et al., 2014). There are more than 1200 alpine lakes larger than $1\,km^2$ on the TP, where glaciers and permafrost are also widely distributed. With little disturbance by human activity in this area, lake storage changes may serve as an important indicator that reflects changes in regional hydrologic processes and responses to climate change. Wang et al. (2018) showed that global endorheic basins are experiencing a decline in water storage, whereas the endorheic basin on the TP is an exception. Given the fact that TP lakes have been expanding for more than 20 years (Pekel et al., 2016), quality data sets on lake water level and/or storage could be the basis for investigating its causes (e.g., climate change/variability) and interactions with the water/energy cycles and human society (e.g., increasing risks of inundation and overflow floods).

As an important component of the hydrosphere, terrestrial water cycle, and global water balance, millions of inland water bodies such as lakes, wetlands, and reservoirs have been investigated globally, and their total storage was estimated to be $181.9 \times 10^3 \text{ km}^3$ based on statistical models (Lehner and Döll, 2004; Messager et al., 2016; Pekel et al., 2016). Lake storage changes that play an important role in the regional water balance can be derived from changes in lake water level and area (Frappart et al., 2005). Lake water levels and areas are mostly derived from satellite remote sensing due to the scarcity of in situ data across the TP, where the harsh environment and complex terrain make in situ measurements difficult to perform and costly (Crétaux et al., 2016; Song et al., 2013; Yao et al., 2018b; Zhang et al., 2017a). Lake water levels can be monitored using satellite altimeters initially designed for sea surface topography or ice sheet/sea ice freeboard height measurements. Satellite altimeters determine the range between the nadir point and satellite by analyzing the waveforms of reflected electromagnetic pulses.

There are two major categories of satellite altimeters, i.e., laser and radar. Laser altimeters, e.g., the Ice, Cloud, and land Elevation Satellite (ICESat), operating in the near-infrared band have smaller footprints and generally higher accuracy than radar altimeters, facilitating applications in glacier/ice mass balance studies (Neckel et al., 2014; Sørensen et al., 2011). Radar altimeters, operating in the microwave band, have larger footprints and are more likely to be contaminated by a signal from complex terrain when applied to inland water bodies. Nevertheless, it is possible to remove these impacts with waveform retracking algorithms (Guo et al., 2009; Huang et al., 2018; Jiang et al., 2017). Zhang et al. (2011) mapped water level changes in 111 TP lakes for the 2003–2009 period using ICESat data that have a temporal resolution of 91 days. ICESat data have relatively denser ground tracks but a lower temporal resolution than most of other altimetric missions. This means that ICESat covers more lakes but provides few water levels for each lake. After ICESat was decommissioned in 2010, CryoSat-2 data starting from 2010 were adopted in related studies (Jiang et al., 2017), due to its similar dense ground tracks and competitive precision compared to ICESat. Other altimetric missions, such as TOPEX/Poseidon (T/P), Jason-1/2/3, the European Remote Sensing (ERS-1/2) satellite, and Envisat, also have some but relatively limited applications in monitoring changes in lake water level on the TP due to sparse ground tracks. In this study, multisource altimetry data (i.e., Jason-1/2/3, Envisat, ICESat, and CryoSat-2) were combined if available for lakes in this study, with the Landsat-derived water levels developed in this study as a critical reference to increase the water level observations and merging data from multiple altimetric missions.

Changes in lake area can be captured by optical or synthetic aperture radar (SAR) images from medium- or high-spatial-resolution remote sensing data, such as Landsat and Sentinel series. Extraction of lake water bodies can be manually (Wan et al., 2016) or automatically (Zhang et al., 2017b) achieved. Automatic water extraction methods based on the water index and auto-thresholding are more efficient in dealing with a mass of remote sensing images. Even so, acquisition and preprocessing of such a large amount of historical data ($\sim 10\,\text{TB}$) covering TP lakes are still intractable for researchers with limited computational resources. With the help of cloud-based platforms, such as the Google Earth Engine (GEE) that significantly reduces data downloading and preprocessing time, tens of thousands of images may be processed online in days instead of months (Gorelick et al., 2017). In this study, more than 20 000 Landsat images were processed online using GEE to extract lake water bodies based on the water index (McFeeters, 1996) and Otsu algorithm (Otsu, 1979).

There have been studies focusing on changes in lake water storage on the TP over recent decades; e.g., Zhang et al. (2017a) examined changes in water storage for $\sim 70$ lakes from the 1970s to 2015 with ICESat altimetry data and Landsat archives. An individual lake area data set from the 1970s and annual area data after 1989 were used. Due to the short time span of ICESat, they used the hypsometric method to convert lake areas into water levels. Yao et al. (2018b) used digital elevation models (DEMs) and optical images to develop hypsometric curves for lakes on the central TP and estimated annual changes in water storage for 871 lakes from 2002 to 2015. These studies have a wide spatial coverage of lakes but relatively lower temporal resolution and little spaceborne altimetric information, which may limit the accuracy of trends in lake water level/storage in some cases and short-term monitoring of lake overflow floods. The Laboratoire d'Etudes en Géophysique et Océanographie Spatiales (LEGOS) Hydroweb provides a lake data set, including multisource altimetry-based changes in lake water level and storage as well as hypsometric curves for 22 TP lakes (Crétaux et al., 2016, 2011b). The data set incorporates more spaceborne altimetric information and has a higher temporal resolution. However, there may be a remaining bias when different sources of altimetric data are merged, due to the lack of some important reference that can be derived from optical remote sensing to be shown in this study. We term the reference data the "Landsat-derived water level" to be introduced in Sect. 3.2. Here, we list recent studies and data sets (Table 1) to provide a concise summary on remote sensing monitoring of water levels and storage changes over lakes on the TP.

The overall objective of this study was to examine multiyear and short-term changes in water level and storage across 52 lakes with surface areas larger than $150\,\text{km}^2$ on the TP by merging multisource altimetry and optical remote sensing images to generate more coherent high-temporal-resolution lake water level and storage change data sets ranging from weekly to monthly timescales during 2000–2017 and the hypsometric curve (i.e., the lake-area–water-level relationship) for each lake. To investigate changes in lake storage,

**Table 1.** Recent studies and data sets on TP lakes. H, A, and V in the table denote lake water level, area, and volume, respectively.

| Reference | No. of lakes | Data type | Time span | Temporal resolution | Source data |
|---|---|---|---|---|---|
| Song et al. (2013) | 30 | H, A, V, and hypsometric curve | 4 records for the 1970s, 1990, 2000, and 2011 | Decadal | Altimetry data: ICESat Optical data: Landsat 5/7 TM/ETM+ |
| Crétaux et al. (2016) | 22 | H, A, V, and hypsometric curve | 1995–2015 relative bias partially removed (only for altimeters with overlapping period) | Submonthly for lakes with T/P and Jason-1/2 data, and ∼ monthly for lakes without Jason-1/2 or T/P data | Altimetry data: T/P, ERS-2, GFO, Envisat, Jason-1/2, SARAL, ICESat, and CryoSat-2 Optical data: Landsat 5/7/8 TM/ETM+/OLI and MODIS |
| Jiang et al. (2017) | 70 | H | 2003–2015 relative bias between ICESat and CryoSat-2 unremoved | ∼ Monthly | Altimetry data: ICESat and CryoSat-2 |
| Zhang et al. (2017a) | 60–70 | H, A, V, and hypsometric curve | One record for the 1970s, and annual data for 1989–2015 | Annual | Altimetry data: ICESat Optical data: Landsat 5/7/8 TM/ETM+/OLI |
| Li et al. (2017b) | 167 | H | 2002–2012 | ∼ Monthly | Altimetry data: ICESat and Envisat |
| Yao et al. (2018b) | 871 | H, A, V, and hypsometric curve | 2002–2015 | Annual | Optical data: Landsat 5/7/8 TM/ETM+/OLI and HJ-1A/1B DEM data: SRTM and ASTER |
| Hwang et al. (2019) | 59 | H | 2003–2016 relative bias partially removed (only for lakes with Jason data/in situ data) | Submonthly for lakes with Jason-2 data, and ∼ monthly for lakes without Jason-2 data | Altimetry data: Jason-2/3, SARAL, ICESat, and CryoSat-2 (Jason-3 data for validation) |
| Our study | 52 | H, A, V, and hypsometric curve | 2000–2017 all relative biases removed | Submonthly for most lakes | Altimetry data: Jason-1/2/3, Envisat, ICESat, and CryoSat-2 Optical data: Landsat 5/7/8 TM/ETM+/OLI |

lake water levels and areas need to be derived from multi-source remote sensing.

First, water levels from various satellite altimeters (Fig. 1) for each lake as well as lake shoreline positions and lake areas from optical remote sensing images (i.e., Landsat) were derived. Second, systematic biases between different altimetry data were removed by either comparing the mean water levels during the overlap period (Fig. 1) or comparing the two water level time series with lake shoreline positions, depending on the length of the overlap period (details can

be found in Sect. 3.1). Lake-shoreline-position-derived water levels, termed the Landsat-derived water levels in this study, can serve as a unique source of information reflecting water levels as well as a data merging reference. We will show that after deriving two or three regression parameters, lake shoreline positions can well reflect lake water levels with comparable accuracy to altimetry-derived water levels. Third, with information on lake water levels and areas derived from altimetry data and optical remote sensing images, the hypsometric curve that describes the relationship between the lake

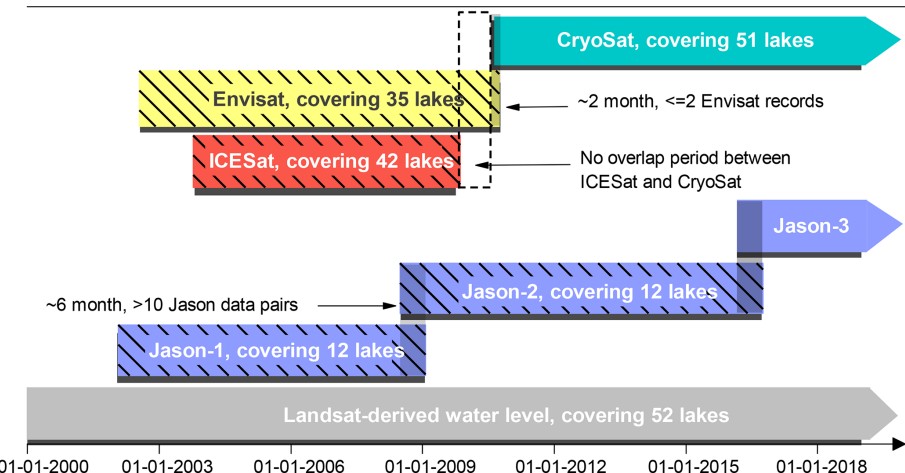

**Figure 1.** Spatial (the number of lakes covered) and temporal coverage and their overlap periods of multiple satellite altimetric missions used in this study, including Jason-1/2/3, Envisat, ICESat, and CryoSat-2.

water level and storage changes can be derived. Fourth, the integral of the hypsometric curve was performed to convert lake water levels into storage changes.

Results of this study provide a comprehensive and detailed assessment of changes in lake level and storage on the TP for the recent 2 decades and short-term monitoring of lake over-flow floods for some lakes. This study could largely benefit more detailed investigations into lakes, lake basins, and regional climate change, because the generated data sets have the highest temporal resolution during the study period with systematic biases well removed. To ensure the data quality, field experiments were carried out and in situ data were collected to examine the uncertainty in the Landsat-derived water levels. Users are free to access the data set described in this paper at https://doi.org/10.1594/PANGAEA.898411 (Li et al., 2019).

## 2    Study area and data

### 2.1    Study area

The TP can be generally divided into 12 major basins (Wan et al., 2016; Zhang et al., 2013), among which the inner/central TP (CP) is the only endorheic basin and home to most TP lakes, including $\sim 300$ TP lakes larger than $10\,\mathrm{km}^2$. Therefore, it was chosen as the main study area. The endorheic basin covers an area of $\sim 710\,000\,\mathrm{km}^2$ ($\sim 28\,\%$ of total TP) with a mean elevation of $\sim 4900\,\mathrm{m}$ and has a semiarid plateau climate with annual precipitation ranging from 96 to 295 mm (Li et al., 2017c). Most lakes in the endorheic basin were expanding under the influence of climate change/variability as opposed to other areas in the TP, e.g., Selin Co exceeded Nam Co in area and consequently became the largest lake in the endorheic basin between 2011 and 2012 and expanded by 26 % over the past 40 years (Zhou et al., 2015), whereas

Yamzhog Yumco (also known as Yamdrok Lake; outside the endorheic basin, 350 km to the southeast of Selin Co) shrunk by $\sim 11\,\%$ during 2002–2014 according to Wan et al. (2016). Located in the southeast endorheic basin, the Nam Co basin covering about $10\,800\,\mathrm{km}^2$, with 19 % of the basin lake water area and a mean lake elevation of $\sim 4730\,\mathrm{m}$, was chosen as a field experiment spot. The mean annual temperature and precipitation of Nam Co are 1.3° and 486 mm, respectively (Li et al., 2017a). The other experiment spot was Yamzhog Yumco, which has a mean lake elevation of $\sim 4440\,\mathrm{m}$. Subject to steep mountainous terrain, the lake has a narrow-strip shape with complex shorelines. The basin of Yamzhog Yumco covers $\sim 6100\,\mathrm{km}^2$, with mean annual temperature and precipitation of 2.8° and $\sim 360\,\mathrm{mm}$, respectively (Yu et al., 2011). An overall map of experiment lakes is given in Fig. 2.

### 2.2    Data

Multisource altimetry data were used in this study as shown in Table 2. The earliest record dates back to 2002 (i.e., Envisat and Jason-1) and the latest record ends in 2017 (i.e., Jason-3 and CryoSat-2, Fig. 1). Most of the 52 lakes examined in this study were covered by ICESat, Envisat, and CryoSat-2 data. ICESat data provided by the National Aeronautics and Space Administration (NASA) were available on 42 lakes in this study. Envisat and CryoSat-2 data provided by the European Space Agency (ESA) were available on 35 and 51 lakes in this study. Jason-1/2/3 data provided by the Centre National d'Etudes Spatiales (CNES) were available only on 12 lakes in this study due to the relatively sparse ground tracks or data quality issues. Note that Jason-2 inherited the orbit of Jason-1 after its launch in 2008, whereas Jason-1 was shifted into an interleaved orbit and continued functioning until 2013, thereby increasing the spatial coverage of Jason altimetry series to some degree, e.g., Jason-

**Table 2.** Multisource altimetry data used in this study.

| Mission | Sensor (type) | Data record | Duration | Repeat cycle (day) | Footprint interval (m) | Footprint diameter (km) | Lake no. with data | Data source |
|---------|---------------|-------------|----------|--------------------|------------------------|-------------------------|--------------------|-------------|
| Jason-1 | Poseidon-2 (radar) | S-GDR | 2002–2013 | 10 | $\sim$ 300 | 2–4 | 12 | CNES Aviso+ |
| Jason-2 | Poseidon-3 (radar) | S-GDR | 2008– | 10 | $\sim$ 300 | 2–4 | 12 | CNES Aviso+ |
| Jason-3 | Poseidon-3B (radar) | S-GDR | 2016– | 10 | $\sim$ 300 | 2–4 | 12 | CNES Aviso+ |
| Envisat | RA-2 (radar) | GDR | 2002–2010 | 35 | $\sim$ 390 | 3.4 | 35 | ESA |
| CryoSat-2 | SIRAL (radar) | InSAR Level 1 | 2010– | 369 (subcycle 30) | $\sim$ 280 | $\sim$ 1.65 (across track), $\sim$ 0.3 (along track) | 51 | ESA |
| ICESat | GLAS (laser) | GLAH 14 | 2003–2009 | 91 | $\sim$ 170 | $\sim$ 0.07 | 42 | NASA |

S-GDR stands for Sensor Geophysical Data Record; GDR stands for Geophysical Data Record; GLAH 14 stands for GLAS/ICESat L2 Global Land Surface Altimetry Data (HDF5), version 34; CNES stands for Centre National d'Etudes Spatiales; Aviso stands for Archiving, Validation and Interpretation of Satellite Oceanographic data; Aviso+ data set is available via FTP at http://ftp-access.aviso.altimetry.fr with a registered username and password (last access: 18 August 2019); ESA Envisat products are available via FTP at http://ra2-ftp-ds.eo.esa.int with a registered username and password (last access: 18 August 2019); ESA CryoSat-2 products are available via FTP at http://calval-pds.cryosat.esa.int with a registered username and password (last access: 18 August 2019); NASA ICESat products are available at https://nsidc.org/data/icesat/data.html (last access: 18 August 2019).

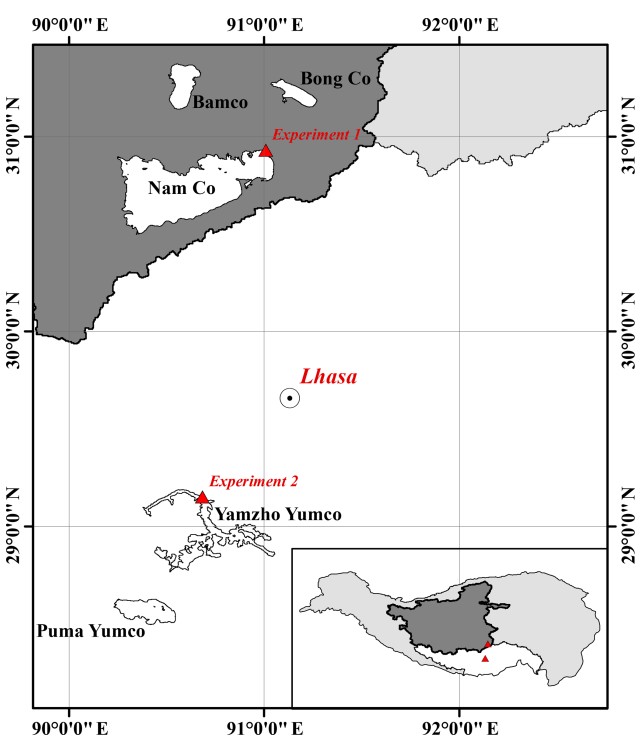

**Figure 2.** Experiment locations: Nam Co and Yamzhog Yumco. Nam Co is located in the endorheic basin of the TP, while Yamzhog Yumco is located in the Yarlung Zangbo river basin (the upper Brahmaputra River). Both lakes are close to Lhasa city.

1 data in Lake Qinghai, the largest lake on the TP, were only available after 2008 due to the orbit shift. ICESat and CryoSat-2 data have the largest spatial coverage but relatively long repeat cycles of 91 and 369 days, respectively (Bouzinac, 2012; Zhang et al., 2011). The Envisat mission has a lower orbit than Jason-1/2/3 but higher than ICESat, resulting in a moderate spatial coverage and a temporal resolution of 35 days (Benveniste et al., 2002). To determine if

the altimetry data fall into the lakes, a lake shape data set generated by Wan et al. (2016) was used. An example of using the lake shape data set to determine altimetry data falling into the lake boundaries is given in Fig. 3a, showing that data from all altimeters are available in Zhari Namco.

It should be noted that different altimeters vary with wavelengths of electromagnetic radiation and mechanisms. For instance, Jason-1/2/3 using the Ku and C bands and Envisat/RA-2 using the Ku and S bands work in the low-resolution mode (LRM). These dual-frequency radar altimeters can provide more accurate range corrections due to the ionospheric effect (Tournadre, 2004). The LRM is typical for the early version of satellite altimeters such as TOPEX/Poseidon. There are more advanced modes, such as SAR and interferometric synthetic aperture radar (InSAR), for recent radar altimeters, which generally have smaller footprints than the LRM mode. CryoSat-2/SIRAL working at a single Ku band has three modes, including LRM, SAR, and InSAR, which were designed to have an increasing resolution in turn and work in different zones. The InSAR mode uses interference phenomena so that the shift of the nadir point across the track can be detected, improving the altimeter's performance on ice sheets with slopes (Bouzinac, 2012). The Geoscience Laser Altimeter System (GLAS) is the laser altimeter carried by ICESat working in the near-infrared band.

We used Landsat 5 TM (2000–2011), Landsat 7 ETM+ (2000–2017), and Landsat 8 OLI (2013–2017) surface reflectance data sets provided by GEE to generate information on lake shoreline positions and lake areas. Landsat 7 ETM+ was subject to sensor failure, and all the Landsat 7 ETM+ images contain gaps after 2003 (Markham et al., 2004). There were more than 20 000 images processed, and half of them were excluded from the final results due to cloud contamination or gaps. We collected daily in situ water level measurements in Yamzhog Yumco for validation purposes with a pressure-type water level sensor. The in situ water level measurements spanned half a year from May to Octo-

ber 2018. We also performed unmanned aerial vehicle (UAV or drone) imaging over a 1 km lake bank in Yamzhog Yumco and Nam Co for obtaining better knowledge on the experimental environment.

In addition, GaoFen-2 (GF-2, the China High-Resolution Earth Observation System mission) images were used to perform a rigorous statistical analysis of uncertainty in the Landsat-derived water levels by taking the GF-2-derived lake shoreline positions as the ground truth to analyze the sub-pixel water area ratios of Landsat image pixels (see Sect. 4). GF-2 images have a spatial resolution of 0.8 m for the panchromatic band, and preprocessing including orthorectification and radiometric calibration was performed. Before analysis, we performed an image-to-image registration with manually selected tie points between GF-2 and corresponding Landsat 8 OLI images until the coregistration error reduced to ∼ 2 m.

## 3    Method

### 3.1    Satellite altimetry water level

The first step of deriving satellite altimetry water levels is to select correct ground tracks and valid footprints falling on the lakes. Because there is a random ground track shift at ∼ 1 km in different cycles for most altimetry missions, it is uncertain whether valid lake footprints can be obtained for each cycle, even though the nominal ground track seems to cross the lake. This problem can be addressed by comparing geographic coordinates of the footprints with a lake shape data set (Wan et al., 2016). After picking out the valid footprints, the lake surface height can be calculated for each footprint. All radar altimetry data share a relationship:

$$LSH = Alt − (Range + cor),  \quad (1)$$

where LSH represents the lake surface height with respect to the geoid; Alt represents the altimeter height with respect to the reference ellipsoid; Range represents the distance between the altimeter and lake surface; and cor represents several range corrections due to atmospheric effects, sensor design defects, or geophysical effects. Radar altimeters and laser altimeters need different corrections, given that they are working in different wavelengths and have different designs. For instance, corrections for radar altimeters include waveform retracking correction, wet/dry troposphere correction, ionosphere correction, pole/solid tide correction, and geoid correction. Laser altimeters also need atmospheric delay correction, geoid/pole tide correction, and geoid correction. Unlike radar altimeters, saturation correction instead of waveform retracking correction is more important to laser altimeters.

The retracking correction plays an important role in removing the contamination of land signal when radar altimetry data are applied to inland water bodies. In this study, Jason-1/2/3 data were retracked using a classical waveform

retracking algorithm, i.e., the improved threshold method (ITR), whereas CryoSat-2 data were retracked using the narrow primary peak threshold (NTTP) method (Birkett and Beckley, 2010; Cheng et al., 2010; Jain et al., 2015). Retracking corrections were not performed for Envisat and ICESat data, because the Envisat/RA-2 product already provided a retracked range using the ICE-1 method, and the ICESat GLAH14 product already included several corrections (such as saturation correction) that are sufficient for most applications including studies on inland water bodies (Zhang et al., 2011). The original idea of the NTTP, ICE-1, and ITR is quite similar. All of them adopt a threshold defined as the percentage of the waveform peak to determine the retracking gate and then to convert the difference between the retracking gate and the nominal gate into range correction by multiplying the gate range ($c\Delta t/2$, where $c$ is the speed of light and $\Delta t$ is the time duration of a gate). The differences lie in the choice of thresholds as well as the calculation of waveform peaks. For instance, ICE-1 uses a 30 % threshold, whereas ITR uses a 50 % threshold.

For each cycle of an altimeter, it is common that more than one footprint fall on a lake, thereby providing several lake surface height (LSH) observations on the same day. After removing outliers with the three-sigma rule, frequency distributions of the LSHs from the same cycle were calculated. The mean value of the histogram bin with the highest frequency was selected to represent the LSH for the cycle. Meanwhile, the frequency of the chosen histogram bin was reserved to evaluate the data quality for the cycle; e.g., a cycle was marked as high quality if the frequency was higher than 0.8, moderate quality if it was only higher than 0.5, and poor quality if the frequency was lower than 0.5. The LSH from each cycle constituted the water level time series for a lake. LSHs that were marked as poor quality and obviously deviated from the moving average were removed from the altimetry-based lake water level time series.

It is not uncommon that systematic biases exist in different altimetry data sets due to variations in orbit, the discrepancy between correction models, errors associated with sensors, and even the choice of the reference datum. After deriving lake water level time series for each altimeter, we first merged the Envisat and ICESat water levels if both were available for a lake, because they have the longest overlap period (Fig. 1). We chose Envisat-derived water levels as the baseline and removed the difference of the mean values of the two products during the overlap period from the ICESat data, because Envisat data are generally denser and longer than ICESat data. A similar process was applied to Jason-1/2/3, as there are two overlap periods connecting the three altimeters together. Figure 3b shows a result of merged altimetry data when all sensors are available. There are trade-offs between CryoSat-2 and Jason-2/3 data in terms of spatial coverage and time span. CryoSat-2 data are available for all lakes in this study but they only have an overlap period with Jason-2/3 data, whereas Jason-2/3 data are only available for

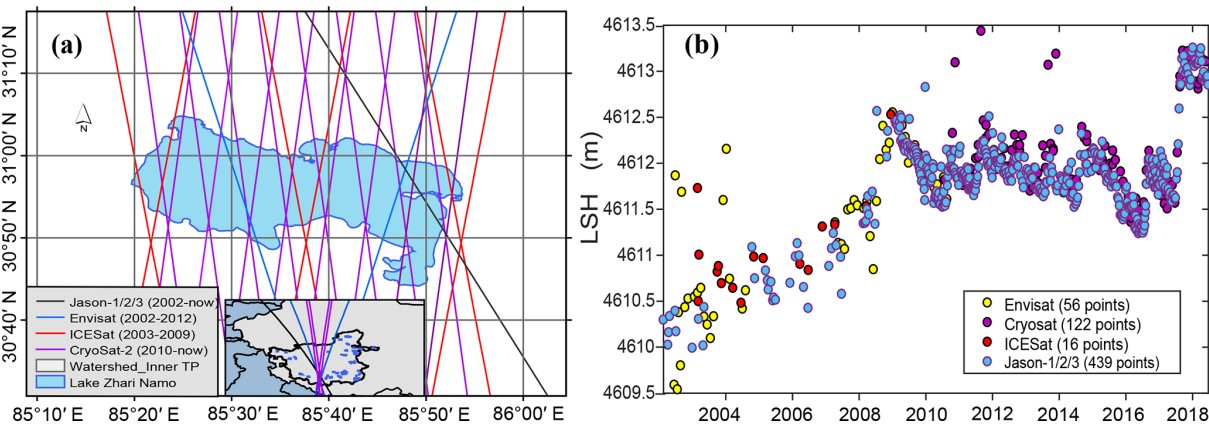

**Figure 3. (a)** Ground tracks of multiple altimetric missions over Zhari Namco and **(b)** the merged altimetry water levels for Zhari Namco. LSH stands for lake surface height.

12 lakes. For most lakes without Jason-2/3 data, we merged CryoSat-2 data with either ICESat or Envisat using Landsat-derived water levels spanning from 2000 to 2017, because there is no overlap period between these altimetry water levels (Fig. 1). Details on how Landsat-derived water levels aid in merging the altimetry water levels are shown in Sect. 3.2.

## 3.2 Landsat-derived water level

For most lake basins, it is possible to find a relatively flat portion of lake banks with an average slope of 1/30 or even smaller, where obvious interannual or intra-annual changes in lake shoreline positions can be detected using Landsat images (30 m). These locations can be found by comparing lake images from the first year and the last year of the study period if the lake shows a clear expanding/shrinking trend. Otherwise, we can compare images acquired in early summer when the LSH is at its lowest level with those acquired in late autumn when the lake expands to its limit. In this study, we assumed that the selected lake bank was flat enough such that the relationship between the lake water level and shoreline position can be depicted in a linear or quasilinear (parabolic) way. Thus, we can transform the lake shoreline positions into Landsat-derived water levels by fitting with altimetry water levels. The validity of this assumption can be evaluated with the coefficient of determination ($R^2$) for each lake as shown in Table 3. For most lakes, the goodness of fit is higher than 0.7, suggesting the generally good fitting relationship between the lake water levels and shoreline positions.

Though there were $\sim 500$ Landsat images obtained for the selected lake banks during the study period, many of them were largely affected by cloud or cloud shadow. All the images were processed online using the GEE application programming interface (API). Preprocessing such as radiometric calibration, atmospheric correction, and geometric correction was already performed in the production of the data sets. In

addition, the failure of the Landsat 7 sensor SLC left all the Landsat 7 ETM+ images with gaps after 2003 (Markham et al., 2004), making the available images even fewer. We managed to make use of some images with gaps in generating lake shoreline positions. By choosing the region of interest (ROI) that is parallel to the image gaps, we made most of the Landsat 7 ETM+ images useable. However, the width of ROI must be reduced to avoid shifting gaps as shown in Fig. 4b. The gaps may vary with time but are more like vibration around the midpoint. The ROI did not fill the interval of gaps, because the wider the ROI, the higher possibility of shifting gaps cross it.

Lake shoreline positions were characterized by water area ratios detected in the ROI. To automatically extract water areas from a mass of Landsat archives on GEE, the water index and Otsu threshold method were jointly used. We calculated the normalized difference water index (NDWI) and the modified normalized difference water index (MNDWI) of the images and compared their performance in different seasons. It was found that the MNDWI tends to be more sensitive to shallow water in summer but is less effective than the NDWI when the lake bank is covered by snow in the cold season as shown in Fig. 5c. Therefore, the two water indices were jointly used by applying the MNDWI to images acquired during May to October and applying the NDWI to the remaining months. The NDWI and MNDWI can be calculated as follows (McFeeters, 1996; Xu, 2005):

$$NDWI = \frac{B_{\text{green}} - B_{\text{NIR}}}{B_{\text{green}} + B_{\text{NIR}}}, \tag{2}$$

$$MNDWI = \frac{B_{\text{green}} - B_{\text{SIR}}}{B_{\text{green}} + B_{\text{SIR}}}, \tag{3}$$

where $B_{\text{green}}$, $B_{\text{NIR}}$, and $B_{\text{SIR}}$ refer to surface reflectance of bands 2, 4, and 5 for Landsat 5/7 TM/ETM+ images and bands 3, 5, and 6 for Landsat 8 OLI images.

After calculating the water index, the grayscale image was binarized using the Otsu method. If the selected ROI com-

**Table 3.** Summary of regression analyses and hypsometric function by lake.

| Lake name | Lake area (km$^2$) | No. of Landsat-derived water level | $R^2$ of Landsat-derived water level (no. of data pairs) | $R^2$ of hypsometric curve (no. of data pairs) | Hypsometric function |
|---|---|---|---|---|---|
| Ake Sayi Lake | 260.74 | 113 | 0.8951 (14) | 0.9556 (21) | $S = 0.45\,\mathrm{d}h^2 + 11.26\,\mathrm{d}h + 163.97$, $\mathrm{d}h = H - 4846$ |
| Lake Aqqikkol | 538.21 | 354 | 0.9717 (44) | 0.9353 (36) | $S = 2.36\,\mathrm{d}h^2 + 0.21\,\mathrm{d}h + 370.29$, $\mathrm{d}h = H - 4252$ |
| Lake Ayakum | 987.23 | 183 | 0.9651 (50) | 0.9695 (57) | $S = 0.16\,\mathrm{d}h^2 + 65.72\,\mathrm{d}h + 658.28$, $\mathrm{d}h = H - 3878$ |
| Bam Co | 255.81 | 209 | 0.901 (14) | 0.9287 (27) | $S = 0.28\,\mathrm{d}h^2 + 2.84\,\mathrm{d}h + 206.46$, $\mathrm{d}h = H - 4560.5$ |
| Bangong Co | 661.64 | 232 | 0.5164 (172) | 0.7991 (29) | $S = 1.43\,\mathrm{d}h^2 + 15.67\,\mathrm{d}h + 619.28$, $\mathrm{d}h = H - 4238$ |
| Chibzhang Co | 541.96 | 49 | 0.8766 (19) | 0.9792 (15) | $S = 0.69\,\mathrm{d}h^2 + 3.36\,\mathrm{d}h + 475.79$, $\mathrm{d}h = H - 4930$ |
| Co Ngoin1 | 268.37 | 174 | 0.6637 (15) | 0.8803 (62) | $S = 3.67\,\mathrm{d}h^2 + -1.33\,\mathrm{d}h + 263.1$, $\mathrm{d}h = H - 4564.5$ |
| Cuona Lake | 192.15 | 254 | 0.7607 (12) | 0.8876 (27) | $S = 1.77\,\mathrm{d}h^2 + 3.6\,\mathrm{d}h + 184.79$, $\mathrm{d}h = H - 4585.5$ |
| Dagze Co | 310.8 | 192 | 0.8334 (67) | 0.8862 (28) | $S = 0.08\,\mathrm{d}h^2 + 6.14\,\mathrm{d}h + 230.51$, $\mathrm{d}h = H - 4460$ |
| Dogai Coring | 492.39 | 257 | 0.8624 (152) | 0.9048 (33) | $S = 3.2\,\mathrm{d}h^2 + 5.66\,\mathrm{d}h + 427.17$, $\mathrm{d}h = H - 4816$ |
| Dogaicoring Qangco | 403.18 | 162 | 0.9202 (37) | 0.9218 (47) | $S = 0.53\,\mathrm{d}h^2 + 3.93\,\mathrm{d}h + 279.6$, $\mathrm{d}h = H - 4786$ |
| Donggei Cuona Lake | 247.83 | 561 | 0.8776 (38) | 0.925 (101) | $S = 0.54\,\mathrm{d}h^2 + 7.22\,\mathrm{d}h + 222.19$, $\mathrm{d}h = H - 4084$ |
| Dung Co | 139.4 | 145 | 0.9218 (49) | 0.8652 (28) | $S = 0.07\,\mathrm{d}h^2 + 2.3\,\mathrm{d}h + 137.06$, $\mathrm{d}h = H - 4547$ |
| Goren Co | 477.95 | 191 | 0.6166 (24) | 0.9096 (41) | $S = 2.91\,\mathrm{d}h^2 + -0.03\,\mathrm{d}h + 468.33$, $\mathrm{d}h = H - 4648.5$ |
| Gozha Co | 246.91 | 96 | 0.4297 (12) | 0.5564 (19) | $S = 1.57\,\mathrm{d}h^2 + -0.06\,\mathrm{d}h + 254.43$, $\mathrm{d}h = H - 5082$ |
| Gyaring Lake | 535.84 | 253 | 0.6217 (20) | 0.3451 (71) | $S = 1.99\,\mathrm{d}h^2 + 2.8\,\mathrm{d}h + 517.18$, $\mathrm{d}h = H - 4292$ |
| Har Lake | 621.52 | 370 | 0.8652 (63) | 0.9893 (50) | $S = 1.1\,\mathrm{d}h^2 + 1.52\,\mathrm{d}h + 582.34$, $\mathrm{d}h = H - 4075$ |
| Hoh Xil Lake | 351.3 | 132 | 0.9038 (12) | 0.9355 (27) | $S = 1\,\mathrm{d}h^2 + 5.29\,\mathrm{d}h + 300.5$, $\mathrm{d}h = H - 4887.1$ |
| Jingyu Lake | 339.69 | 224 | 0.8978 (51) | 0.989 (34) | $S = 0.37\,\mathrm{d}h^2 + 4.77\,\mathrm{d}h + 238.43$, $\mathrm{d}h = H - 4710$ |
| Kusai Lake | 328.8 | 295 | 0.9787 (151) | 0.8987 (49) | $S = 0.52\,\mathrm{d}h^2 + 5.04\,\mathrm{d}h + 254.67$, $\mathrm{d}h = H - 4473$ |
| Kyebxang Co | 187.32 | 233 | 0.75 (12) | 0.8753 (135) | $S = 0.16\,\mathrm{d}h^2 + 5.4\,\mathrm{d}h + 150.9$, $\mathrm{d}h = H - 4619$ |
| Langa Co | 256.03 | 167 | 0.859 (157) | 0.888 (47) | $S = -0.19\,\mathrm{d}h^2 + 4\,\mathrm{d}h + 249.28$, $\mathrm{d}h = H - 4564$ |
| Lexiewudan Co | 273.63 | 286 | 0.9216 (49) | 0.9496 (40) | $S = 0.13\,\mathrm{d}h^2 + 4.63\,\mathrm{d}h + 219.65$, $\mathrm{d}h = H - 4868$ |
| Lumajiangdong Co | 386.71 | 220 | 0.9135 (28) | 0.9708 (17) | $S = 0.62\,\mathrm{d}h^2 + 2.09\,\mathrm{d}h + 353.95$, $\mathrm{d}h = H - 4812$ |
| Mapam Yumco | 412.69 | 163 | 0.7096 (30) | 0.9973 (30) | $S = 1.18\,\mathrm{d}h^2 + 5.16\,\mathrm{d}h + 399.68$, $\mathrm{d}h = H - 4584$ |
| Margai Caka | 137.7 | 247 | 0.9399 (12) | 0.9955 (35) | $S = 0.03\,\mathrm{d}h^2 + 5.14\,\mathrm{d}h + 112.12$, $\mathrm{d}h = H - 4791$ |

| Lake name | Lake area (km$^2$) | No. of Landsat-derived water level | $R^2$ of Landsat-derived water level (no. of data pairs) | $R^2$ of hypsometric curve (no. of data pairs) | Hypsometric function |
|---|---|---|---|---|---|
| Memar Co | 167.3 | 193 | 0.911 (39) | 0.8626 (20) | $S = 0.27\,\mathrm{d}h^2 + 3.17\,\mathrm{d}h + 134.69$, $\mathrm{d}h = H - 4923$ |
| Nam Co | 2028.5 | 187 | 0.9064 (62) | 0.8749 (18) | $S = 2.43\,\mathrm{d}h^2 + 5.55\,\mathrm{d}h + 1970.1$, $\mathrm{d}h = H - 4724.5$ |
| Ngangla Ringco | 492.86 | 88 | 0.4652 (25) | 0.9498 (7) | $S = 3.87\,\mathrm{d}h^2 + 3.86\,\mathrm{d}h + 490.69$, $\mathrm{d}h = H - 4715$ |
| Ngangze Co | 471.21 | 245 | 0.9538 (236) | 0.9332 (49) | $S = 0.2\,\mathrm{d}h^2 + 7.03\,\mathrm{d}h + 391.21$, $\mathrm{d}h = H - 4680$ |
| Ngoring Lake | 656.83 | 86 | 0.844 (71) | 0.8613 (14) | $S = 4.69\,\mathrm{d}h^2 + -5.04\,\mathrm{d}h + 613.66$, $\mathrm{d}h = H - 4270$ |
| Paiku Co | 272.85 | 231 | 0.8341 (21) | 0.9079 (61) | $S = 0.91\,\mathrm{d}h^2 + 2.64\,\mathrm{d}h + 264.89$, $\mathrm{d}h = H - 4578.5$ |
| Puma Yumco | 290.98 | 250 | 0.6871 (18) | 0.5629 (53) | $S = 0.48\,\mathrm{d}h^2 + 0.8\,\mathrm{d}h + 286.34$, $\mathrm{d}h = H - 5011$ |
| Pung Co | 176.93 | 187 | 0.8017 (12) | 0.9841 (31) | $S = 0.03\,\mathrm{d}h^2 + 3.75\,\mathrm{d}h + 151.66$, $\mathrm{d}h = H - 4526$ |
| Qinghai Lake | 4495.33 | 323 | 0.9011 (151) | 0.8181 (19) | $S = 3.45\,\mathrm{d}h^2 + 155.03\,\mathrm{d}h + 4084.73$, $\mathrm{d}h = H - 3193$ |
| Rola Co | 169.83 | 347 | 0.7842 (18) | 0.9403 (96) | $S = -0.88\,\mathrm{d}h^2 + 14.87\,\mathrm{d}h + 115.59$, $\mathrm{d}h = H - 4816$ |
| Salt Lake | 144.3 | 206 | 0.9344 (16) | 0.9858 (32) | $S = 0.16\,\mathrm{d}h^2 + -0.69\,\mathrm{d}h + 37.42$, $\mathrm{d}h = H - 4430$ |
| Salt Water Lake | 212.47 | 347 | 0.9086 (27) | 0.9494 (151) | $S = -0.82\,\mathrm{d}h^2 + 17.31\,\mathrm{d}h + 133.71$, $\mathrm{d}h = H - 4901$ |
| Selin Co | 2300.49 | 179 | 0.9777 (70) | 0.945 (22) | $S = 1.05\,\mathrm{d}h^2 + 45.86\,\mathrm{d}h + 1754.31$, $\mathrm{d}h = H - 4536.4$ |
| Tangra Yumco | 862.94 | 100 | 0.9155 (18) | 0.8072 (11) | $S = 0.94\,\mathrm{d}h^2 + -0.28\,\mathrm{d}h + 862.94$, $\mathrm{d}h = H - 4536$ |
| Taro Co | 485.15 | 268 | 0.8903 (39) | 0.9576 (19) | $S = 0.18\,\mathrm{d}h^2 + 4.97\,\mathrm{d}h + 477.32$, $\mathrm{d}h = H - 4567.3$ |
| Tu Co | 448.64 | 257 | 0.9276 (41) | 0.9875 (26) | $S = 0.02\,\mathrm{d}h^2 + 4.91\,\mathrm{d}h + 396.59$, $\mathrm{d}h = H - 4926$ |
| Urru Co | 356.35 | 260 | 0.71 (49) | 0.8994 (27) | $S = 1.35\,\mathrm{d}h^2 + 2.67\,\mathrm{d}h + 345.34$, $\mathrm{d}h = H - 4553$ |
| Wulanwula Lake | 652.08 | 225 | 0.9679 (81) | 0.9285 (10) | $S = 2.05\,\mathrm{d}h^2 + 16.49\,\mathrm{d}h + 513.15$, $\mathrm{d}h = H - 4856$ |
| Xijir Ulan Lake | 463.36 | 316 | 0.9736 (84) | 0.9691 (44) | $S = 0.93\,\mathrm{d}h^2 + 13.3\,\mathrm{d}h + 366.35$, $\mathrm{d}h = H - 4770.8$ |
| Xuru Co | 209.87 | 144 | 0.5984 (9) | 0.5527 (58) | $S = 0.12\,\mathrm{d}h^2 + 0.22\,\mathrm{d}h + 206.53$, $\mathrm{d}h = H - 4714$ |
| Yamzho Yumco | 549.61 | 398 | 0.9215 (140) | 0.9364 (72) | $S = 0.51\,\mathrm{d}h^2 + 9.63\,\mathrm{d}h + 531.79$, $\mathrm{d}h = H - 4436$ |
| Yelusu Lake | 203.4 | 486 | 0.7014 (21) | 0.8352 (92) | $S = 14.84\,\mathrm{d}h^2 + -5.77\,\mathrm{d}h + 185.15$, $\mathrm{d}h = H - 4686.5$ |
| Yibug Caka | 178.22 | 118 | 0.9206 (12) | 0.9615 (25) | $S = -1.25\,\mathrm{d}h^2 + 15.79\,\mathrm{d}h + 147.03$, $\mathrm{d}h = H - 4558.5$ |
| Zhari Namco | 1000.18 | 143 | 0.9177 (164) | 0.8388 (52) | $S = 2.66\,\mathrm{d}h^2 + 10.07\,\mathrm{d}h + 962.57$, $\mathrm{d}h = H - 4610$ |
| Zhuonai Lake | 160.1 | 260 | 0.9528 (11) | 0.973 (21) | $S = 0\,\mathrm{d}h^2 + 10.06\,\mathrm{d}h + 124.29$, $\mathrm{d}h = H - 4742$ |
| Zige Tangco | 238.67 | 171 | 0.9008 (186) | 0.976 (24) | $S = 0.06\,\mathrm{d}h^2 + 4.62\,\mathrm{d}h + 212.71$, $\mathrm{d}h = H - 4565$ |

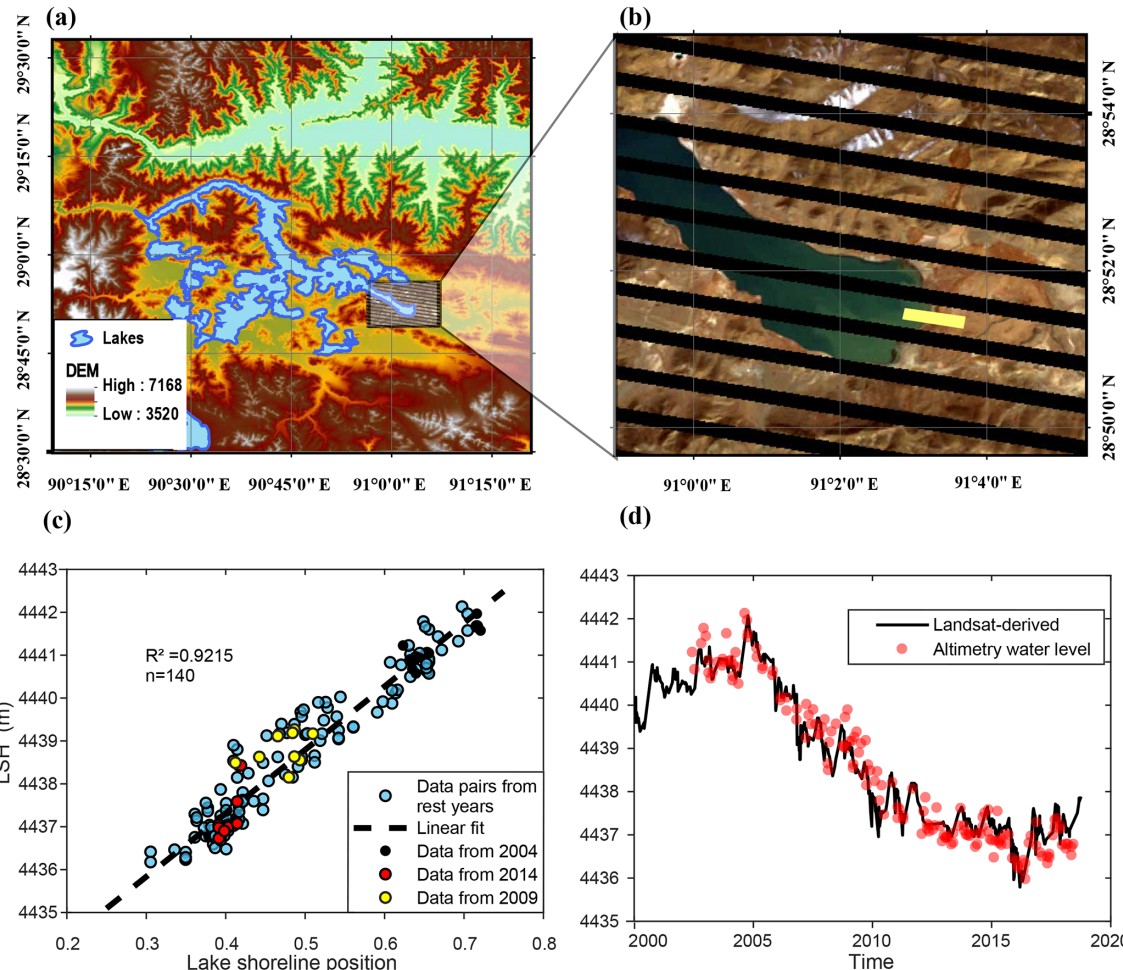

**Figure 4. (a)** Yamzhog Yumco and its surroundings. The DEM was extracted from the SRTM Global 90 m DEM product (Jarvis et al., 2008). **(b)** Region of interest (ROI, yellow area) selected from a Landsat 7 ETM+ image for detecting changes in lake shoreline (black areas represent gaps in the image). **(c)** Linear regression of lake shoreline positions that are represented by water area ratios in the ROI and altimetry water levels for Yamzhog Yumco. **(d)** Landsat-derived water levels and altimetry water levels for Yamzhog Yumco.

prises ∼ 50 % water and ∼ 50 % land, the performance of the method is good, as the distribution of digital numbers of the grayscale image is close to the assumption of the bimodal histogram implicit in the Otsu algorithm (Kittler and Illingworth, 1985; Otsu, 1979). The binarized images were further processed to provide the water area ratio in the ROI, which represents the lake shoreline position. The lake shoreline position time series were then converted into Landsat-derived water levels using linear regression or second-order polynomial fit with altimetry-derived water levels (Fig. 4c–d). For most cases, we only used linear regression, and we performed the second-order polynomial fit only for 2 lakes with Jason-1/2/3 data, because a higher-order regression requires more input information to ensure the reliability. However, cloud, cloud shadow, and shifting gaps may contaminate the ROI and cause errors in the Landsat-derived water levels. Therefore, the QA band of the Landsat surface reflectance product was used to filter the images. Data were excluded if the fraction of the cloud or cloud-shadow-covered area in the ROI was higher than 5 %. For every Landsat 7 ETM+ image acquired after 2003, the pixel number of the ROI was counted and compared with those acquired before 2003. If the loss of pixels exceeded 2 %, the ROI was considered to be affected by a gap and the data were consequently excluded from the subsequent analysis.

A critical function of Landsat-derived water levels is to aid in merging altimetry water levels when there was no overlap period between altimeters or the overlap period was too short. For lakes without Jason-1/2/3 data, lake shoreline positions were firstly translated into Landsat-derived water levels by fitting with CryoSat-2 data functioning as extrapolation of CryoSat-2 to 1–2 years. Then, we applied the same method of merging Jason-1/2/3 to merge the extrapolated CryoSat-2 data with either Envisat or ICESat data. In doing so, we were able to remove all systematic biases between multisource altimetry water levels. After merging the altimetry water lev-

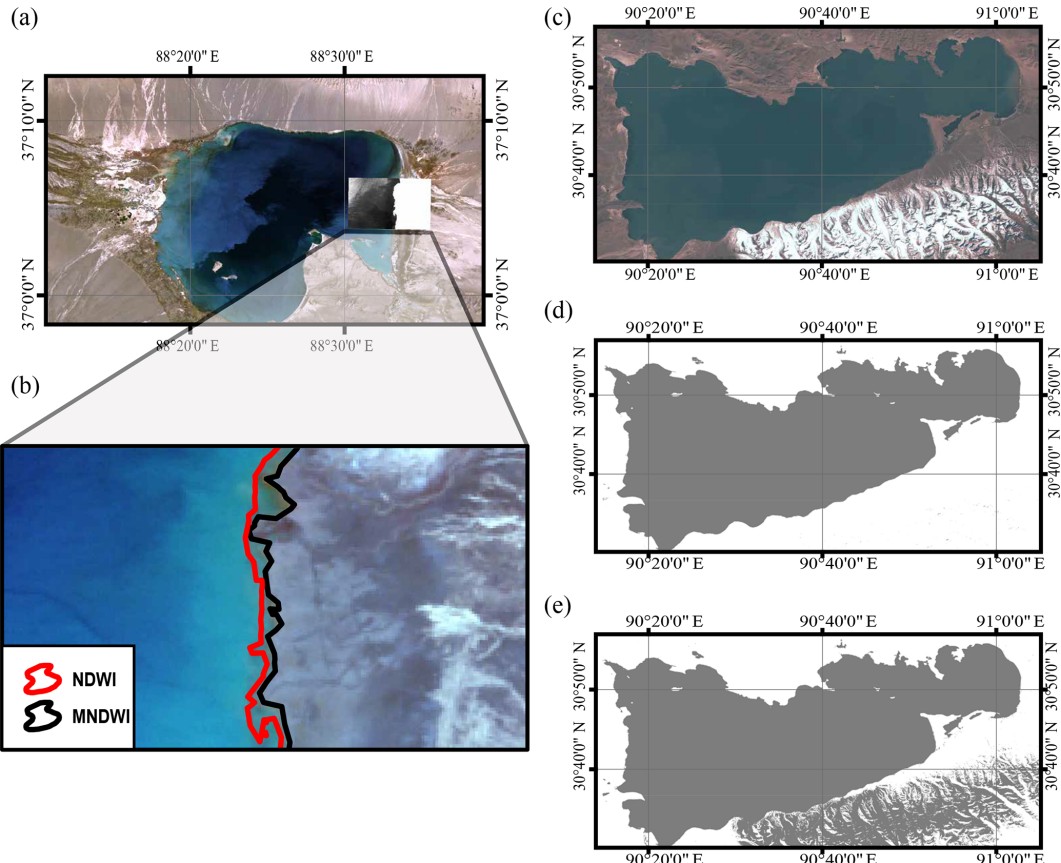

**Figure 5. (a)** A Landsat 7 ETM+ image of Lake Aqqikkol acquired in summer in 2001; **(b)** water area extractions using the modified normalized difference water index (MNDWI) and the normalized difference water index (NDWI), showing that the MNDWI performs better in detecting shallow water; **(c)** a Landsat 8 OLI image of Nam Co acquired in winter in 2015; **(d)** water area extraction using the NDWI, showing good performance in distinguishing water from snow; and **(e)** water area extraction using the MNDWI, showing some confusion of water and snow.

els, we performed regression analysis for the second time between the Landsat-derived water levels and merged altimetry water levels to check if the linear relationship is stable during the entire study period and at different elevations. If the linear relationship was stable, the Landsat-derived water levels would be merged with the altimetry water levels using the linear fitting parameters from the second regression analysis. Otherwise, there may have been a change in the lake bank slope, and, therefore, the extrapolation of CryoSat-2 data with Landsat-derived water levels was not suited. In this case, we reselected the ROI to extract lake shoreline positions and redid altimetry data merging until the Landsat-derived water levels and merged altimetry water levels agreed well with one another in the second linear regression. Detailed analysis about the potential extrapolation issue can be found in the Supplement.

In summary, the basic idea of removing systematic biases of different altimetry water levels is to calculate the means of two altimetry water level time series during the overlap period. The difference between the means is removed from one altimetry water level time series to make both altimetry water level time series consistent and to form a longer time series. This process was subsequently applied to all water level time series with overlap periods to merge them into a single time series for each lake. However, the overlap period may not be long enough, such as Envisat and CryoSat-2 (e.g., there are limited data points (e.g., 1–2) during the overlap period), or does not exist at all, such as ICESat and CryoSat-2. On these cases, Landsat-derived water levels are used to extend or create an overlap period that links two altimetry water level time series.

### 3.3 Hypsometry

We derived the hypsometric curve for each lake by polynomial fitting of the lake area and level time series. The lake area comprises two parts: the inner invariable part and the outer variable part. As the variable water area was of more concern in this study, ROIs for extracting changes in lake area only cover the lake shoreline and its neighboring areas as shown in Fig. 6. The inner part of the water body was

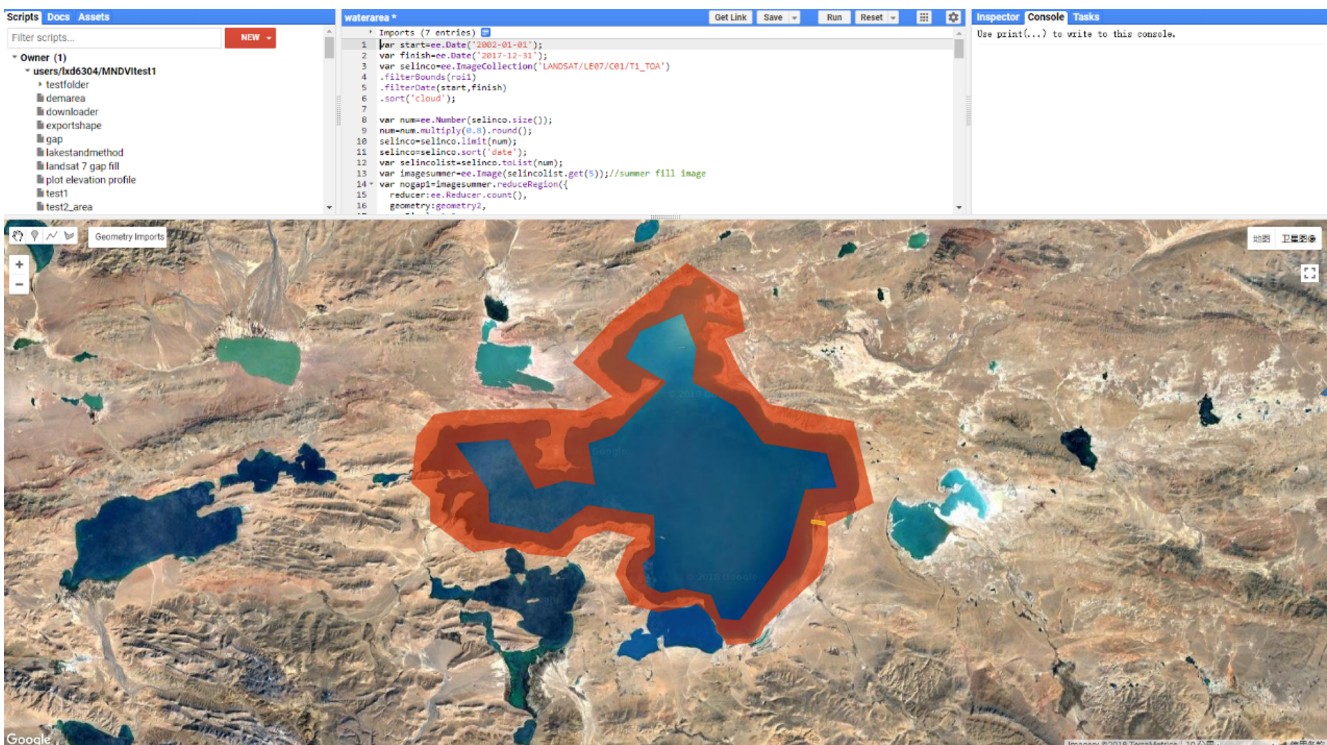

**Figure 6.** Programming interface of Google Earth Engine©. The red polygon is the region of interest for lake area change extraction of Selin Co.

calculated only once and considered invariant, making the calculation more efficient on GEE. Meanwhile, more images are available as the area of ROI becomes smaller, because the possibility of clouds covering the ROI is reduced compared with an ROI covering the entire lake. Landsat 7 ETM+ images after 2003 were not included in this part of the calculation as gaps negatively affected the ROI for lake area extraction. Similar to Sect. 3.2, we selected images with less than 5 % cloud cover on an ROI to generate time series of changes in lake area, obtaining 20–30 data points on average for regression. $R^2$ values for each lake are listed in Table 3, indicating that most lake basins agree well with the parabolic hypsometric curve.

## 4   Validation of data quality

### 4.1   Field experiment

Most Tibetan lakes are located in remote and inaccessible regions, resulting in the scarcity of ground-based in situ measurements that are vital for data quality assessment. We made some in situ measurements in two lakes to validate the data quality of Landsat-derived water levels. The data quality of satellite altimetry on lakes or rivers has been widely investigated, and thus it is beyond the scope of our study. Many studies used in situ water levels to calculate statistical metrics, e.g., root mean squired error (RMSE). However, results

provided by different studies vary, which could be associated with the cross-section width of the study water body in the ground track panel (Nielsen et al., 2017). This means that these results may not be comparable due to their unique applications. In addition, it is not rigorous to use in situ data of only one lake to represent the overall performance in the uncertainty assessment for altimetry water levels. Instead, we used the standard deviation of valid footprints acquired in the same cycle as an estimate of uncertainty in satellite altimetry water levels. In contrast, the applicable condition of Landsat-derived water levels is not so variable as that of satellite altimetry data. Derivation of Landsat-derived water levels requires a relatively flat bank as well as some altimetric information, which were available in all lakes. Since these selected bank slopes were similarly small ($\sim 1/30$), it was possible to use a few typical lakes to represent all lakes. Therefore, we carried out a field experiment (Fig. 7) in Yamzhog Yumco and Nam Co to validate the Landsat-derived water levels.

There were two main goals in our experiments: (1) collecting daily in situ water level data in a TP lake to validate the corresponding Landsat-derived water levels statistically and (2) testing the performance of extracting lake shoreline positions from high-resolution optical images (GF-2) to provide a theoretical uncertainty analysis of Landsat-derived water levels. On Yamzhog Yumco, we installed a pressure-type water level sensor (type H5110-DY, manufactured by

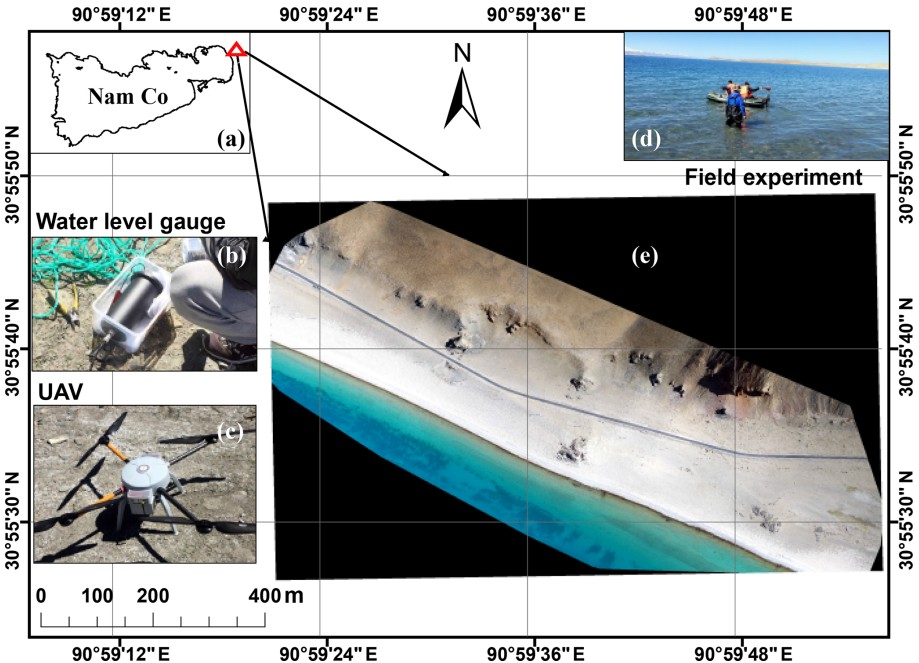

**Figure 7.** Field experiments in two typical lakes: **(a)** an overview map of the experiment spot; **(b)** pressure-type water level sensor; **(c)** unmanned aerial vehicle (UAV); **(d)** installation of the water level sensor; and **(e)** UAV image of a portion of the bank of Nam Co.

Shenzhen Hongdian Technologies Co., Ltd.), which measured water pressure and temperature at the installation depth that were converted into water depths with a relative accuracy of $\sim 0.1\,\%$. The device was carried onto the lake and put $\sim 20$ m below the water surface and 0.5 m above the lake bottom, suggesting an absolute error of $\sim 2$ cm. As for GF-2 images, the spatial resolution of the panchromatic band is 0.8 m, which is able to provide a very accurate reference of lake boundaries for assessing water classification results for Landsat images. We used three GF-2 images acquired at different seasons (two in July and September 2015 and one in February 2016) and different places on the TP to better represent the local conditions when extracting Landsat-derived water levels or areas. Image coregistration was performed to make sure that there was no obvious spatial shift between the GF-2 images and corresponding Landsat images. The accuracy of the image coregistration was $\sim 2$ m.

## 4.2 Uncertainty analysis of Landsat-derived water levels

Based on the in situ water level measurements made by the pressure-type water level sensor, we evaluated the accuracy of Landsat-derived water levels statistically. We first calculated anomalies of in situ water levels and Landsat-derived water levels, and then water levels from the two sources acquired on the same day were used for analysis. There were 16 Landsat-derived water level records available for the comparison against the in situ measurements, indicating an RMSE of the water level anomaly of 0.11 m. The linear fit shows a slope close to 1 and an $R^2$ of 0.89, suggesting the con-

sistency between the in situ water level measurements and the Landsat-derived water levels (Fig. 8b). It should be noted that the Landsat-derived water levels used for validation here were translated from lake shoreline positions using parameters derived from fitting with CryoSat-2 data, i.e., there is no in situ information involved in generating the Landsat-derived water levels shown in Fig. 8.

Furthermore, we performed a theoretical uncertainty analysis of the Landsat-derived water levels by looking at the original optical data and the generation process with the help of high-resolution GF-2 images. First, we took GF-2 images (after coregistration with the Landsat image for the same period and after the coregistration errors were $\sim 2$ m) as the ground truth to determine the accurate position and shape of the lake shoreline. Second, we performed water classification from the Landsat 8 OLI image for the same period jointly using the water index method and Otsu algorithm to derive the binarized image. Landsat image pixels where the lake shorelines from the GF-2 images cross were delineated and marked as shoreline pixels as shown in Fig. 9a. Then the water area in each shoreline pixel was calculated.

Given that these shoreline pixels were classified as either water or land, a relationship between the water area ratio of the shoreline pixel and the probability of the pixel being classified as water can be derived. This relationship generally describes the function of the water classification method by telling how likely a pixel is to be determined as water, given the water area ratio of the pixel. Based on the observations of a total of 4128 Landsat shoreline pixels, a power function

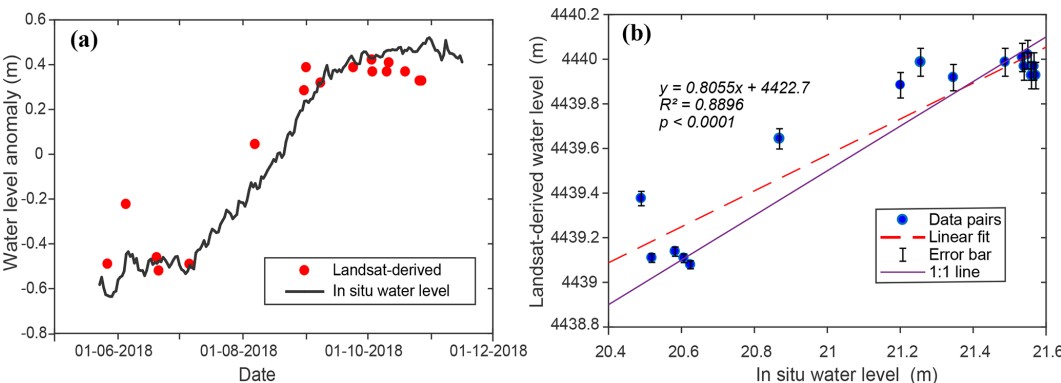

**Figure 8. (a)** In situ water level anomaly versus Landsat-derived water level anomaly in Yamzhog Yumco; **(b)** linear regression between the Landsat-derived water levels and in situ water levels during the same period.

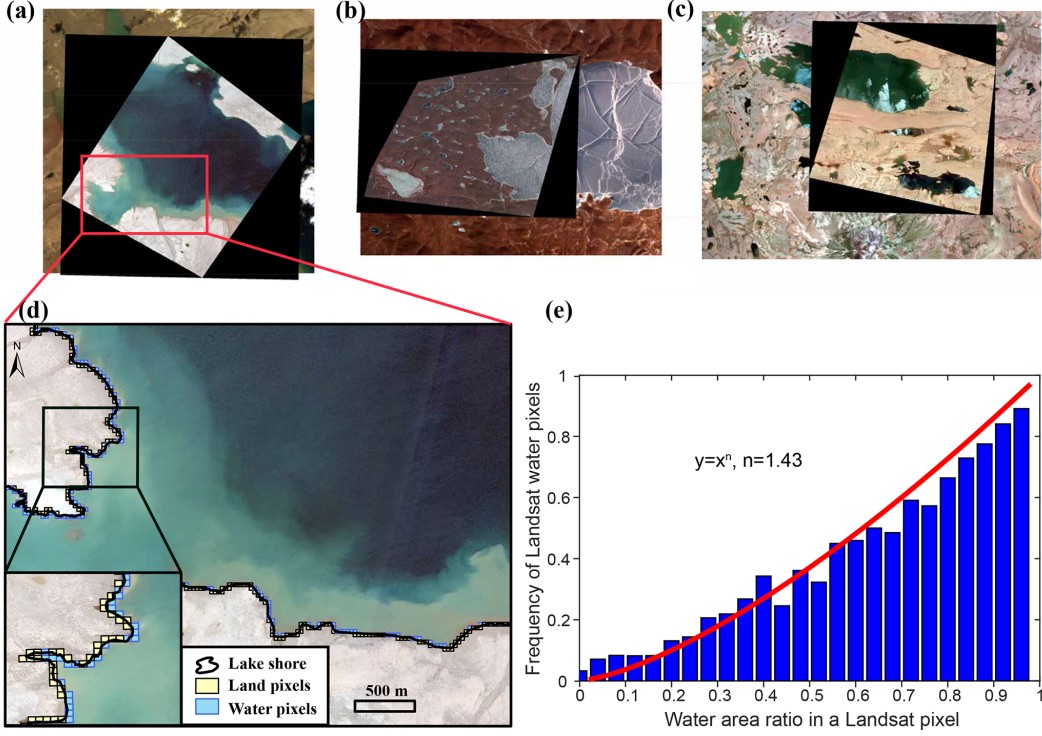

**Figure 9. (a–c)** GF-2 images (upper layer) and corresponding Landsat 8 OLI images (bottom layer) acquired on 7 September 2015, 29 January 2016, and 5 July 2015; **(d)** Landsat 8 OLI shoreline pixels (the background is the GF-2 image) – blue pixels were classified as water, and yellow pixels were classified as land; **(e)** the relationship between the water area ratio in a pixel and the frequency of the pixel being classified as water. Blue bars are sampled at a 0.04 bin space from the 4128 pixels. The red line shows the fitting curve based on the maximum likelihood method.

was chosen to represent the water classifier as Eq. (4) shows:

$$f(x) = x^n, \tag{4}$$

where $x$ represents the water area ratio in the shoreline pixel, $f(x)$ represents the probability of the shoreline pixel being classified as the water pixel, and $n$ is the parameter that determines the shape of the curve. Parameter $n$ was determined using the maximum likelihood method (Fig. 9e).

As expected, the probability of the pixel being classified as water increases with the water area ratio in the pixel (Fig. 9e). The enclosed area of the fitting curve ($y = x^{1.43}$) is smaller than that of $y = x$ on [0, 1], suggesting that there may be a lower probability of the occurrence of water pixels that is associated with a systematic bias of the lake shoreline detection. Note that the systematic bias can be removed when linearly fitting the lake shoreline positions and altimetry wa-

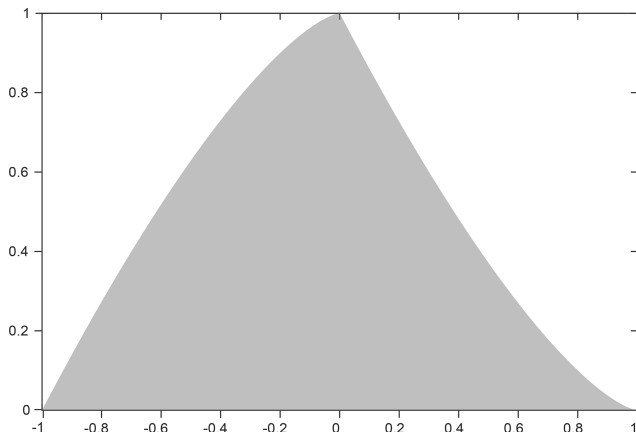

**Figure 10.** $F(X)$: probability density function of the bias ($X$) between the classified water ratio ($X_1$) and real water ratio ($X_0$) in a shoreline pixel.

ter levels as long as the bias is stable. Therefore, uncertainty in Landsat-derived water levels developed in this study arises mainly from the variation in this systematic bias.

To describe the variation in the systematic bias, a new random variable $X$ was introduced to represent the bias between the classified water area and the real water area in a shoreline pixel. Given the shape and position of the lake shoreline, the real water area in each shoreline pixel is a complex function of the relative position between the pixel and the shoreline. To simplify the derivation, we assumed that the water area ratio in a shoreline pixel is uniformly distributed on [0,1], meaning that the probability of any value between 0 and 1 is equal. If we use $X_0$ to represent the true water area ratio in the shoreline pixel and $X_1$ to represent the classified results based on the water area ratio, the random variable $X$ can be expressed as

$$X = X_1 - X_0,  \tag{5}$$

where $X_1$ can take on 1 or 0 (i.e., the classified results only tell us whether a pixel is a water pixel or not), so $X$ can only take on either $1 - X_0$ or $-X_0$. Because the range of $X_0$ is [0,1], it is obvious that the range of $X$ is $[-1,1]$. A derivation of $F(X)$, i.e., the probability density function (PDF) of $X$, can be found in the Supplement (Part 2).

Overall, $F(X)$ describes how the bias between the classified water ratio and real water ratio in shoreline pixels is distributed as shown in Fig. 10. If there are $N$ shoreline pixels in an ROI, we can take them as $N$ independent observations of $X$ and calculate the mean value $\overline{X}$. This value $\overline{X}$ can represent an average shift of the detected lake shoreline from the real lake shoreline in the unit of 1-pixel width (30 m). As we mentioned above, the systematic bias can be removed in the regression between the lake shoreline positions and altimetry water levels. As such, it is the variation of the bias that determines the accuracy of the Landsat-derived water levels. We can calculate the standard variation of $\overline{X}$ to represent the

uncertainty in lake shoreline position. Note that there is a simple relationship between $\sigma_{\overline{x}}$ and $\sigma_x$:

$$\sigma_{\overline{x}} = \frac{\sigma_x}{\sqrt{N}}.  \tag{6}$$

One only needs to calculate $\sigma_x$:

$$\overline{X} = \int_{-1}^{1} F(X) \cdot X \, dX \approx -0.09,  \tag{7}$$

$$\sigma_x = \sqrt{\int_{-1}^{1} F(X) \cdot (X - \overline{X})^2 \, dX} \approx 0.39.  \tag{8}$$

Combined with Eqs. (4) and (7), Eq. (8) was resolved numerically, resulting in $\sim 0.39$-pixel width. Substituting $\sigma_x$ in Eq. (6) with Eq. (8) gives

$$\sigma_{\overline{x}} = \frac{0.39}{\sqrt{N}}.  \tag{9}$$

If the slope of the shoreline is known, e.g., $\tan\theta$, the uncertainty of the Landsat-derived water level can be expressed as

$$\sigma_{ho} = \sigma_{\overline{x}} \cdot d \cdot \tan\theta = \frac{0.39 \times 30 \times \tan\theta}{\sqrt{N}},  \tag{10}$$

where $\sigma_{ho}$ is the uncertainty of Landsat-derived water levels and $d$ is the spatial resolution of the satellite image (30 m). In this study, a typical width of ROI for deriving Landsat-derived water levels is $\sim 10$-pixel width, meaning that $N$ is $\sim 10$. In addition, lake shores used for generating Landsat-derived water levels here generally have a relatively mild slope of $\sim 1/30$ or even smaller, which can be roughly estimated from the maximum shoreline change and altimetry water level change within a year. Here if we use $1/30$ as the slope $\tan\theta$, the uncertainty of the Landsat-derived water levels can ultimately be estimated to be $\sim 0.12$ m, which is very close to the RMSE of 0.11 m based on the comparison between the optical water levels and in situ water level measurements mentioned earlier.

However, for most cases we do not know the exact lake bank slope $\tan\theta$, which is the reason why we performed the regression analysis between the lake shoreline positions and altimetry-derived water levels. Information on the real lake bank slope is implicitly expressed in the linear fitting slope $\beta$ (if the fitting line is $y = \beta x + \alpha$). Uncertainty in altimetric information could evolve into the fitting parameters and impact the accuracy of the generated Landsat-derived water levels. Given that the observed lake shoreline position is $X_1$ (e.g., $X_1 = 5.6$, meaning that the observed lake shoreline position is 5.6 Landsat pixels away from the initial position corresponding to the lowest water level, which is different from Eq. (5), $X_1$ here can be a rational number because it is determined by averaging all shoreline pixels in the ROI, whereas

in Eq. 5 we focused on only 1 shoreline pixel), combining Eq. (5), the Landsat-derived water level ($y$) can be expressed as

$$y = \beta\left(X_1 - \overline{X_1}\right) + \overline{Y} = \beta\left(X_0 - \overline{X_0}\right) + \beta\left(X - \overline{X}\right) + \overline{Y}, \quad (11)$$

where $\left(X_1 - \overline{X_1}\right)$ denotes the observed lake shoreline change (in the unit of a Landsat pixel), $\overline{X_1}$ denotes the mean of observed lake shoreline positions used for linear regression, $\overline{Y}$ denotes the mean of altimetry water levels used for linear regression, $\left(X_0 - \overline{X_0}\right)$ denotes the real lake shoreline change, $\left(X - \overline{X}\right)$ denotes the variation of the Landsat-derived shoreline position caused by the water extraction method, and $\beta$ is the linear fitting slope. It is obvious that the expected value $\left(X - \overline{X}\right)$ is 0. As we discussed earlier, a systematic bias does not affect the accuracy of the Landsat-derived water level but the variation of the systematic bias does.

Based on Eq. (11), the overall uncertainty of the Landsat-derived water level $\sigma_y$ can be given as

$$\sigma_y = \sqrt{\sigma_\beta^2\left(\frac{\partial y}{\partial \beta}\right)^2 + \sigma_{\overline{x}}^2\left(\frac{\partial y}{\partial\left(X - \overline{X}\right)}\right)^2 + \sigma_{\overline{Y}}^2\left(\frac{\partial y}{\partial \overline{Y}}\right)^2}$$
$$= \sqrt{\sigma_\beta^2(X_1 - \overline{X_1})^2 + \sigma_{\overline{x}}^2\beta^2 + \sigma_{\overline{Y}}^2}, \quad (12)$$

where $\beta$ and $\sigma_\beta$ can be derived from the linear regression analysis, $\sigma_{\overline{x}}$ is given in Eq. (9), and $\sigma_{\overline{Y}}$ is the uncertainty of the mean altimetry water level which can be estimated from the altimetry data. For a typical lake like Yamzhog Yumco, $\beta = 0.35\,\text{m}$, $\sigma_\beta = 0.02\,\text{m}$, $\text{Max}(|X_1 - \overline{X_1}|) = 11$, $\sigma_{\overline{x}} = 0.13$, and $\sigma_{\overline{Y}} = 0.015\,\text{m}$, which gives a maximum $\sigma_y$ of $0.22\,\text{m}$. Note that $\left(X_0 - \overline{X_0}\right)$ is assumed to be the ground truth so there is no error associated with this term. This relationship shows that the uncertainty in the Landsat-derived water level increases with the distance from the center point $(\overline{X_1}, \overline{Y})$ represented by $(X_1 - \overline{X_1})^2$. Interpretation of this phenomenon is that extrapolation of Landsat-derived water levels (far from the center point) may cause some errors and should be used with caution. More detailed discussion on the extrapolation can be found in the Supplement.

Overall, the uncertainty quantification of the Landsat-derived water levels developed in this study indicates clearly that the accuracy of Landsat-derived water levels depends on the width of an ROI, e.g., the number of pixels/observations, slope of the lake shore, the effectiveness of the water classification method, and the uncertainty in the altimetry water level used for regression. One of the advantages of the Landsat-derived water level is that an ROI does not necessarily cover a large area of lake shores, which maximizes the potential of optical remote sensing images to increase the spatial coverage and temporal resolution of lake water level estimates that may not be realized by using satellite altimetry alone. Optical remote sensing images provide important complementary information on altimetry water levels and can subsequently facilitate lake water storage estimation.

## 4.3 Cross validation with similar products

We compared our product with a widely used lake water level/storage data set provided by the LEGOS Hydroweb, indicating that the two products are, on the whole, consistent with each other (shown in Fig. 11), but our product may perform better in terms of the temporal continuity as well as the temporal resolution (shown in Sect. 6.2). Both advantages are important in improving our understanding of responses of lakes to climate change. There are 21 lakes that are the same in both our study and LEGOS Hydroweb. Annual trends in water level and lake storage during 2003–2015 are compared in Fig. 11, indicating the overall consistency of the two products in terms of $R^2$ of the linear fit.

## 4.4 Data set description

The data sets cover 52 large lakes (50 lakes with a surface area larger than $150\,\text{km}^2$ and 2 lakes that are $100$–$150\,\text{km}^2$) on the TP. The data sets consist of two parts: (1) a table containing hypsometric curves and corresponding regression statistics ($R^2$ and the number of data pairs) for each lake, with parameters of the hypsometric curves listed in separate columns for the convenience of batch processing; and (2) time series for each lake archived as 52 entities with geographic information (i.e., latitude, longitude, and size of the lake) that can be checked in an online map provided by PANGAEA, avoiding the confusion of lake names. The time series of each lake include lake water levels and lake storage changes.

For data points in the water level time series, satellite or sensor type is shown (i.e., from Jason-1/2/3, Envisat, ICESat, CryoSat, or optical images). Uncertainty was calculated using the standard deviation of valid footprints in the cycle (only for altimetry data). The lake water storage time series were transformed from water level time series using the hypsometric relationship so that they have equal data size. The lake water storage time series represent changes in lake storage with respect to a reference water level, which is listed in the corresponding hypsometric curve table as a parameter. The overall uncertainty of Landsat-derived water levels within the regression range (the range of altimetry water levels) is $0.1$–$0.2\,\text{m}$ based on the experiment and analysis in this paper. The extrapolation of Landsat-derived water levels may occur during the time gap between altimetric missions and before 2002. The average uncertainty of altimetry water levels is $0.11\,\text{m}$.

## 5 Applications

### 5.1 Spatiotemporal analysis of changes in lake water storage in the Tibetan Plateau

Based on the lake water storage changes we derived, spatial patterns of lake storage trends during 2000–2017 were

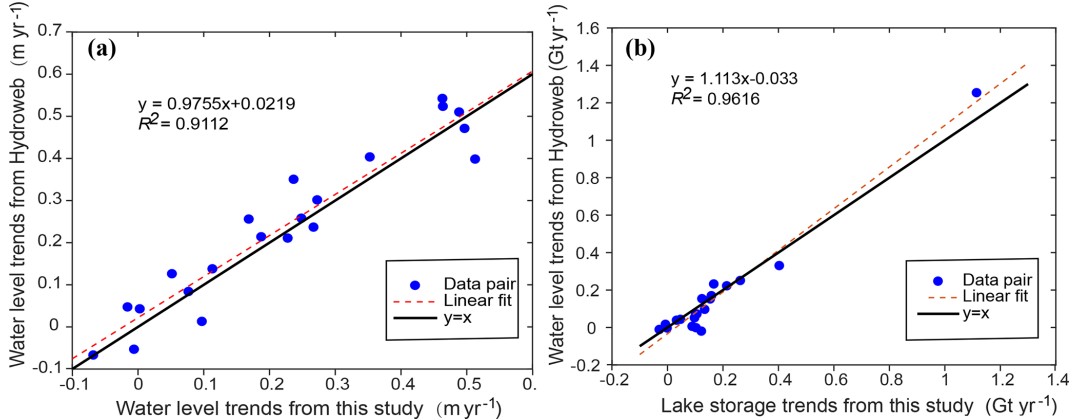

**Figure 11.** Cross validation of the TP lake level and storage changes derived from our study with those provided by the LEGOS Hydroweb database (Crétaux et al., 2011a): **(a)** trends in lake water levels from 2003 to 2015 and **(b)** trends in lake water storage from 2003 to 2015.

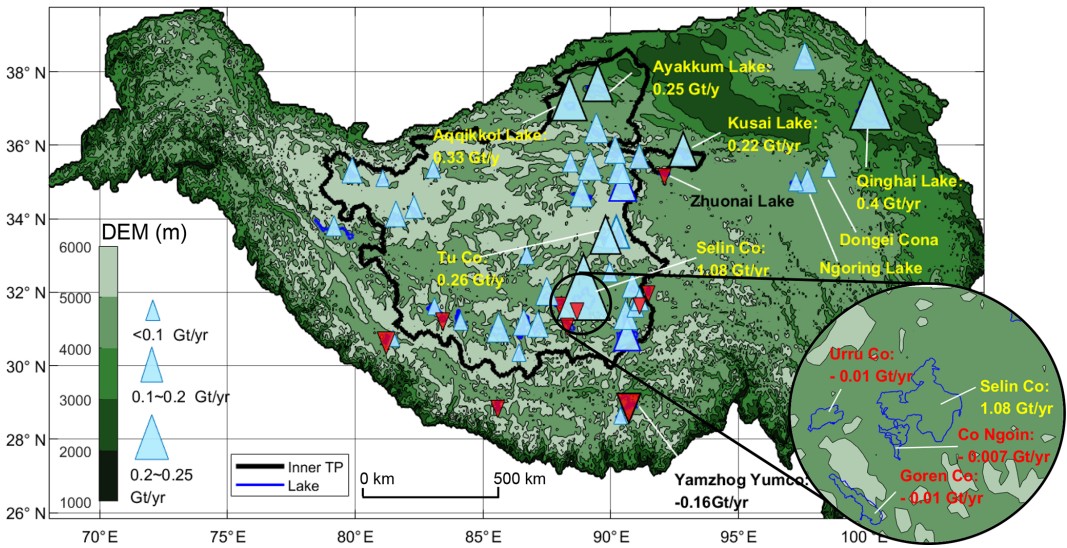

**Figure 12.** Spatial distribution of trends in lake storage on the TP during 2000–2017. The black line shows the boundary of the endorheic basin of the TP including 39 lakes in this study. The other 13 lakes are located outside the endorheic basin.

shown in Fig. 12. In the endorheic basin of the TP, similar to some reported results (Yao et al., 2018b; Zhang et al., 2017a), most lakes have been expanding rapidly; e.g., Selin Co (31.80° N, 89.00° E) gained $19.7 \pm 2.0 \, \text{km}^3$ of water during the study period, and Lake Kusai (35.70° N, 92.90° E) experienced an abrupt expansion due to flood and gained $2.2 \pm 0.2 \, \text{km}^3$ of water in 2011, as reported in related work (Yao et al., 2012). In contrast, some lakes in the southern part of the TP experienced shrinkage, e.g., Yamzhog Yumco (28.93° N, 90.70° E) gained a total of $0.8 \pm 0.4 \, \text{km}^3$ water during 2000–2004 but has shrunk during the remaining 13 years (2005–2017) at a rate of $-0.19 \pm 0.03 \, \text{km}^3 \, \text{yr}^{-1}$. In contrast to Yamzhog Yumco, Lake Qinghai (36.90° N, 100.00° E) lost $2.2 \pm 0.7 \, \text{km}^3$ water during 2000–2004 but gained $7.7 \pm 0.6 \, \text{km}^3$ of water during 2005–2017. Similar patterns can be detected in adjacent lakes of Lake Qinghai, e.g.,

Lake Donggei Cuona (35.28° N, 98.55° E) and Lake Ngoring (34.90° N, 97.70° E).

However, spatial proximity cannot fully explain the intricate trend distribution in the Selin Co basin, where large lakes such as Selin Co were expanding, whereas smaller adjacent lakes showed an opposite decreasing trend, e.g., Urru Co (31.70° N, 88.00° E), Lake Co Ngoin (31.60° N, 88.77° E), and Goren Co (31.10° N, 88.37° E). In fact, we found that the decreasing trends in some small lakes like Goren Co were not detected in Yao et al. (2018b), which is likely due to the lower temporal resolution as shown in Fig. 13. The three shrinking lakes are located in the upstream region and feed Selin Co through two small rivers. One of the rivers links lakes Goren Co, Urru Co, and Selin Co, whereas the other links lakes Co Ngoin and Selin Co.

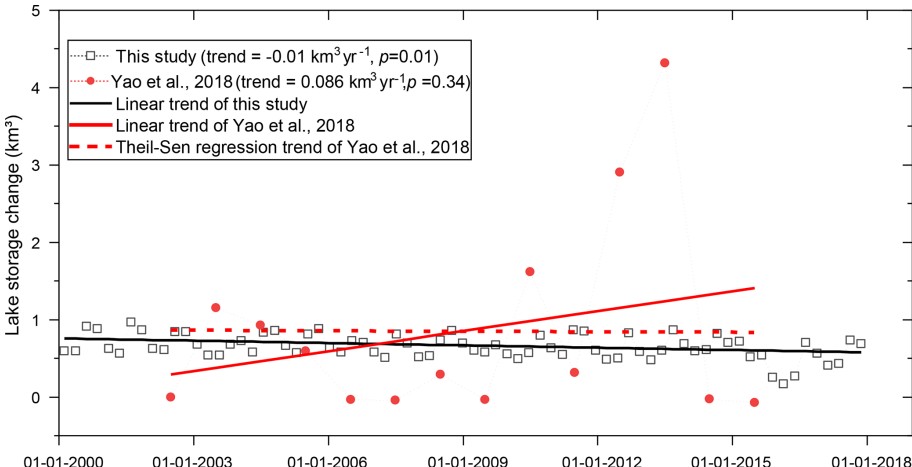

**Figure 13.** Discrepancy of lake storage trends in Goren Co between Yao et al. (2018a) and our study.

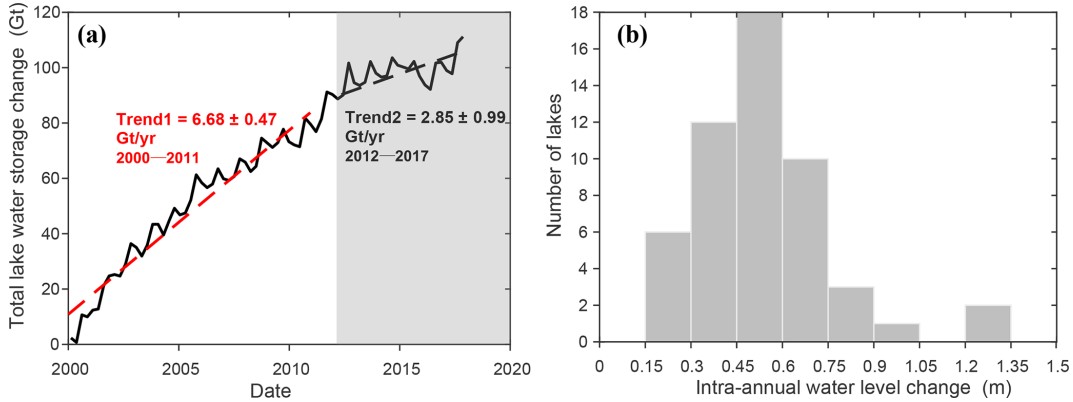

**Figure 14.** **(a)** Total storage changes in the 52 lakes on the TP, which can be generally divided into two stages: (1) a rapidly increasing stage (2000–2011) with a higher increasing rate of $6.68 \, \text{km}^3 \, \text{yr}^{-1}$ and (2) a mildly increasing stage (2012–2017) with an increasing rate of $2.85 \, \text{km}^3 \, \text{yr}^{-1}$. **(b)** Histogram of intra-annual changes in lake water levels of the 52 lakes on the TP.

A possible explanation of the disparity of changes in lake water storage in the Selin Co basin could be the principle of minimum potential energy. If we simplify the basin with the tank model and take the upstream small lake as a tank with a leaking hole, the storage of the small lake is mainly controlled by the height of the leaking hole. Given that surface water of the small lake increased, most of the increased water would flow into the large lake (a lower tank), and the outflow discharge of the small lake at higher elevations would increase accordingly. The height of the leaking hole would decline (erosion) so as to increase the overflow capacity, which eventually results in the decrease in small lake storage. Another possible situation is that the height of the leaking hole remains the same and the water surface height of the small lake increases, but this situation is not consistent with the minimum potential energy principle, as more water potential energy is stored in the small lake. This phenomenon shows that river-lake interactions may cause complex patterns of the regional surface water distribution. Therefore, decreases in

small lake water storage and increases in water storage of Selin Co in the basin detected by our study seem reasonable. Increases in small lake water storage in this basin reported in some published studies may be associated with the sparse sampling of lake water levels.

We averaged the total lake water storage change in each season to generate time series shown in Fig. 14a. The overall storage change in the 52 lakes is $98.3 \pm 2.1 \, \text{km}^3$. The total lake water storage was increasing rapidly during the first 12 years but became relatively stable since 2012. Intra-annual variation in the TP lakes can also be investigated using the densified lake water level time series generated by our study. We removed the linear trend (sometimes there were multiple linear trends for a lake in different periods, which were removed in a stepwise fashion) and calculated the mean monthly water level anomaly for each lake over the study period. Then the intra-annual water level change was represented by the difference between the maximum and minimum values of the monthly water level anomaly. The

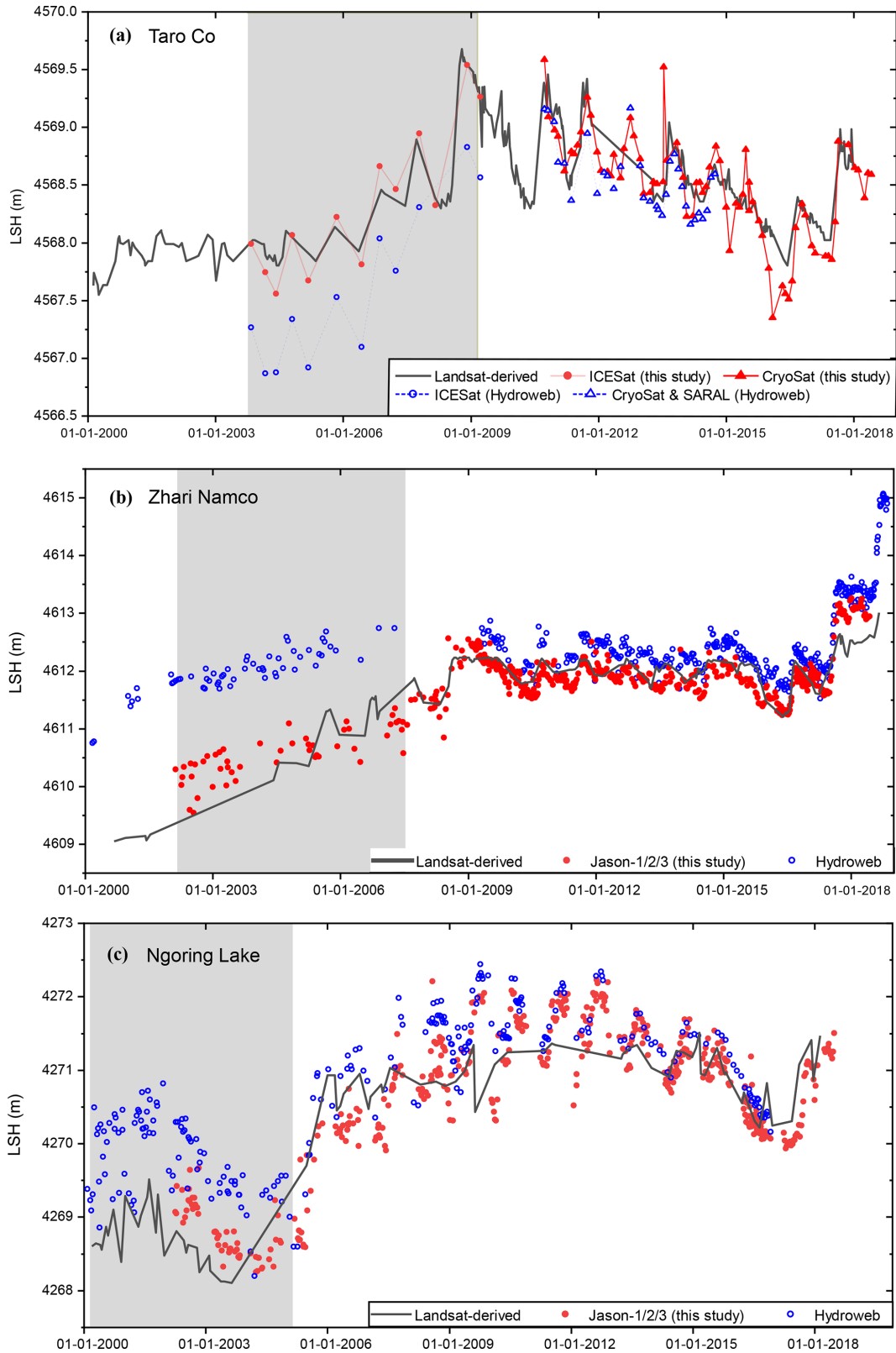

**Figure 15.** Similarities and differences between water level time series from the LEGOS Hydroweb database (Crétaux et al., 2011a) and our study. **(a)** Taro Co (31.14° N, 84.12° E). **(b)** Zhari Namco (30.93° N, 85.61° E). **(c)** Ngoring Lake (34.90° N, 97.70° E). Shading areas highlight the differences between the two data sets. LSH represents lake surface height.

histogram of the intra-annual water level change in Fig. 14b shows that most of the TP lakes have water level variations ranging from 0.3 to 0.75 m in a year on average. Similar work was performed by Lei et al. (2017), but only a small number of lakes were investigated in their study.

## 5.2   Quality assessment of similar data products

Some obvious discrepancies between the two data sources can be noticed, e.g., water levels of Taro Co. Both Hydroweb data and our estimation used ICESat and CryoSat-2 data. The difference lies in the fact that our CryoSat-2 product was more updated with a longer time span but Hydroweb used an additional altimetry satellite SARAL. Because the systematic biases of both products were removed, it is possible that we chose different baselines that resulted in the overall shift as shown in Fig. 15a. For instance, we may use different sets of ellipsoid and geoid models. In addition to the overall shift, some time-dependent discrepancy can be found in Fig. 15, e.g., periods highlighted by shading areas.

The black curve shows the Landsat-derived water level we derived, which is a critical reference for connecting two different altimetry data time series without an overlap period. The Landsat-derived water level shows that the last two samples of ICESat data should not be lower than the first few samples of the CryoSat-2/SARAL data (see the dashed boxes). However, it is apparent that Hydroweb data display a reverse relationship, showing that the last two ICESat measurements are smaller than the first few CryoSat/SARAL measurements. It is likely due to an unremoved systematic bias between ICESat and CryoSat/SARAL time series from Hydroweb data in Taro Co.

Even though the Landsat-derived water levels were generated by linearly fitting the lake shoreline positions with altimetry data, the relative magnitude of water levels during different periods should not be largely affected by the fitting parameters, e.g., if Landsat-derived water levels show that $H_a >= H_b$, where $H_a$ ($H_b$) means water levels acquired in period A (B), the $H_a >= H_b$ relationship would not change with the fitting parameters used to generate the Landsat-derived water levels. This is the main reason for us to use Landsat-derived water levels as reference. Therefore, Hydroweb data may overestimate the increasing trends in the water levels of Taro Co as their ICESat data are $\sim 0.3$ m lower than the SARAL/CryoSat data. A similar issue can be observed in Zhari Namco and Ngoring lakes shown in Fig. 15b–c, and the explanation is similar to that of Taro Co. This problem may also exist in some similar studies when multisource altimetry data without overlap periods were used.

As shown in Fig. 16, optical data can be less noisy than altimetry data in certain lakes and significantly improve the continuity of lake level and storage change monitoring. In addition, a more apparent seasonality in lake level change can be seen from the generated lake level time series. These advantages would largely benefit a better understanding of responses of TP lakes to climate change and facilitate hydrologic modeling of lake basins, regional water balance analysis, and even hydrodynamic analysis of lake water bodies.

## 5.3   Lake overflow flood monitoring

As mentioned earlier in Sect. 5.1, Lake Kusai experienced an abrupt expansion in 2011, resulting from the dike break of an upstream lake (Hwang et al., 2019; Liu et al., 2016; Xiaojun et al., 2012), named Lake Zhuonai (35.54° N, 91.93° E). The outburst of Lake Zhuonai occurred on 14 September (Liu et al., 2016), with $2.47 \pm 0.06$ km³ of water leaking into the Kusai River (as shown in Fig. 17b), the main inflow of Lake Kusai. The water level of Lake Kusai increased by up to $7.9 \pm 0.5$ m within 20 days (from 11 September to 1 October in 2011) based on Jason-2 data and then started to drop as water overflowed from the southeast corner into Lake Haidingnuoer (35.55° N, 93.16° E) and Lake Salt (35.52° N, 93.40° E). Lake Salt, the lowest part of the basin close to the basin boundary, has gained $3.0 \pm 0.1$ km³ of water since 2011 and has become a critical threat to the surrounding residents and railway $\sim 10$ km southeast to the boundary. Note that there are few satellite altimetry data available for Lake Salt except several CryoSat-2 observations, where Landsat-derived water levels can provide a near-real-time monitoring of changes in lake water level and storage that are crucial to flood early warning and risk management.

Aided by the high-temporal-resolution lake water level series, it was possible to estimate the height of the outlet of Lake Kusai, an important parameter for overflow estimation. The overflow of Lake Kusai can help predict the water level rise in Lake Salt and even serve as an indicator of flood forecast, as Jason-3 data with a 10-day revisit cycle are now available on Lake Kusai. Several pairs of Landsat 8 OLI images and lake water levels for the same period were compared to provide a range of possible outlet heights, which are likely to be 4483.9 to 4484.1 m, as shown in Fig. 18a. Then we measured the mean width of the outlet from high-resolution optical images provided by Planet Explorer (Planet, 2017), which is relatively stable in Dec 2011 at $31.5 \pm 2.3$ m in recent years. Given lake water levels and the outlet height and width, an estimation of overflow can be made using the broad crest weir formula:

$$Q = C \cdot b \cdot H^{1.5}\sqrt{2g}, \tag{13}$$

where $C$ is a parameter mainly reflecting geometric characteristics of the weir that mainly varies from 0.3 to 0.4, $b$ is the width of the weir, $H$ is the water head with respect to the top of the weir, and $g$ is the acceleration of gravity.

We determined $C$ ($\sim 0.3$) by using stage 1 shown in Fig. 19 as a calibration period. Details can be found in the supplementary file. Then we applied this result to stage 2 shown in Fig. 19 to estimate the total overflow from Lake Kusai and compared the overflow with total water gain in stage 2

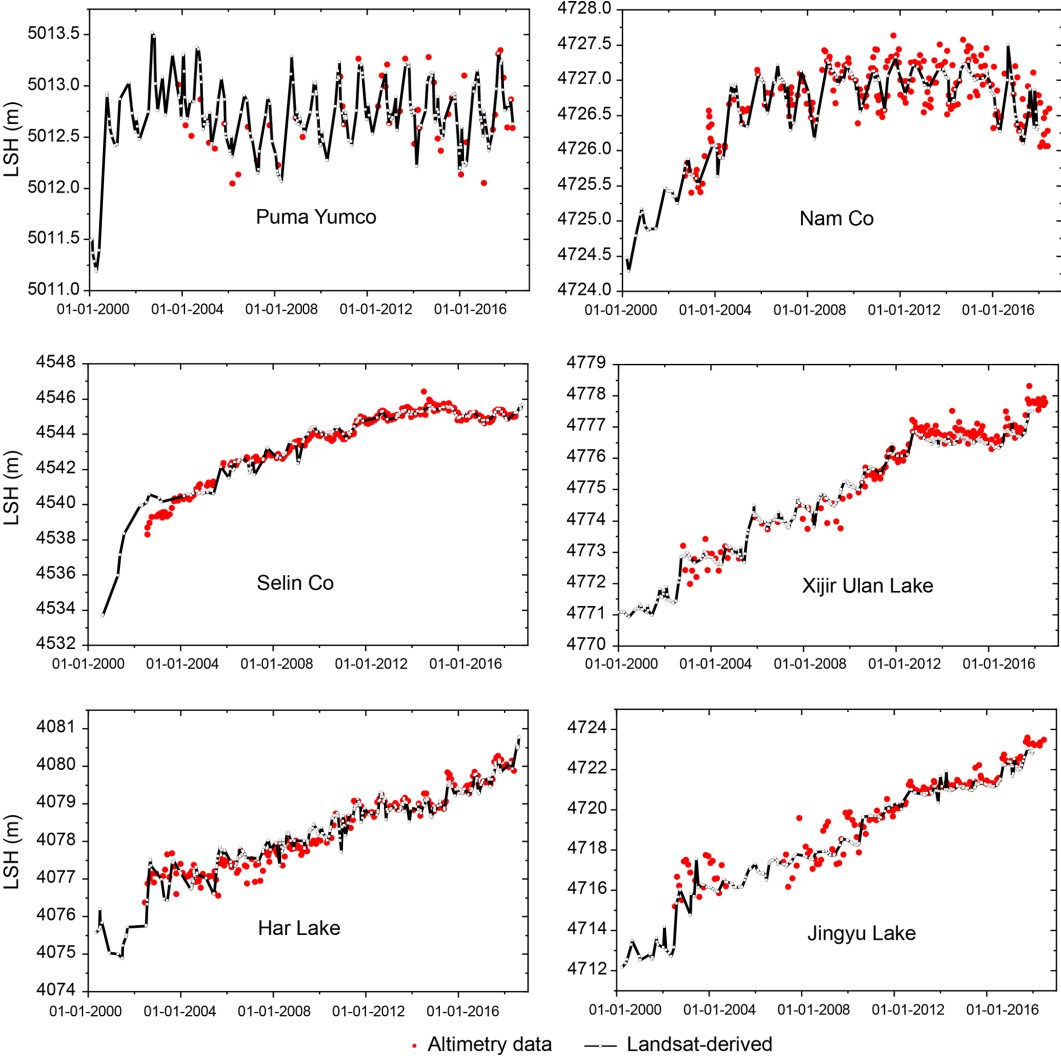

**Figure 16.** Lake water level (left $y$ axis) estimates from our approach for six TP lakes. Black lines represent optical data and red dots represent altimetry data. LSH represents lake surface height.

in Lake Salt. Since Lake Salt mainly relied on the replenishment of Lake Kusai during that period, with little precipitation input and negligible glacier meltwater in winter, the outflow of Lake Kusai can be comparable with the water gain in Lake Salt derived from remote sensing, though there was a small amount of evaporation loss. This relationship can provide a straightforward validation of our developed method. However, it was not available in stage 1, because the outflow of Lake Kusai first replenished Lake Haidingnuoer until the latter began overflowing. Results based on Eq. (13) indicate that the total outflow from Lake Kusai in stage 2 ranged from 0.21 to 0.22 km$^3$, whereas the water gain in Lake Salt from remote sensing was 0.19±0.01 km$^3$. This indicates that our high-temporal-resolution lake water level time series are valuable in monitoring and predicting lake outflow flooding that is crucial for the safety of downstream residents and infrastructure.

## 6   Data availability

The derived TP lake water levels, hypsometric curves, and water storage changes are archived and available at https://doi.org/10.1594/PANGAEA.898411 (Li et al., 2019).

## 7   Conclusion

In this study, we develop high-temporal-resolution (i.e., weekly to monthly timescales) water levels and storage change data sets for 52 large lakes on the TP during 2000–2017 by combining multiple altimetric missions and optical remote sensing images. Generated from lake shoreline positions and regression analysis with altimetry data, the Landsat-derived water level serves as a unique reference covering the entire study period, enabling a more consistent merging of multisource altimetry time series. Multisource al-

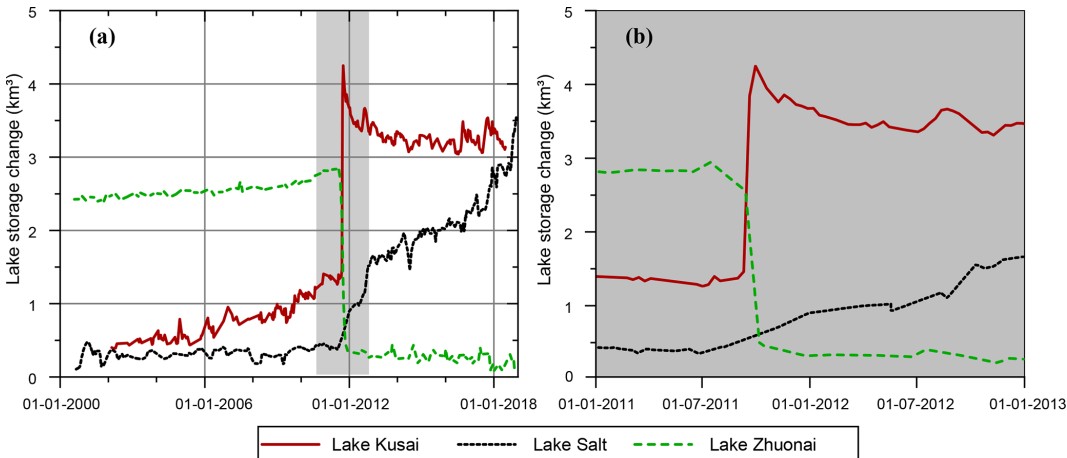

**Figure 17. (a)** Lake storage changes in Lake Zhuonai, Lake Kusai, and Lake Salt corresponding to the outburst event in September 2011 and **(b)** storage changes in relevant lakes during the outburst event (a magnified plot of the shading area in **a**).

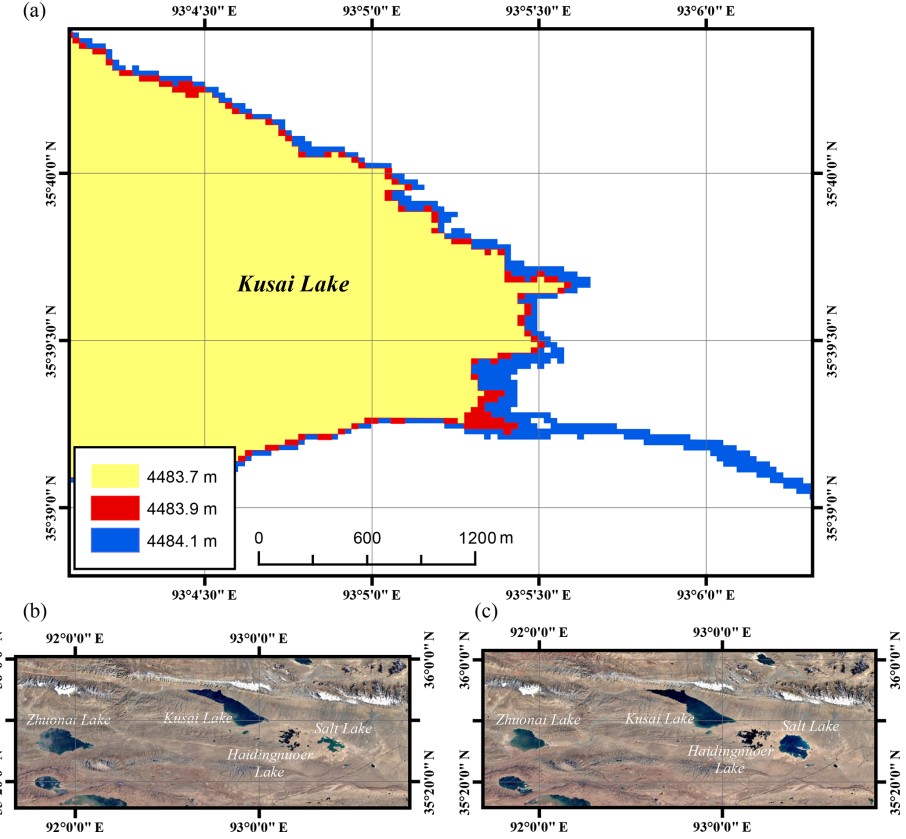

**Figure 18. (a)** Height variations in the outlet of Lake Kusai estimated from Landsat 8 OLI images indicate that the overflow would occur when the water level increased from 4483.9 to 4484.1 m. **(b)** Google Earth© image before the outburst of Lake Zhuonai (December 2010). **(c)** Google Earth© image after the outburst event (December 2013).

timetry water levels are first extracted separately from spaceborne altimetry products and then combined into a longer and denser altimetry water level time series with systematic biases well removed using Landsat-derived water levels as reference. The combined altimetry and Landsat-derived water levels increase the overall sampling frequency to submonthly regardless of the lake size.

By comparison with a widely used LEGOS Hydroweb data set, we show that without Landsat-derived water levels as a reference there may be a remaining bias in the combined

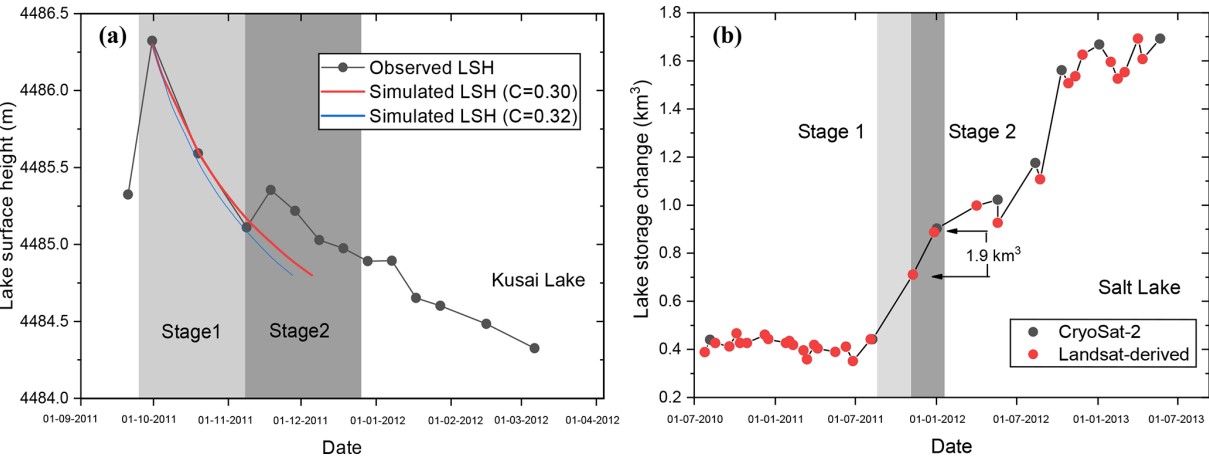

**Figure 19.** Changes in the water level of Lake Kusai after receiving the outburst flood from Lake Zhuonai. Stage 1 was used to determine the range of parameter C in Eq. (13). Stage 2 was used to compared the simulated lake outflow from Kusai Lake based on Eq. (13) with the water gain estimate from remote sensing of Lake Salt downstream during the same period; and **(b)** changes in water storage of Lake Salt derived from remote sensing using our developed method. There was $0.19\,\text{km}^3$ of water gained in stage 2, which was comparable to the outflow estimate of Lake Kusai ($0.22\,\text{km}^3$) based on Eq. (13).

altimetry water levels in certain lakes. Our study has considerably improved the temporal resolution of the monitoring of lake water level and storage changes in the TP. For most lakes examined in the published studies, to our best knowledge, the estimates from our study provide the observations of the highest temporal resolution that can better reveal the interannual and intra-annual variability and trends in lake water level and storage, even in some relatively small lakes whose annual trends may, however, be incorrectly estimated by sparse sampling of lake water levels. The developed data sets can also facilitate the monitoring of some rapidly expanding lakes with overflow risks and provide important information on flood prediction and early warning.

We evaluate the uncertainty in the Landsat-derived water levels by field experiments and rigorous uncertainty analysis. Both methods are consistent that the magnitude of the uncertainty is $\sim 0.1\,\text{m}$, which suggests that Landsat-derived water levels are often more efficient and less noisy than altimetry data when the altimeter footprints on the lake surface are insufficient, especially for small lakes. Based on our estimates, 52 large TP lakes accounting for $\sim 60\,\%$ of the total TP lake area have gained $98.3 \pm 2.1\,\text{km}^3$ of water during the past 18 years. Lakes in the endorheic basin on the TP have mostly expanded. The complex spatial pattern of lake storage changes in the Selin Co basin was quantified and a possible explanation was proposed in this study. Note that the quality of the Landsat-derived water levels before 2002 may not be as good as those after 2002, because no altimetry data before 2002 are used in this study. Extrapolation of the relationship between lake shoreline positions and water levels may not be stable if the water level during 2000–2001 was much lower or higher than those from 2002 to 2017. Discussions on how the extrapolation may affect the data quality can be found in the Supplement.

**Supplement.** The supplement related to this article is available online at: https://doi.org/10.5194/essd-11-1-2019-supplement.

**Author contributions.** LD and LX designed the research. LX, LD, HQ, and ZF developed the approaches and data sets. LX, HQ, HP, and LD carried out the field experiment. LX, LD, and YW contributed to the analysis of results and writing of the paper.

**Competing interests.** The authors declare that they have no conflict of interest.

**Acknowledgements.** Mingda Du's assistance in the field experiments, discussion on the use of satellite altimetry data for lake monitoring with Gang Qiao from Tongji University, and efforts in improving and archiving the data sets made by Daniela Ransby from PANGAEA data publisher are acknowledged here.

**Financial support.** This research has been supported by the National Natural Science Foundation of China (grant nos. 91547210 and 51722903) and the National Key Research and Development Program of China (grant no. 2017YFC0405801).

**Review statement.** This paper was edited by Birgit Heim and reviewed by two anonymous referees.

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
