# Peer review of "High temporal resolution water level and storage change data sets for lakes on the Tibetan Plateau during 2000–2017 using multiple altimetric missions and Landsat-derived lake shoreline positions"

_Earth System Science Data, 2019_

## Short Comment (SC1) · 8 Apr 2019

In this manuscript, the authors generated a dense (monthly and ever higher such as 10 days on average) continuous 18-year data set of changes in lake water level and storage for 52 large lakes on the Tibetan Plateau by combining multisource optical and altimetric information. Uncertainty in the optical water levels was evaluated by field experiments and rigorous uncertainty analysis, which is important to the generated data sets. The UAV imaging of lake shorelines for evaluating Landsat-based lake shoreline detection and the derivation of the mathematic expression of the uncertainty in the op-

tical water levels look really interesting and solid. The magnitude of the uncertainty was found to be around 0.1 m, suggesting that the optical water levels are often more efficient and less noisy than altimetry data when the altimeter footprints on the lake surface are insufficient, especially for small lakes.

I strongly believe that the data set is extremely valuable for the long-term and short-term monitoring of lake water level and storage changes on the Tibetan Plateau, and are also useful for lake water level and storage studies in other areas. Many studies on this aspect present long-term trends in these lake water storage. But the authors of this study have additionally explored the potential of these multiple remote sensing data sets in monitoring short-term variability in lake water storage and lake overflow floods that are really new and look fantastic to me.

Some suggestions are given as follows:

Pg. 6, Line 8: "the systematic biases between different altimetry data were removed by either comparing the mean water level of the overlap period or comparing the two water level time series with changes in lake shoreline, depending on the length of the overlap period" would be discussed in more detail.

Pg. 28, Lines 7‒8: "where optical water levels can provide a near real-time monitoring of changes in lake water level and storage that are crucial to flood early warning and risk management." However, I have not seen the results.
* * *

---

## Short Comment (SC2) · 23 Apr 2019

This is a valuable and interesting manuscript. The authors have exploited multisource remote sensing (i.e., multiple altimetric missions and Landsat archives) to create dense time series of lake water level and storage changes across 52 large lakes on the Tibetan Plateau. There are some previous studies focusing on changes in water level and storage on the Tibetan Plateau; however, these studies just got relatively lower temporal sampling and little altimetric information was used. It may limit the accuracy of trends in lake water level/storage in some cases and short-term monitoring of lake

overflow flood. Therefore, I am firmly convinced that the densified water-level dataset derived by the authors can have tremendous practical value in studying water storage changes and regional hydrological processes on the Tibetan Plateau.

There are some questions or suggestions for your consideration. 1. As far as I am concerned, deriving altimetry water levels through multiple altimetry missions (including Jason-1/2/3, ENVISAT, ICESat-1, and CryoSat-2) is the key component. I think the manuscript needs a more detailed description of this methodology in section 3.1. 2. To validate the derived optical water levels, the authors used pressure type water level sensors to measure water pressure and converted them into water depths. How to convert the water depths into the actual water level and unify to the same reference datum with optical water levels? It should be clarified. 3. Pg.1, Line 14 "(>100km2) " should be " (>150km2) "? 4. The legend of figure 11 should be revised (add unit and scale). 5. Figure 16, miss unit in y axis

---

## Referee Comment (RC1) · Anonymous Referee #1 · 7 May 2019

In this study, the authors developed a lake level dataset with dense samples for large lakes in 2000-2017 in the Tibetan Plateau (TP). The lake level product is validated by in situ water level measurements for Yamzhog Yumco. The water volume changes of 52 lakes with lake level were also estimated. This dataset is very valuable for studies of lake variations and their response to climate change in the TP and lake water balance. I recommend this manuscript to publish in ESSD, but some improvements based on comments below are necessary.

General comments: 1) The uncertainties for lake volume changes and other number

should be added through the manuscript.

2) What is optical water level? It is estimated by the correlation between lake area and level, and then to reconstruct the corresponding lake level using known lake area?

3) How all the lake level datasets are converted to same geoid?

4) In this study, lake boundaries were extracted using GEE. The visual checking and manual editing of delineated lake boundaries with original Landsat images are very necessary. How this was done at GEE platform?

5) How the in situ water level for Yamzhog Yumco is converted to consistent datum with satellite altimetry data? For validation of lake water classification with UAV, how about classification accuracy?

Specific comments:

1) Page1: "There are more than 1,200 alpine lakes larger than 1 km2 (Zhang et al., 2017a)" This result should come from Zhang, G. et al., 2014. Lakes' state and abundance across the Tibetan Plateau, Chinese Science Bulletin, 59(24):3010−3021. Please correct this cite here.

Page 2: ETM should be ETM+, not a superscript symbol of +, others are similar.

2) Page 4: "examine long-term", 2000-2017 is not long-term.

3) "The TP can be generally divided into 12 major basins. . .". Two suggested reference here:

Wan, W. et al., 2016. A lake data set for the Tibetan Plateau from the 1960s, 2005, and 2014, Scientific Data, 3:160039.

Zhang, G. et al., 2013. Increased mass over the Tibetan Plateau: From lakes or glaciers?, Geophysical Research Letters, 40(10):2125−2130.

4) Lake Selin Co-> Selin Co, others are similar

5) Page 5: "a lake shape data set generated by Wan et al. (2016) was used", This lake shape data was derived from GF data. How about the shift of lake outline? Did you check it with original Landsat images or Google Earth?

6) "we managed to make use of some images with gaps in generating lake shore changes." How to understand it?

7) "a half of them were excluded from the final results due to cloud contamination or gaps." How this is determined? Some lakes are missed? How to make sure a high-quality output of lake boundary, especially lake with little ice or turbid water?

8) Table 2: "d, m, km" can be put first row of table, then others below can be removed.

9) "either comparing the mean water level of the overlap period or comparing the two water level time series with changes in lake shoreline" How about the uncertainty and it is reasonable?

10) As the differences of extracted lake outlines, it is better to use a unique NDWI or MNDWI in classification of water and other land-cover in the study period? In addition, the differences from NDWI or MNDWI are not apparent?

11) "we selected images with less than 5% cloud cover" Some images with free-cloud coverage on lake shorelines are still useful?

12) Figure 11: background of this figure is not clear?

13) Figure 12: What is a high peak in Figure 12 in about 2010?

14) Figure 13: The trend of lake storage change is more robust than the result from Yao et al (2018) from Yao, F. et al., 2018. Lake storage variation on the endorheic Tibetan Plateau and its attribution to climate change since the new millennium, Environmental Research Letters:1-16. What is the cause for this difference?

15) Figure 15: How to understand the difference of lake level between these different datasets, especially polylines for optical water levelïij§

16) "5.3 Lake overflow flood monitoring" Many similar Chinese papers have been published. It is not need to include in the Title of this manuscript and put some in discussion is enough? In addition, some sentences such as equation can be moved into Method section?

17) Xiaojun et al., 2012 -> Yao et al., 2012

18) "Water loss was more likely to be found among the southern TP lakes. In the Selin Co basin, a more complicated spatial pattern of lake storage changes was detected, as small lakes were slowly losing water whereas the large lake was gaining water, which we speculated to be caused by lake-river interactions that need further investigation." These conclusions have found in previous studies. The summary here should more focus on the lake level data developed in this study.

19) Section 4 is too long? It can be shorten?

---

## Referee Comment (RC2) · Anonymous Referee #2 · 27 May 2019

This study combines altimetry data that measure lake levels directly with shoreline positions from optical data to create extended and denser lake level time series for the largest lakes of the TP. In that sense, the resulting dataset differs from existing lake level time series and seems thus a valuable resource for the scientific community as well as other users. The study is relevant for ESSD and worth publishing. To properly document the data and methods and to comply with ESSD's guidelines, the manuscript needs to be improved - in particular to better describe important parts of the methods, include/consider uncertainties, and properly validate the time series against existing

data sets.

**1 General comments**

1) The study would benefit from a clearer story line and justification how this work/data fills a current knowledge gap. I only understood the plot halfway through the methods. What are the shortcomings of the existing studies/datasets, and how do you overcome these with your study? This is especially important for the introduction, but also the abstract and conclusion would benefit from an easier to understand quick summary. See also comment paragraph P8, L11ff below.

2) Method: the important novelty of your approach is the use of shoreline positions from optical data to increase the temporal resolution and extend the length of the water level time series. To do so, you relate shoreline positions to lake level elevations from spaceborne altimetry data, using a statistical relationship between the two. Currently, the statistics part is not well enough described, and uncertainties from the found relationship do not seem to be propagated to your final "optical water levels". I suggest you extend this part to provide more transparency and include also a discussion of the uncertainties, considering in particular the assumption of a linear relationship (?) and whether it is appropriate to extrapolate beyond the range of measured lake levels.

3) Dataset: I'm missing a detailed description of the final dataset and its attributes and uncertainties, e.g. after the validation section.

4) Validation (and uncertainties):

- What is the accuracy/uncertainty of the altimetry products, and how does this propagate to your optical water levels? Consider also the uncertainty of the statistical relationship(s) you compute to derive the optical water levels.

- The theoretical computation of an uncertainty (most of 4.2) based on a single UAV image is not convincing to me as it is based on a single image pair only with unknown coregistration accuracy (see comment below). The lack of hands-on data basis and the extensive length of the theoretical part makes this off-topic. Maybe this could fit as supplementary information in a separate document.

- Rather than treating the comparison to the LEGOS Hydroweb data as an application case this should be part of the validation section. How do your time series compare to the other datasets listed in table 1?

For data description, uncertainties and validation see ESSD's guidlines at https://www.earth-syst-sci-data.net/10/2275/2018/, in particular sections 3.3, 3.5 and 3.6.

**2  Specific comments**

A simpler title might make it easier to understand what the study is about. Especially the rather unclear terms "densified" and "developed optical water levels" should be replaced. Focus on the data and not the application cases.

Abstract

The abstract could be more to the point. Add some information on the performance of your data (uncertainties and validation). Consider removing already published findings (applications).

L12: which altimetric missions? If there are too many to list all, specify how many and which types (e.g. lidar altimetry, interferometric SAR altimetry...)

L13: avoid putting important information in brackets. Monthly to weekly time series?

L16: "partial altimetry data" and "optical water levels" are unclear terms

L19: "densified" is unclear

L20ff: Are these groundbreaking new numbers/findings? Consider removing them and focus on the dataset.

Introduction

P2, L3: A strong motivation for TP lake studies not mentioned here is to find out why they are expanding, i.e. a good data set will contribute to a better understanding of climate and circulation patterns and changes thereof. This is important as the TP has a strong influence on both regional climate.

P2, L6: source of that number?

P2, L8: I wonder why you selected exactly these references? There are many more lake studies on the TP. References for the method (general) and local application should be separated.

P2, L11: It is better to introduce radar and lidar separately as the systems and data are quite different. Also, these data are not meant for ice berg height - you probably mean ice sheet surface elevation or sea ice freeboard?

P2, L16: The satellite is called ICESat, not ICESat-1. Change everywhere.

P2, L25: it seems you mainly mean (and in your study only use) optical data. Do you have an example for a sensor and study that used SAR data?

P2, L26: why exactly these references? These are not the only or first such studies.

P2, L33: references for the water index and Otsu algorithm?

P3, L10f: remaining bias: is this not true for your study, too? Or how do you avoid/remove such bias?

P3, table 1: Does this table include all studies, or how did you select? Either remove all that don't compute lake levels, otherwise consider also including "complete" TP water studies for a larger number of lakes than the ones you are listing (e.g. Pekel et al (2016) to whom you refer to earlier, or Yang et al. 2019, doi:10.5194/tc-2018-238; Treichler et al. 2018, doi:10.5194/tc-2018-238...)

P4, L4: the meaning of "hypsometric curve" is unclear to me in this context.

Study area and data

Parts of this (e.g. from P5, L24, or P6, L1ff) rather belongs to the method section.

P4, L16: "as opposed to many other places..." - I tend to disagree, as nearly all seem to have expanded. Can you justify or explain more clearly?

P5, L4ff: why did you choose these lakes in particular? And where is Lake Yamzhog Yumco? An overview map might be useful.

P5, L17: "moderate set of orbital parameters" is unclear

P5, L30: when were the drone data acquisitions?

P5, L31: "similar" in what sense? What have Huang et al done?

P6, table 2: Some of the missions included many instruments (e.g. ENVISAT: 10 sensors). You need to specify which sensor and data you used. Here, you distinguish between "radar" and "interferometer", which is also based on radar (SAR/interferometric radar altimeter). This is confusing, and it would be useful to explain the technologies/differences either in the introduction or in a separate paragraph in the data or methods section.

[Figure]

Methods

The first paragraph seems to explain what this study is about and would thus fit (better?) to the introduction (it is missing there!).

P6, L8: "comparing the mean water level of the overlap period" is vague. Explain better.

P7, figure 1: refer to the figure in the text, e.g. when you introduce the data and where you are talking about overlap periods. Consider adding the optical data to the figure to show the overlap periods you use to create the optical lake level - lake surface elevation relationship.

P7, L18: It is very unclear what "ENVISAT product" you used.

P7, L23: "highest bucket" is an unclear term. What elevation bin spacing did you choose for your frequency histograms? It seems you are losing information by binning your surface elevation measurements. How does that affect the accuracy of the extracted lake level elevations? I assume you have t-distributed data, i.e. roughly bell-shaped elevation distributions with long tails. It might be more appropriate to use the median elevation measurement, maybe in combination with a threshold to remove biased measurements in the tails. From reference DEMs, you should know the true surface elevation (of the lake shore).

P8, L4ff: How large are the biases you found? Are they constant over time and in space? I assume you compute this per lake?

P8, L13: it is unclear what you mean with "merging using optical water levels"

paragraph P8, L11ff: Only after reading this paragraph I think I finally understood the purpose of this study: You want to generate continuous lake level (volume?) series for as many lakes as possible. This requires elevation (and areal?) data from different sensors, as missions only last for a few years. As an additional challenge, the satellites in question have different orbits that only cover some lakes each, so not all elevation

datasets can be used for each lake. For each lake, you therefore combine lake level elevation time series from the different sensors with data for that lake, using the overlap periods to correctly align the records, i.e. you remove potential elevation bias between the time series and make sure they are consistent. Where there is no sufficient overlap, you use optical data as a proxy: you create a statistical relationship between lake levels (from altimetry data) and corresponding shoreline position (from optical data acquired at the same time), and then apply (extrapolate?) the relationship to (optical) shoreline positions for time periods where you lack surface elevation data, but do have optical data. I propose you add something like this to the introduction.

Secondly, this paragraph would be much easier to understand if you first introduce optical water levels and refer to figures 2 and 3 in the text. Given the importance of the relationship for your results you might want to explain your method in more detail. An important missing detail is whether you only interpolate or also extrapolate beyond the available data range?

P9, Figure 2: refer to the figure in the text, e.g. where you introduce the data sets and in section 3.1

P9, 3.2: The optical water levels should be introduced before P8, L15ff.

P10, L4ff: the part about "shifting gaps" and the ROI is unclear. Do you mean that the Landsat 7 gaps are not always exactly at the same place? Did you choose your ROIs such that they never contain no-data pixels? How did you ensure that, given the large amount of Landsat 7 data?

P10, L17: reference for the Otsu method?

P10, L22: How did you decide whether to use a linear or 2nd order polynomial fit?

P10, L25: How did you determine cloud cover?

P11, L2: How much data pairs did you end up with per lake, and how did you select pairs with regard to acquisition dates? I assume you did not always have altimetry and

shoreline data from the same date (?)

P11, figure 3: c) You might want to colour the dots according to time to check for (and show the readers that there is no) temporal bias. From d), it seems optical water levels are somewhat too high around 2004 and 2015, but too low around 2009?

P12, L9: How did you derive these ROIs? Are they drawn manually?

P12, L15: regression between the lake area and ..?

P13: As far as I am aware, Strahler's catchment hypsometric model is based on river catchments with a pour point, not endorheic lake catchments as it is the case for the TP. I am not entirely sure what you used this model for (to compute lake water volumes?), but I am not convinced that this is a correct approach. I am also not sure why you need that relationship at all? If you have lake area and lake level time series, you can directly compute volume changes from these?

P14, Figure 6: It is unclear what the parameters y, x, z, a and d represent.

P14, table 3: state nr. of data pairs (optical shoreline position + altimetric lake level) rather than optical data points

Validation

P16, L5: unclear sentence

P16, L25: the drone GPS tracker alone might not be very accurate, you may easily get a skewed/stretched image composite. Did you use ground control points?

P16, L27: This seems a rather dodgy way to determine the resolution of your image composite.

P17, figure7: which lake? images a) and e) should have the same size/spatial resolution. An overview map would be useful.

P17, L9ff: extrapolated or interpolated? Provide the parameters and statistical relationship here, maybe even in an additional figure.

P18, figure 8b: add 1:1 line and error bars for the data points

P18, 19: How did you coregistrate the UAV image composite and Landsat image? It seems a spatial shift will completely alter the (relative) shoreline position and thus the basis for your entire analysis: In Figure 9, shifting the shoreline only slightly in e.g. north-south direction will greatly change water/land (sub)pixel counts and thus the basis for the relationship in (b). In my opinion, an error analysis would require several image pairs (UAV and satellite-borne) and a solid coregistration basis, e.g. river/road crossings as clear tie points, or at least a round lake or elongated penninsula rather than a straight shore line.

P18, L7: what do you mean with "concurrent"? What dates?

P22, L1ff: don't forget the local conditions: ice, snow, wet, dry, muddy shore conditions or also waves greatly affect the water classification result.

Applications

P22, L10f: Are these your own numbers? How do they compare to previous estimates?

P22, L14ff: mark all lakes mentioned in the text in the map. If they are very close to each other, an extra zoom-in map might be useful.

P23f: restructure section 5 to avoid splitting the Selin Co basin analysis in two sections (5.1 and 5.3). How much of this is new, i.e. has not been published before? How does your dataset make a difference?

P23, L5ff: You mention only the study of Yao et al. (2018). How about other publications? Also, from figure 12 it seems quite clear that the Yao data contains two outliers. Consider using a robust fitting method rather than regular linear regression.

P24, L10: Depicting intra-annual variation is a strength of your dataset that you might want to emphasize more.

P24, 5.2: Rather than treating the comparison to the LEGOS Hydroweb data as an application case this should be part of the validation section!

P25, L8: "some kind of" bias removal: be more specific. The magnitude of the vertical shift between the two datasets fits to e.g. geoid/ellipsoid height confusion, but the temporal variability of the shift is worrying. Rather than speculating about the cause and assuming that the Hydroweb data is wrong you ought to find the reason for the differences - which may lie in your data processing/method.

P25, L13ff: "reverse relationship" and the conclusion you draw (Hydroweb may "underestimate decreasing trends"): unclear what you mean

P25, figure 15: What are the red/blue shaded areas? (a) compare the series after removing the shift. Sadly, the series (b) and (c) are nowhere discussed. The temporally varying offsets between the series from different data sources should be analysed and removed, or at least explained.

P26, figure 16: again, there seems to be some time-dependent offset between the optical and altimetry-based lake levels, e.g. optical levels are too high around 2005 in the top left panel, and too low around 2005 vs. too high from ca. 2013 in the middle right panel. Can you explain this?

P28, figure 17: What does the blue shaded area show? What data are you showing in these time series? Is the right panel a zoom-in of the left panel?

P28, L17: "Team, 2017": check author name

P29, figure 18: acquisition dates of the images in b) and c)?

P29ff: The entire overflow analysis (lots of new methods introduced) seems to be a study on its own and somewhat out of place in the applications ( results) section of this

paper.

Conclusions

A short summary of your methods should be provided, in particular the novelty of using shoreline positions from optical data to interpolate between available lake level measurements.

P31, L7: rephrase the sentence to avoid brackets.

P31, L10f: Unclear what you mean. From the comparison you provide currently, I am not yet convinced that your dataset is more correct than the Hydroweb data.

P31, L18: "rigorous uncertainty analysis": As mentioned above, I am not convinced about the theoretical uncertainty exercise you provide.

P31, L25ff: These insights about extrapolating using the derived statistical relationship are very important, but currently not quantified, mentioned or discussed anywhere else in the paper.

---

## Author Comment (AC3) · 20 Jun 2019

We really appreciate these overall comments and recommendation by this reviewer. Please find our point-by-point responses to the reviewer's comments and the modified manuscript based on this review's suggestions in the attached zip file.

Please also note the supplement to this comment:
https://www.earth-syst-sci-data-discuss.net/essd-2019-34/essd-2019-34-AC3-supplement.zip

---

## Author Comment (AC4) · 30 Jun 2019

We really appreciate the overall evaluation, insightful comments, and recommendation by this reviewer. Our point-by-point responses to the reviewer's comments and modified manuscript based on these comments can be found in the attached zip file.

Please also note the supplement to this comment:
https://www.earth-syst-sci-data-discuss.net/essd-2019-34/essd-2019-34-AC4-supplement.zip

---

## Author Response (AR1)

**Response to short comment 1**

General Comment:

In this manuscript, the authors generated a dense (monthly and ever higher such as 10 days on average) continuous 18-year data set of changes in lake water level and storage for 52 large lakes on the Tibetan Plateau by combining multisource optical and altimetric information. Uncertainty in the optical water levels was evaluated by field experiments and rigorous uncertainty analysis, which is important to the generated data sets. The UAV imaging of lake shorelines for evaluating Landsat-based lake shoreline detection and the derivation of the mathematic expression of the uncertainty in the optical water levels look really interesting and solid. The magnitude of the uncertainty was found to be around 0.1 m, suggesting that the optical water levels are often more efficient and less noisy than altimetry data when the altimeter footprints on the lake surface are insufficient, especially for small lakes.

I strongly believe that the data set is extremely valuable for the long-term and short-term monitoring of lake water level and storage changes on the Tibetan Plateau, and are also useful for lake water level and storage studies in other areas. Many studies on this aspect present long-term trends in these lake water storage. But the authors of this study have additionally explored the potential of these multiple remote sensing data sets in monitoring short-term variability in lake water storage and lake overflow floods that are really new and look fantastic to me.

Response:

We thank Dr. Xu for thoroughly reviewing the manuscript and making such encouraging comments. It is important for us to receive these feedbacks to further improve the data set and the manuscript. Comments and issues raised by Dr. Xu have been addressed and are illustrated as follows.

Specific Comment:

1) Pg. 6, Line 8: "the systematic biases between different altimetry data were removed by either comparing the mean water level of the overlap period or comparing the two water level time series with changes in lake shoreline, depending on the length of the overlap period" would be discussed in more detail.

Response:

We agree that the original description of the method in Pg.6, Line 8 is not quite clear and needs further clarification. The basic idea of removing the systematical bias is to calculate the mean of two altimetry-based water level time series from different sources during the overlap period. Then, the difference between the mean time series and either water level time series is removed to make both altimetry-based water level time series

consistent and form a longer water level time series. This process was subsequently applied to all water level time series with overlap periods to merge them into a single time series for each lake.

However, the overlap period could be short between some altimeters such as Envisat and CryoSat (e.g., there are only one or two data points in the overlap period), or does not exist at all, such as ICESat and CryoSat. On these cases, optical water levels (i.e., changes in lake shoreline that need to be translated into water levels using linear regression with one of the altimetry water level series) are used to extend or create an overlap period that links the two altimetry missions. We chose a one-year or two-year optical water level series which has an overlap period with both altimetry water level series as the baseline, and calculated the differences between altimetry water levels and the baseline during the overlap period. Then the differences from altimetry water levels were removed.

Therefore, three water level time series (i.e., one optical water level series and two altimetry water level series) from different sources are merged together. The reason why we used one-year or two-year optical water levels is because a longer overlap period may introduce some unexpected errors, such as a rapid increase in water level, which may, however, not be detected by optical water levels (only if the lakeshore slope happened to be steep in the exact region of increases in water level, which, for most cases, can be avoided by checking $R^2$ of linear regression when generating optical water levels).

This part will be modified in the revised manuscript.

Modifications: a separate paragraph summarizing the altimetry data merging process was added to the end of section 3.2.

2) Pg. 28, Lines 78: "where optical water levels can provide a near real-time monitoring of changes in lake water level and storage that are crucial to flood early warning and risk management." However, I have not seen the results.

Response:

Thanks for this comment. The expression here is indeed a perspective as to the potential and advantages of optical water levels rather than a strong statement. In this study, we provided water levels of Lake Salt, which has limited altimetry information but mainly consists of optical water levels. Though we used Landsat ETM$^+$ and OLI images, i.e., four observations were available in ~ one month, more than half of the images were useless due to cloud contamination or gaps. Therefore, the temporal resolution of optical water levels in Lake Salt is still ~ 1 month, which cannot be regarded as near real time monitoring at this stage. Nevertheless, the temporal resolution of optical water levels can be further improved by adding other missions such as Sentinel-2 that has a higher temporal resolution than Landsat series. A new data set termed harmonized Landsat and Sentinel-2 Reflectance Product has been generated recently, which we believe would improve the quality of optical water levels and make it near-real-time

observation.

**Response to short comment 2**

General Comment:

This is a valuable and interesting manuscript. The authors have exploited multisource remote sensing (i.e., multiple altimetric missions and Landsat archives) to create dense time series of lake water level and storage changes across 52 large lakes on the Tibetan Plateau. There are some previous studies focusing on changes in water level and storage on the Tibetan Plateau; however, these studies just got relatively lower temporal sampling and little altimetric information was used. It may limit the accuracy of trends in lake water level/storage in some cases and short-term monitoring of lake overflow flood. Therefore, I am firmly convinced that the densified water-level dataset derived by the authors can have tremendous practical value in studying water storage changes and regional hydrological processes on the Tibetan Plateau.

Response:

We thank Dr. Wu for these encouraging and constructive comments. As Dr. Wu's indicated, this work aims to provide improved lake water level and storage change estimates in terms of temporal resolution as well as accuracy. We appreciate all these comments from the community. Our responses to these comments are given as follows.

Specific Comment:

1) As far as I am concerned, deriving altimetry water levels through multiple altimetry missions (including Jason-1/2/3, ENVISAT, ICESat-1, and CryoSat-2) is the key component. I think the manuscript needs a more detailed description of this methodology in section 3.1.

Response:

Thanks for this comment. It is indeed important to clarify the method we used and developed to derive lake water levels from altimetry data. The waveform retracking methods in this section could be the most important part regarding technical details. Here we provided a general equation (Eq 1) for surface height calculation mainly because different sensors have different correction items, e.g., the saturation correction for ICESat (laser altimeter) was not applicable to radar altimeters.

As for waveform retracking correction, which is crucial to radar altimeters, we performed existing algorithms (e.g., the NTTP method for Croysat-2) or used a default method provided by the altimetry product (e.g., the ICE-1 retracking method). These methods have been widely tested and recommended based on in situ measurements. However, there can be a paradox when several studies suggested different methods for the same altimeter. If so, we can only apply the rule of thumb to choose those that

balance the robustness, computational cost, and accuracy (e.g., the Improved Threshold Method for Jason-1/2/3).

In fact, the original idea of the NTTP, ICE-1, and Improved Threshold Method is quite similar. All of them adopt a threshold defined as the percentage of waveform peak to determine the retracking gate, and then transfer the difference between the retracking gate and the nominal gate into range correction by multiplying the gate range (c$\Delta$t/2, where c is the speed of light and $\Delta$t is the time duration of a gate). The differences lie in the choice of thresholds as well as the calculation of waveform peaks. Therefore, we think it would be more suitable to provide some general information in the manuscript about threshold retracking schemes and to clarify the similarities and differences among the retracking methods we used.

This part will be modified in the revised manuscript.

Modifications: A separate paragraph was inserted following the 3$^{rd}$ paragraph of section 3.1 to briefly introduce the threshold waveform retracking scheme.

2) To validate the derived optical water levels, the authors used pressure type water level sensors to measure water pressure and converted them into water depths. How to convert the water depths into the actual water level and unify to the same reference datum with optical water levels? It should be clarified

Response:

Thanks for this comment. The water level measured by the pressure type sensor is the water depth (~20 m) of the installed location, while the water level acquired from optical images/satellite altimeters is the surface height with respect to EGM96 which generally has a value ~4000 m. To make them comparable, we calculated water level anomalies for the both time series. This part will be illustrated in detail in the revised manuscript.

Modifications: An explanation was inserted to the 1$^{st}$ paragraph of section 4.2.

3) Pg.1, Line 14 "(>100km2) " should be " (>150km2) "?

Response:

Thanks for this comment. Actually, we do have investigated almost all Tibetan lakes larger than 150 km$^2$ (except for one or two lakes with too limited altimetry/optical data) and several lakes between 100–150 km$^2$ (e.g., Lake Salt). This part will be modified in the revised manuscript to avoid confusion.

Modifications: An explanation was inserted to the Abstract to clarify the lake area.

4) The legend of Figure 11 should be revised (add unit and scale).

Response:

Thanks for this comment. It will be corrected in the revised manuscript.

Modifications: Unit and scale were added to Figure 11.

5) Figure 16, miss unit in y axis

Response:

Thanks for this comment. It will be corrected in the revised manuscript.

Modifications: Unit of y axis was added to Figure 16.

**Response to referee comment 1**

Comment:

In this study, the authors developed a lake level dataset with dense samples for large lakes in 2000–2017 in the Tibetan Plateau (TP). The lake level product is validated by in situ water level measurements for Yamzhog Yumco. The water volume changes of 52 lakes with lake level were also estimated. This dataset is very valuable for studies of lake variations and their response to climate change in the TP and lake water balance. I recommend this manuscript to publish in ESSD, but some improvements based on comments below are necessary.

Response:

We really appreciate these overall comments and recommendation by this reviewer. Our point-by-point responses to the reviewer's comments are given as follows.

General comments:

1) The uncertainties for lake volume changes and other number should be added through the manuscript.

Response:

Thanks for this comment. They will be added into the revised manuscript.

Modifications: Uncertainties were added for every lake volume/water level number, most of them appear in the Application section.

2) What is optical water level? It is estimated by the correlation between lake area and level, and then to reconstruct the corresponding lake level using known lake area?

Response:

Thanks for this comment. As illustrated in Page 10, line 20, the generation of optical water levels is similar to the description in this comment but is based on changes in lake shoreline observed in a smaller ROI (region of interest) rather than the whole lake area (e.g., the yellow square shown in Page 12, Figure 3 (b)). The reason for this is due to the increasing computational cost and probability of cloud/gap contamination with

increasing areas of ROI. On the other hand, it is pointed out in section 4.2 that if the ROI had a larger width (here 'width' is in the direction parallel to the shoreline), the uncertainty of optical water level would decrease. Therefore, the choice of the ROI is a trade-off between the accuracy and data availability or computational cost.

3) How all the lake level datasets are converted to same geoid?

Response:

Thanks for this comment. For altimetry water levels, the initial reference ellipsoid and geoid are different for different satellite missions/products. Information on the reference ellipsoid and geoid is listed in Supplement Table 1:

Supplement Table 1

| Altimetry mission | Reference Ellipsoid | Geoid |
|---|---|---|
| Envisat | WGS84 | EGM2008 |
| ICESat | T/P | EGM96 |
| CryoSat-2 | WGS84 | EGM96 |
| Jason-1/2/3 | T/P | EGM96 |

As mentioned in the manuscript, different altimetry water levels were merged by comparing the overlap period (more details are available in the response to short comment 1). Systematical biases caused by the geoid and reference ellipsoid were removed during this process.

For optical water levels, they were generated using linear regression with a certain source of altimetry water level data so they have the same reference ellipsoid and geoid with the respective altimetry data used in the regression. And they can be merged with other altimetry data by comparing the overlap period as well.

However, there is a correction that must be made to the manuscript in Page 8, Line 18. "…all water levels were with respect to EGM96…" was incorrect. For the 12 lakes with Jason data, all kinds of water levels were converted into T/P, EGM96, because the Jason-1/2/3 data were used as the baseline (i.e., the longest records will be used as the baseline). For the rest of the lakes we mainly used Envisat data as the baseline to merge all the water levels. Therefore, for lakes without Jason data but having Envisat data, the water levels were converted into WGS84, EGM2008 (see Supplementary Table 1 above). For lakes without either Jason or Envisat data, Cryosat-2 data were used as the baseline, so water levels for these lakes were converted to WGS84, EGM96. We will provide a supplement document to mark out the Reference Ellipsoid and Geoid for each lake.

Modifications: "…all water levels were with respect to EGM96…" was removed from the manuscript; a table containing reference ellipsoid and geoid for each lake was provided in the supplementary file.

4) In this study, lake boundaries were extracted using GEE. The visual checking and

manual editing of delineated lake boundaries with original Landsat images are very necessary. How this was done at GEE platform?

Response:

Thanks for this comment. Lake areas or lake boundaries were used in two situations in this study: the first is during the process of generating hypsometric curves and the second is during the selection of altimeter footprints. We used lake areas derived from GEE in the first situation but used an existing dataset based on manual delineation produced by Wan et al. 2016 in the second situation. Therefore, problems raised in this comment may only exist in the first situation.

Visual checking can be hard to perform on GEE due to a large number of images we used, but we did visually check and preclude some of the images that resulted in outliers in the extracted lake surface areas (e.g., the entire ROI was covered by snow resulting in the failure of the Otsu method). It is true that manual editing of lake boundaries is important if we only use a small number of images (e.g., less than 10 images) to derive hypsometric curves (which is common in similar studies, e.g., most hypsometric curves provided by Hydroweb used less than 10 data pairs).

Nevertheless, for most lakes (42 out of 52) in our study, we used more than 20 data pairs (i.e., lake water area and corresponding lake water level) to fit hypsometric curves. More data pairs we used make the hypsometric curves more robust, even though there may be some misclassification in a single image. This is evidenced by the fact that most $R^2$ values for the hypsometric curves are higher than 0.9. In addition, all the images used in the regression analysis met the criterion of cloud contamination less than 5%, which has largely reduced uncertainty in the extracted lake water area.

5) How the in situ water level for Yamzhog Yumco is converted to consistent reference ellipsoid with satellite altimetry data? For validation of lake water classification with UAV, how about classification accuracy?

Response:

Thanks for this comment. As illustrated in our response to short comment 2, the in situ water levels of Yamzhog Yumco were made comparable with the satellite altimetry/optical water levels by calculating the anomalies of each water level time series.

Lake water classification with UAV images was performed by manually identifying the lake shoreline using ArcGIS. Therefore, the uncertainty in the UAV derived lake shoreline is considerably small, because the spatial resolution of the UAV image is ~5 cm.

Specific Comments:

1) Page1: "There are more than 1,200 alpine lakes larger than 1 km$^2$ (Zhang et al., 2017a)" This result should come from Zhang, G. et al., 2014. Lakes' state and

abundance across the Tibetan Plateau, Chinese Science Bulletin, 59(24):3010−3021. Please correct this cite here.

Page 2: ETM should be ETM+, not a superscript symbol of +, others are similar.

Response:

Thanks for this comment. They will be modified in the revised manuscript.

Modifications: The reference (Zhang, G. et al., 2014) was added in the 1st paragraph of introduction and "ETM+" was change into "ETM+" everywhere.

2) Page 4: "examine long-term", 2000–2017 is not long-term.

Response:

Thanks for this comment. "Long-term" will be changed into "multiyear".

Modifications: "long-term" was changed into "multiyear" everywhere.

3) "The TP can be generally divided into 12 major basins... ". Two suggested reference here:

Wan, W. et al., 2016. A lake data set for the Tibetan Plateau from the 1960s, 2005, and 2014, Scientific Data, 3:160039.

Zhang, G. et al., 2013. Increased mass over the Tibetan Plateau: From lakes or glaciers?, Geophysical Research Letters, 40(10):2125−2130.

Response:

Thanks for this comment. They will be added into the manuscript.

Modifications: The two references (Wan, W. et al., 2016, Zhang, G. et al., 2013) were added in the 1st paragraph of section 2.1.

4) Lake Selin Co-> Selin Co, others are similar

Response:

Thanks for this comment. They will be modified in the revised paper.

Modifications: "Lake Selin Co" was changed into "Selin Co"; "Lake Nam Co" was changed into "Nam Co": "Lake Zhari Namco" was changed into "Zhari Namco"; "Lake Goren Co" was changed into "Goren Co"; "Lake Urru Co" was changed into "Urru Co"; "Lake Yamzhog Yumco" was changed into "Yamzhog Yumco".

5) Page 5: "a lake shape data set generated by Wan et al. (2016) was used". This lake shape data was derived from GF data. How about the shift of lake outline? Did you check it with original Landsat images or Google Earth?

Response:

Thanks for this comment. Yes, we did notice that lake outlines derived from GF data

that were only used for generating the 2014 subset of the data set by Wan et al. (2016)) have a shift relative to those derived from Landsat ETM+ and CBERS-1 that were used for the 2005 subset of the data set). Therefore, we only used the 2005 subset of the data set to select altimetry footprints in our study.

6) "We managed to make use of some images with gaps in generating lake shore changes." How to understand it?

Response:

Thanks for this comment. As shown in Figure 3 (b), the ROI we used to derive lake shoreline changes is small enough to be fitted into an ETM+ image strip (valid pixels) between two gaps (no-value pixels). In this way, we can make use of some images with gaps. However, the gaps are shifting so we set a criterion (no-value pixels in the ROI should be less than 2%) to remove those images for which the ROI is contaminated by shifting gaps.

7) "A half of them were excluded from the final results due to cloud contamination or gaps." How this is determined? Some lakes are missed? How to make sure a high-quality output of lake boundary, especially lake with little ice or turbid water?

Response:

Thanks for this comment. As illustrated in the manuscript, we have excluded images which have more than 5% cloud pixels or 2% no-value pixels in the selected ROI. "A half" is an approximation for the portion of effective images when generating optical water levels. To further show this, we randomly chose five lakes to present the portion of effective images:

Supplement Table 2

| Lake name | TM | ETM+ | OLI | Total |
|---|---|---|---|---|
| Jingyu | 90/287 | 225/360 | 58/128 | 373/775 |
| Zhari Namco | 127/308 | 47/229 | 178/371 | 352/908 |
| Mapam Yumco | 50/57 | 216/274 | 85/121 | 351/452 |
| Lumajiangdong Co | 74/183 | 78/561 | 68/250 | 220/994 |
| Aqqikkol Lake | 169/387 | 108/557 | 109/246 | 386/1190 |
| Total | 510/1222 | 674/1981 | 498/1116 | 1682/4319 |

Effective ETM+ images have a lower portion due to gaps. But for Landsat TM and OLI images, the portion is appropriately 1/2. However, the portion of effective images that can be used to derive lake water areas, instead of optical water levels mentioned earlier, is much smaller, sometimes less than 10%, which is due to the increasing probability of cloud/gap contamination with increasing areas of ROI.

We did consider the impact of lake ice or snow on the accuracy of lake area/lake shoreline extraction. Both MNDWI and NDWI were not able to well discriminate lake ice and water as what we had expected (i.e., if lake ice was eliminated, the extracted

lake area would be smaller than its real size). We noticed that the MNDWI cannot completely discriminate snow and water either, resulting in artificial increases in lake areas in winter when the lake bank was covered by snow. Therefore, the NDWI was used to better discriminate snow from water/floating lake ice in winter. However, if there was snow cover on the lake ice, the NDWI could also produce artifacts in the derived lake area and we had to remove these outliers manually as we mentioned earlier.

As for the turbid water problem which mainly occurred in summer, we examined the study lakes and found that both MNDWI and NDWI can precisely locate the lake boundary, even though the near-shore water color was affected by turbid inflow as shown in Supplement Figure 1:

[Figure]

[Figure]

[Figure]

Landsat OLI 2017/10/07     MNDWI water classification     NDWI water classification

Supplement Figure 1: Water classification results at the estuary of Lake Kusai based on the MNDWI and NDWI during the flood season.

The difference between the MNDWI and NDWI is that the MNDWI is able to detect shallower turbid water than the NDWI (e.g., shallow rivers can be detected by MNDWI in Supplement Figure 1), which is important in determining the accurate position of lake shorelines. If the NDWI was used in summer, less information on changes in lake shoreline (i.e., optical water level) would be detected. On the other hand, rivers and other small water bodies near the lake can lead to noise to the extracted lake area due to the sensitivity of the MNDWI. Therefore, we carefully chose the ROI to avoid rivers or small water bodies. A comparison between the MNDWI and NDWI was performed by (Huang et al., 2018) based on UAV images, which also shows that the MNDWI has better performance than the NDWI under the condition without snow cover.

8)   Table 2: "d, m, km" can be put first row of table, then others below can be removed.

Response:

Thanks for this comment. Yes, it will be modified in the revised manuscript.

Modifications: Units of Table 2 were moved to the first row.

9) "Either comparing the mean water level of the overlap period or comparing the two water level time series with changes in lake shoreline" How about the uncertainty and it is reasonable?

Response:

Thanks for this comment. The uncertainty of this method is important, because errors induced by this data merging method will evolve into the merged water levels and become remaining systematical biases. Such biases will cause artificial rises or falls of the merged lake water levels, jeopardizing the consistency of the merged lake water levels between different time periods and sensors. Therefore, the consistency of the merged water levels can reflect the remaining systematical biases and the uncertainty of the data merging method that caused these biases.

However, it is a dilemma that evaluating the consistency of the merged water levels is difficult to perform without continuous in situ observations over multiple years. As far as we know, there are few continuous measurements of lake water levels in the Tibetan Plateau due to the equipment failure in the frozen season (e.g., caused by fierce winds, waves, and freezing process). For instance, several water level sensors have been set up in Nam Co since 2005 (Song et al., 2015), but the in situ water level measurements of Nam Co presented in the literature were discontinuous in the frozen season.

Therefore, the best available reference we used to assess the consistency of the multiple altimetry water levels when there is no overlap period is the optical water level. Optical water levels are generally continuous in our study period and could even be more reliable than intermittent ground observations. Given the fact that continuous ground observations do not exist, are not accurate enough, or are not accessible if any, the altimetry data merging method proposed in this study is a reasonable and effective way to generate longer and denser time series on lake water levels.

10) As the differences of extracted lake outlines, it is better to use a unique NDWI or MNDWI in classification of water and other land-cover in the study period? In addition, the differences from NDWI or MNDWI are not apparent?

Response:

Thanks for this comment. Based on response to Comment 7, it is clear that either NDWI or MNDWI has pros and cons and may perform quite differently. Therefore, a combination of the two water indices is a reasonable solution and has been used in this study.

11) "We selected images with less than 5% cloud cover". Some images with free-cloud coverage on lake shorelines are still useful?

Response:

Thanks for this comment. Yes, they are. Moreover, the cloud mask algorithm imbedded

in the Landsat QA band is quite sensitive. Sometimes an image with light cloud in ROI slightly higher than 5% is still useful, because water indices are not largely affected. A 20% threshold was used by (Huang et al., 2018), which also produced satisfactory results.

12) Figure 11: background of this figure is not clear?

Response:

Thanks for this comment. Yes, the background has been changed into green now.

13) Figure 12: What is a high peak in Figure 12 in about 2010?

Response:

Thanks for this comment. It may have been caused by an outlier that was not removed prior to uploading the generated data set. In the uploaded data set, such a peak does not exist. It will be corrected in the revised paper.

Modifications: Figure 12 (now Figure 13) was revised with outliers removed.

14) Figure 13: The trend of lake storage change is more robust than the result from Yao et al (2018) from Yao, F. et al., 2018. Lake storage variation on the endorheic Tibetan Plateau and its attribution to climate change since the new millennium, Environmental Research Letters:1-16. What is the cause for this difference?

Response:

Thanks for this comment. As illustrated in Line 5–7 in the manuscript Page 23, our data (a combination of the merged optical water levels and altimetry water levels) have higher sampling frequency than (Yao et al., 2018a), resulting in a more robust estimation of the trend in the lake water levels. As shown in Figure 12, there are several abrupt changes with magnitudes up to ~3 km$^3$ in lake storage observed by (Yao et al., 2018a), which is not likely to happen, given that there is no report on basin flood/upstream lake overflow. This could be due to the uncertainty in the lake area they derived and applied to estimate changes in lake storage. We also noticed that lake areas derived from Landsat archives could be much noisier than lake shoreline changes, due to cloud contamination/temporary small water bodies within the ROI. Therefore, we have calculated changes in lake storage with water levels and hypsometric curves, instead of directly using water levels and lake areas to reduce the uncertainty in derived lake areas.

15) Figure 15: How to understand the difference of lake level between these different datasets, especially polylines for optical water level?

Response:

Thanks for this comment.

First, the difference between altimetry water levels in our data set and the Hydroweb

data set mainly comes from following processes: (1) different reference ellipsoids and geoid models, (2) different retracking methods, and (3) different schemes of removing systematical bias. The last process is the most significant difference that could make our data set more consistent compared with the optical water levels as we explained in response to specific comment 9.

Second, the difference between the optical water levels and altimetry water levels mainly comes from different mechanisms of observations. Altimetry water levels are based on the time delay between the generated and received signal measured by altimeters. Each cycle corresponds to one water level value averaged from several footprints across the lake. Therefore, the number of footprints in a cycle is crucial to the accuracy of altimetry water levels. Footprints falling on a study lake are determined by the orbit of the satellite altimetry and the size of the lake, both of which are fixed.

On the other hand, optical water levels are derived from optical images, which could be affected by cloud cover. Therefore, there is not a fixed temporal resolution for optical water levels. As illustrated in section 4, optical water levels are mainly affected by the slope of lake shore, the width of ROI, the spatial resolution of the optical image, and the accuracy of the water classification method. Some of these factors, such as the width of ROI, spatial resolution, and slope can be well handled. Therefore, optical water levels are less noisy than altimetry water levels.

16) "5.3 Lake overflow flood monitoring". Many similar Chinese papers have been published. It is not need to include in the Title of this manuscript and put some in discussion is enough? In addition, some sentences such as equation can be moved into Method section?

Response:

We have revised the title of this manuscript according to this comment. Content associated with lake overflow flood monitoring is no longer reflected in the title, but has been put in the method and discussion sections. These modifications will be shown in the revised paper after considering all reviewers' suggestions.

Modifications: The title has been changed with "overflow" removed and part of the section has been moved into the supplementary file.

17) Xiaojun et al., 2012 -> Yao et al., 2012

Response:

Thanks for this comment. It will be modified in the revised manuscript.

18) "Water loss was more likely to be found among the southern TP lakes. In the Selin Co basin, a more complicated spatial pattern of lake storage changes was detected, as small lakes were slowly losing water whereas the large lake was gaining water, which we speculated to be caused by lake-river interactions that need further investigation." These conclusions have found in previous studies. The summary

here should more focus on the lake level data developed in this study.

Response:

Thanks for this comment. It will be modified in the revised manuscript.

Modifications: This part was revomed from the 2$^{nd}$ paragraph of conclusion.

19) Section 4 is too long? It can be shortened?

Response:

Thanks for this comment, we plan to move part of the content in section 4 into the supporting information.

Modifications: Part of section 4 was removed into the supplementary file.

**Response to referee comment 2**

Comment:

This study combines altimetry data that measure lake levels directly with shoreline positions from optical data to create extended and denser lake level time series for the largest lakes of the TP. In that sense, the resulting dataset differs from existing lake level time series and seems thus a valuable resource for the scientific community as well as other users. The study is relevant for ESSD and worth publishing. To properly document the data and methods and to comply with ESSD's guidelines, the manuscript needs to be improved - in particular to better describe important parts of the methods, include/consider uncertainties, and properly validate the time series against existing data sets.

Response:

We really appreciate the overall evaluation, insightful comments, and recommendation by this reviewer. Our point-by-point responses to the reviewer's comments are given as follows.

General comments:

1) The study would benefit from a clearer story line and justification how this work/data fills a current knowledge gap. I only understood the plot halfway through the methods. What are the shortcomings of the existing studies/datasets, and how do you overcome these with your study? This is especially important for the introduction, but also the abstract and conclusion would benefit from an easier to understand quick summary. See also comment paragraph P8, L11ff below.

Response:

Thanks for this constructive comment. As suggested by this Reviewer in specific

comments, we have reorganized several paragraphs and enhanced how our study and developed data set fill a current knowledge gap in the introduction, abstract and conclusion sections. Abstract and conclusion sections have also been improved by reducing all redundant information. Details can be found in the attached modified manuscript.

2) Method: the important novelty of your approach is the use of shoreline positions from optical data to increase the temporal resolution and extend the length of the water level time series. To do so, you relate shoreline positions to lake level elevations from spaceborne altimetry data, using a statistical relationship between the two. Currently, the statistics part is not well enough described, and uncertainties from the found relationship do not seem to be propagated to your final "optical water levels". I suggest you extend this part to provide more transparency and include also a discussion of the uncertainties, considering in particular the assumption of a linear relationship (?) and whether it is appropriate to extrapolate beyond the range of measured lake levels.

Response:

Thanks for this very insightful comment. As suggested by this Reviewer, we have extended section 3.2 (optical water level) and provided a discussion in section 4.2 to better evaluate the uncertainty in the regression relationship and how it propagates into optical water levels. The extrapolation problem is discussed in section 4.2 as an interpretation of the propagated regression uncertainty and in this response letter too (specific comment 8 of the method section). We believe that the impact of extrapolation of optical water levels possibly occurring in the time gap between two altimetry time series has been well addressed in this response letter (specific comment 8 of the method section) and will be added to the supplementary file. However, we acknowledge that little information is available to quantify the effect of extrapolation during the time window from 2000–2002, as little altimetry information is available due to either poor quality or limited observations, and available DEM is too coarse to describe the micro topography of the lake bank. We have informed potential readers/users of such a risk in the validation and conclusion sections of the revised manuscript.

3) Dataset: I'm missing a detailed description of the final dataset and its attributes and uncertainties, e.g. after the validation section.

Response:

Thanks for this comment. The description of the dataset is combined with the data availability following the validation part now.

4) Validation (and uncertainties):

a) What is the accuracy/uncertainty of the altimetry products, and how does this propagate to your optical water levels? Consider also the uncertainty of the statistical relationship (s) you compute to derive the optical water levels.

Response:

Thanks for this insightful comment. We used the standard deviation of water levels from valid footprints in a cycle to represent the uncertainty in the altimetry product. The valid footprints are referred to as the footprints selected with the histogram method as illustrated in the manuscript. For most cases they comprise more than 80% of all available footprints in a cycle. As suggested by this Reviewer, a thorough discussion of the error propagation from the altimetry data to the optical water level through the statistical relationship has been added in section 4.2.

b) The theoretical computation of an uncertainty (most of 4.2) based on a single UAV image is not convincing to me as it is based on a single image pair only with unknown coregistration accuracy (see comment below). The lack of hands-on data basis and the extensive length of the theoretical part makes this off-topic. Maybe this could fit as supplementary information in a separate document.

Response:

Thanks for this insightful comment. We have redone the uncertainty analysis based on high-resolution optical images from GF-2 (i.e., China's high spatial resolution satellite) and investigated a total of 4128 Landsat shoreline pixels after performing co-registration (the co-registration error was estimated to be ~2 m). Based on the new experiments and results, we have modified part of section 4.2, making it more convinced. Considering the excessive content of section 4, we will move part of the theoretical derivation to a supplementary file as suggested by this Reviewer.

c) Rather than treating the comparison to the LEGOS Hydroweb data as an application case this should be part of the validation section. How do your time series compare to the other datasets listed in table 1?

For data description, uncertainties and validation see ESSD's guidelines at https://www.earth-syst-sci-data.net/10/2275/2018/, in particular sections 3.3, 3.5 and 3.6.

Response:

Thanks for this comment. We have moved part of the comparison with Hydroweb data to the validation section. We chose to make a comparison with the Hydroweb because Hydroweb data have exploited most altimetry missions and provided densest altimetry water levels among all listed studies (also for most lakes the systematic biases between altimetry missions seem to have been well removed), very typical for altimetry-based lake studies. Other altimetry-based lake studies may include more lakes, but based on the published results they are subject to some systematic biases. Therefore, we have taken the Hydroweb data as the benchmark to see if there are improvements or advantages in our generated product.

We did compare our lake data with that of Yao et al. (2018b) and show the importance

of temporal resolution, as we are not comparable with the lake quantity of these kind of studies based only on Landsat images and DEM. Studies that primarily use Landsat images and DEM are able to cover a larger number of lakes and are not subject to systematic biases as those using various altimetry data sources. However, most of those studies have a low temporal resolution (e.g., annually or even lower) due to the difficulty of acquiring quality optical images covering entire lake areas at a high temporal resolution, as opposed to our study that needs optical images covering a small portion of the lake shore.

Specific Comments:

A simpler title might make it easier to understand what the study is about. Especially the rather unclear terms "densified" and "developed optical water levels" should be replaced. Focus on the data and not the application cases.

Response:

Thanks for this constructive comment. The tentative title of this study has been revised as: "Generation of high temporal resolution water level and storage change data sets for lakes on the Tibetan Plateau during 2000–2017 using multiple altimetric missions and Landsat-derived lake shoreline positions and areas" for your kind suggestion.

Abstract

1) The abstract could be more to the point. Add some information on the performance of your data (uncertainties and validation). Consider removing already published findings (applications).

Response:

Thanks for this constructive comment. We have removed numerical results similar to some published work such as lake storage trends and lake overflow amount. More information on the validation and uncertainty has been added, as we performed additional experiments with high-spatial-resolution images. Details can be found in the revised manuscript.

Modifications: Numbers of lake volume changes were removed from the abstract; validation of the optical water level was emphasized.

2) L12: which altimetric missions? If there are too many to list all, specify how many and which types (e.g. Lidar altimetry, interferometric SAR altimetry...)

Response:

Thanks for this comment. All altimetric sensors used in this study have been listed in Table 2. For the sake of brevity, we decide not to list them in the abstract.

3) L13: avoid putting important information in brackets. Monthly to weekly time

series? L16: "partial altimetry data" and "optical water levels" are unclear terms

Response:

Thanks for this constructive comment. This sentience has been modified. Brackets in L13 have been removed and a brief explanation to optical water levels has been added.

4) L19: "densified" is unclear

Response:

Thanks for this comment. It has been replaced with "merged".

5) L20ff: Are these groundbreaking new numbers/findings? Consider removing them and focus on the dataset.

Response:

Thanks for raising this comment. These numbers are actually not that different from published studies, but they can serve as an independent source of information from relevant studies, as we have generated a new dataset with temporal resolution being greatly improved and systematic biases being well removed. We have removed these numbers and placed more emphasis on the dataset itself.

Introduction

1) P2, L3: A strong motivation for TP lake studies not mentioned here is to find out why they are expanding, i.e. a good data set will contribute to a better understanding of climate and circulation patterns and changes thereof. This is important as the TP has a strong influence on both regional climate.

Response:

Thank you so much for this comment. We have added this to the first paragraph of introduction to clearly state the motivation of TP lake studies that a good data set should contribute to a better understanding of climate and circulation patterns and changes.

2) P2, L6: source of that number?

Response:

The source is (Messager et al., 2016).

3) P2, L8: I wonder why you selected exactly these references? There are many more lake studies on the TP. References for the method (general) and local application should be separated.

Response:

Thanks for this comment. We agree that more general studies instead of local applications may be cited. Now we have cited the earliest one that we can find to represent this kind of studies using remotely sensed water surface height and extent

performed by Frappart et al. (2005).

Modifications: References here were changed into (Frappart et al. 2005)

4) P2, L11: It is better to introduce radar and lidar separately as the systems and data are quite different. Also, these data are not meant for ice berg height - you probably mean ice sheet surface elevation or sea ice freeboard?

Response:

Yes, this makes sense. We have separately introduced laser and radar altimeters and added a supplementary description of the two types of altimeters to underscore the differences between them in this paragraph. We agree that the altimetry data are not meant for ice berg height. It has now been corrected in the revised manuscript.

Modifications: "ice berg" was changed into "ice sheet/ice freeboard".

5) P2, L16: The satellite is called ICESat, not ICESat-1. Change everywhere.

Response:

Done.

6) P2, L25: it seems you mainly mean (and in your study only use) optical data. Do you have an example for a sensor and study that used SAR data?

Response:

Yes, SAR images from Sentinel-1 were used by Huang et al. (2018) from our group to derive the effective river width, which is calculated with the river surface area divided by the river length. The automatic extraction of the river surface area is similar to that of the lake surface area or lake shoreline changes. We may take advantage of SAR data in future studies.

7) P2, L26: why exactly these references? These are not the only or first such studies.

Response:

It is true there are many published studies on water classification/extraction. We chose these two references mainly because they are similar in study area, data source, and publishing time, showing a good comparison between methods. We would like to show a change in this kind of studies and to stress the point that manual extraction of lake boundary could be labor-intensive and low-efficiency.

8) P2, L33: references for the water index and Otsu algorithm?

Response:

Done.

9) P3, L10f: remaining bias: is this not true for your study, too? Or how do you avoid/remove such bias?

Response:

We have done our best to remove the systematic bias between different altimetry missions by using optical water levels as reference data, which is rarely seen in the literature. Hwang et al. (2019) showed that the systematic bias among different altimeters is hard to remove unless in situ water level measurements or Jason-1/2/3 data are available. Our method could provide a better solution to this problem. We would not say there is no remaining systematic bias in our data, but we are confident that the biases have largely been reduced. Even though there might be some concern about the accuracy of the optical water levels because altimetry information is involved in the generation, they are currently the best available long-term reference data for ungauged lakes.

10) P3, table1: Does this table include all studies, or how did you select? Either remove all that do not compute lake levels, otherwise consider also including "complete" TP water studies for a larger number of lakes than the ones you are listing (e.g. Pekel et al (2016) to whom you refer to earlier, or Yang et al. 2019, doi:10.5194/tc-2018-238; Treichler et al. 2018, doi:10.5194/tc-2018-238...)

Response:

Thanks for this constructive comment. We consider it is quite reasonable to exclude those references without water levels, as our study focuses on improving the quality of merged water levels and subsequently improving lake storage change estimation.

Modifications: (Wan et al., 2016) and (Yang., 2017) were removed from Table 1.

11) P4, L4: the meaning of "hypsometric curve" is unclear to me in this context.

Response:

We noticed that in some studies hypsometric curves represent the total area above a certain elevation, which means that at the lowest elevation the hypsometric curve reaches its maximum value. However, in this study, hypsometric curves represent the lake surface area at a given water level, which means that the curve reaches its maximum value when the water level is maximized. We adopted this denotation as same as the LEGOS Hydroweb. To make it clear, we have added an explanation in brackets in the context.

Modifications: An explanation in brackets was added following "hypsometric curve".

Study area and data

1)  Parts of this (e.g. from P5, L24, or P6, L1ff) rather belongs to the method section.

Response:

Thanks for this comment. We have moved partial content to the method section. For instance, we have moved P5 L24–L26 to the second paragraph of section 3.2.

2) P4, L16: "as opposed to many other places..." - I tend to disagree, as nearly all seem to have expanded. Can you justify or explain more clearly?

Response:

We only studied 12 lakes outside the endorheic basin for the recent twenty years, which possibly caused such an impression that all lakes have experienced expansion. Exorheic lake shrinkage in the TP in the past 50 years can be seen from (Zhang et al., 2019) as shown in the figure below.

[Figure]

In addition, most global endorheic basins have experienced water loss in recent years, whereas the endorheic region in the TP has gained water (Wang et al., 2018). This phenomenon has also drawn a lot of attention for the endorheic basin in the TP.

3) P5, L4ff: why did you choose these lakes in particular? And where is Lake Yamzhog Yumco? An overview map might be useful.

Response:

The reasons why we chose Yamzhog Yumco and Nam Co are threefold: (1) they are close to the city, making it easier for logistics and transportation; (2) they are both large lakes, typical in our study; and (3) one of them is located in the endorheic basin (Nam Co), and the other is from the exorheic basin (Yamzhog Yumco), increasing the representativeness of the experiment.

Following figure will be added into the manuscript to clearly show the two experiment locations:

[Figure]

4) P5, L17: "moderate set of orbital parameters" is unclear

Response:

Thanks for this comment. We have made it clear. We meant to show that Envisat has a lower orbit than Jason-1/2/3 but higher than ICESat, thereby for sampling frequency: ICESat<Envisat<Jason-1/2/3, and for spatial coverage: ICESat>Envisat>Jason-1/2/3.

Modifications: "moderate set of orbital parameters" was removed, replaced by "...a lower orbit than Jason-1/2/3 but higher than ICESat..."

5) P5, L30: when were the drone data acquisitions?

Response:

The drone images were acquired in the morning on May 19 and 21, 2018, for Yamzhog Yumco and Nam Co respectively. The Landsat images used for validation purposes were both acquired on May 19, 2018.

6) P5, L31: "similar" in what sense? What have Huang et al done?

Response:

Huang et al. (2018) used UAV images to evaluate the performance of water auto-extraction with four water indices based on Landsat 8 images. The accurate water surface boundary was extracted manually from the UAV images using ArcGIS, and then

water extraction results from Landsat using different water indices were compared with the accurate water surface area from the UAV images. Our data source and method are similar, but focused on different targets. On the other hand, we have performed a systematic analysis to link the uncertainty in water surface area extraction to the uncertainty in optical water levels.

7) P6, table 2: Some of the missions included many instruments (e.g. ENVISAT: 10 sensors). You need to specify which sensor and data you used. Here, you distinguish between "radar" and "interferometer", which is also based on radar (SAR/interferometric radar altimeter). This is confusing, and it would be useful to explain the technologies/differences either in the introduction or in a separate paragraph in the data or methods section

Response:

Thanks for this constructive comment. It is important to clarify the sensors and data we used, and they have been added to the table now. The classification of different radar altimeters in the original manuscript might be confusing as indicated by the reviewer. Therefore, we have provided a brief explanation after the first paragraph of section 2.2 on the mechanism of different altimeters including SIRAL onboard CryoSat-2.

Modifications: Information of sensor name and type was added into Table 2, as well as data record name; A separate paragraph was added following the 1$^{st}$ paragraph of section 2.2 to clarify the difference between different altimeters.

Methods

1) The first paragraph seems to explain what this study is about and would thus fit (better?) to the introduction (it is missing there!).

Response:

Thanks for this comment. They have been moved to the introduction section.

2) P6, L8: "comparing the mean water level of the overlap period" is vague. Explain better.

Response:

Thanks for raising this issue. It has been explained in detail in our response to short comment 1 (the first question in short comment 1). We have also added a separate paragraph at the end of section 3.2 to better explain this part.

3) P7, figure 1: refer to the figure in the text, e.g. when you introduce the data and where you are talking about overlap periods. Consider adding the optical data to the figure to show the overlap periods you use to create the optical lake level-lake surface elevation relationship.

Response:

Yes, we have added references to Figure 1 in three places where we think it is necessary. In addition, the optical data are presented in Figure 1. However, it is not easy to show the time period we used to derive optical water levels from altimetry data, because for different altimetry missions may be used to derive optical water levels for different lakes. For instance, if Jason-1/2/3 data are available, optical water levels are generated by fitting with the merged Jason-1/2/3 water levels. If ICESat and CryoSat-2 data are available for a lake, optical water levels are generated first by fitting with CryoSat-2 data. After the extended CryoSat-2 data are merged with the ICESat data, the optical water levels generated throughout the entire study period are checked again by fitting with the merged altimetry water levels to see if there is an extrapolation problem. We will discuss this issue in detail in response to the specific comment 8 below in this section.

Modifications: Reference to Figure 1 was added in several places; More information about the second regression was added.

4) P7, L18: It is very unclear what "ENVISAT product" you used.

Response:

Thanks for this comment. It has been changed to Envisat/RA-2.

5) P7, L23: "highest bucket" is an unclear term. What elevation bin spacing did you choose for your frequency histograms? It seems you are losing information by binning your surface elevation measurements. How does that affect the accuracy of the extracted lake level elevations? I assume you have t-distributed data, i.e. roughly bell-shaped elevation distributions with long tails. It might be more appropriate to use the median elevation measurement, maybe in combination with a threshold to remove biased measurements in the tails. From reference DEMs, you should know the true surface elevation (of the lake shore).

Response:

Thanks for this insightful comment. We used a 0.6-meter bin space to generate a histogram and the 'highest bucket' represents the histogram bin with the highest frequency. It has now been clarified in the revised manuscript. We do not think much information is lost, as for most cycles (>70%) there are more than 80% measurements falling into the highest bin. We first used the median value of each cycle to represent the lake water level, which is noisier/less smoother than that using the histogram. It turns out that a 0.6 m bin space is large enough to capture valid measurements in a cycle.

It is true that a bell distribution is quite common for most lakes. But setting constant thresholds to remove outliers for each lake does not seem to work well in our study. We did try this method before but it always ends up in how to choose an appropriate threshold. If the threshold is too large, invalid measurements will be involved in a lake. Otherwise, certain amount of information would be lost. For instance, Lake Kusai

experienced a water level jump up to ~10 m in 2011. If we do not know this information before, then a threshold must be larger than ± 10 m from the mean water level/DEM to capture the water level jump, which will definitely introduce a number of inaccurate measurements in normal cycles.

6) P8, L4ff: How large are the biases you found? Are they constant over time and in space? I assume you compute this per lake?

Response:

Thanks for this comment. The spatial distribution of systematic biases seems quite random to us, varying from place to place, even the sign of the systematic biases is not stable between two certain altimeters (except for Jason-1/2/3). The range of biases is within ± 5 m. Fortunately, the systematic bias is quite stable in time, as we compared the merged altimetry data with the optical water levels. If the bias is not stable in time/elevation, which means that the additive correction is not effective enough, the multiplicative correction may be needed. Overall, we did not see the necessity of using the multiplicative correction nor did we find any relative research reporting such corrections.

7) P8, L13: it is unclear what you mean with "merging using optical water levels"

Response:

It should be clear now as we have provided a separate paragraph at the end of section 3.1 to summarize the merging process. Thanks for this comment.

8) paragraph P8, L11ff: Only after reading this paragraph I think I finally understood the purpose of this study: You want to generate continuous lake level (volume?) series for as many lakes as possible. This requires elevation (and areal?) data from different sensors, as missions only last for a few years. As an additional challenge, the satellites in question have different orbits that only cover some lakes each, so not all elevation datasets can be used for each lake. For each lake, you therefore combine lake level elevation time series from the different sensors with data for that lake, using the overlap periods to correctly align the records, i.e. you remove potential elevation bias between the time series and make sure they are consistent. Where there is no sufficient overlap, you use optical data as a proxy: you create a statistical relationship between lake levels (from altimetry data) and corresponding shoreline position (from optical data acquired at the same time), and then apply (extrapolate?) the relationship to (optical) shoreline positions for time periods where you lack surface elevation data, but do have optical data. I propose you add something like this to the introduction. Secondly, this paragraph would be much easier to understand if you first introduce optical water levels and refer to Figures 2 and 3 in the text. Given the importance of the relationship for your results you might want to explain your method in more detail. An important missing detail is whether you only interpolate or also extrapolate beyond the available data range?

Response:

We really appreciate these accurate and comprehensive summary and highlights on our work. As the referee suggested earlier, we have enhanced the introduction section to clarify the purpose and underscore the contributions of this study. We have added references to Figure 2 and Figure 3 in this paragraph and we have moved part of it to the end of the optical water levels section (section 3.2). The interpolation and extrapolation may be the most concerned issue here. Below we provide a few examples to justify our methodology.

Note that we have performed two regressions to generate the optical water levels. For the first regression, we only used one altimetry data product and optical images-derived lake shoreline positions. After merging the altimetry water levels, we performed the second regression using the merged altimetry water levels and the optical water levels temporally close to the altimetry water levels throughout the entire study period. This information is missing in the original manuscript and we will add it in the revised manuscript/supplementary file. Here we show that part of the extrapolation problem is evitable in nature with the second regression:

a) When and where does extrapolation exist?

First, extrapolation here means the extrapolation of the linear relationship developed from the regression analysis between altimetry water levels and lake shoreline changes. For instance, if the altimetry water levels used for the regression analysis have a range of 4500–4502 m, then the generated optical water levels beyond/below this range are regarded as extrapolated values. On the other hand, if an optical water level $H_1$ acquired in 2003 is within 4500–4502 m, though the altimetry water levels used for such a regression were from 2010 to 2017, $H_1$ is still regarded as an interpolated value because it is within the elevation range of the linear regression.

As shown in following figures (both are conceptualized examples, optical water levels are fitted with the second altimetry product), when seasonal signal is dominated in the time series, there is no need for extrapolation. The red line in the optical water levels (which serves as the merging reference to altimetry data 1) are within the range of the linear regression. The merging between the two altimetry water levels can subsequently be achieved by removing the difference (symmetrical bias) of the mean water levels between altimetry data 1 and altimetry data 2 during the reference period (the red solid line) from altimetry data 1 (typically ICESat data).

[Figure]

When a multiyear trend is dominated in the time series, the merging reference is out of the range of the regression relationship, and then extrapolation may occur. Both situations are common in our study. The first situation comprises 60% of all study lakes, and extrapolation can take place in ~40% lakes. The two altimetry datasets in the extrapolation case can still be merged using the similar procedure and optical water levels shown in the interpolation case above.

[Figure]

b) How does extrapolation become a problem?

In the merging process, extrapolation becomes a problem only if the lake bank slope experiences an abrupt change at the exact elevation where both altimetry products fail to cover, as illustrated in the following figure:

[Figure]

Such a situation may happen, but the possibility is relatively low. If it happens, the extrapolation will result in a remaining systematic bias in the merged altimetry water levels and consequently jeopardizing the accuracy of the optical water levels.

c) How can the problem be avoided?

By performing the regression analysis twice, it is possible to detect if there is an abrupt change in lake bank slope. If the situation in b) does happen, we can easily see from the scatterplot of the second regression analysis that the linear assumption is no longer met (i.e., the scatterplot would show two slopes/curvature). Once an obvious failure in the second linear regression occurs, we will re-choose the region of interest (ROI) and go through the entire process of generating optical water levels again. However, it only happened twice or three times in our study.

We will provide the details of generating the optical water levels discussed above in the supplementary file as they may be too detailed for general readers.

9) P9, Figure 2: refer to the figure in the text, e.g. where you introduce the data sets and in section 3.1

Response:

Yes, we have added reference to Figure 2 (now Figure 3, because we inserted a new figure after Figure 1) in the first paragraph of section 2.2 and fourth paragraph of section 3.1.

10) P9, 3.2: The optical water levels should be introduced before P8, L15ff.

Response:

The sequence has been changed.

11) P10, L4ff: the part about "shifting gaps" and the ROI is unclear. Do you mean that the Landsat 7 gaps are not always exactly at the same place? Did you choose your ROIs such that they never contain no-data pixels? How did you ensure that, given the large amount of Landsat 7 data?

Response:

Yes, the position of gaps in Landsat 7 data is various with time. But they are more like vibrating around a fixed location. So, narrowing down the width of ROI can assure higher data availability. It is true that filtering a large amount of Landsat 7 archives is really tough, but our study was primarily based on GEE and we performed an invalid-pixel detection to get rid of images with missing pixels in the ROI. The algorithm is straightforward: comparing the valid pixel number in the ROI with that from an intact image. If the missing pixels in the ROI exceed 2% then the image will be excluded. Using 2% instead of 0% is due to the consideration of the algorithm robustness, but there is not much difference in the results as the ratio of in-valid pixels is either very high (>20%) or extremely close to zero.

12) P10, L17: reference for the Otsu method?

Response:

It has been added.

13) P10, L22: How did you decide whether to use a linear or 2nd order polynomial fit?

Response:

Thanks for raising this comment. In fact, it only happened in two lakes: Zhari Namco and Chibzhang Co, where we already have Jason-1/2/3 data for altimetry data merging. For other lakes we only performed linear regression, and if the scatterplot of the regression has a clear curvature, we will re-choose the ROI (see our response to comment 8 in the method section). For Zhari Namco and Chibzhang Co, if we use linear regression, a clear discrepancy will show up at either low water levels or high water levels. Therefore, using a higher order regression is a choice.

14) P10, L25: How did you determine cloud cover?

Response:

The cloud cover was calculated in GEE based on the quality band of Landsat 5/7/8. Pixels in the quality band categorized as cloud or cloud shadow will be masked with a mask function provided in GEE. Then, the cloud/cloud shadow pixels will be regarded as invalid pixels and a corresponding rate can be calculated by dividing invalid pixels with the total pixels in the ROI. If the cloud rate is higher than 5%, the image will be discarded.

15) P11, L2: How much data pairs did you end up with per lake, and how did you select

pairs with regard to acquisition dates? I assume you did not always have altimetry and shoreline data from the same date (?)

Response:

We have an average of 55 data pairs for the second regression of optical water levels. About 70% of the study lakes have more than 20 data pairs. The time difference of data pairs is within 5 days. However, if there are not enough data pairs (<10 pairs) with a time difference smaller than 5 days, we will increase the time difference to 10 days. Only for a lake named Xuru Co, where altimetry information is very limited, we increased the time difference to 30 days.

16) P11, figure 3: c) You might want to colour the dots according to time to check for (and show the readers that there is no) temporal bias. From d), it seems optical water levels are somewhat too high around 2004 and 2015, but too low around 2009?

Response:

Yes, this is a nice suggestion. Based on the following figure, it seems that in 2009 the optical water levels might be a little lower than expected. It may be caused by the uncertainty in altimetry water levels. In 2009 the main data source is Envisat, which has poorer quality than other altimetry products (except Jason-1) in our study. Overall, the impact of this problem is quite limited as the linear relationship is still strong.

[Figure]

Modifications: Figure 4 (c) was replaced.

17) P12, L9: How did you derive these ROIs? Are they drawn manually?

Response:

Yes, they are drawn manually. Selection criterion is illustrated in the manuscript. However, it still requires some experience.

18) P12, L15: regression between the lake area and ..?

Response:

It is between lake area and merged lake water levels, including altimetry water levels and optical water levels, but most data pairs are lake areas and optical water levels, because they usually come from the same Landsat image.

19) P13: As far as I am aware, Strahler's catchment hypsometric model is based on river catchments with a pour point, not endorheic lake catchments as it is the case for the TP. I am not entirely sure what you used this model for (to compute lake water volumes?), but I am not convinced that this is a correct approach. I am also not sure why you need that relationship at all? If you have lake area and lake level time series, you can directly compute volume changes from these?

Response:

Thanks for raising this comment. We intended to provide some justification that a parabolic relationship between the lake area and lake water level is reasonable. But it seems that such a justification is unnecessary and inappropriate because the assumption of exorheic basins is not met. We will remove this analysis from the revised manuscript.

The reason why we use the lake area-water level relationship to calculate the volume change is the lack of lake water areas with a sufficient temporal resolution. In general, we only have ~20 lake area observations for each lake, because the ROI for lake area extraction is much larger than that of lake shoreline changes, reducing the data availability. If we use the volume formula for computation, we can only get ~20 volume change values. With a lake area-water level relationship, we can derive the lake volume-water level relationship and convert all water level estimates into lake volume changes.

Modifications: The Strahler's model was removed from the manuscript.

20) P14, Figure 6: It is unclear what the parameters y, x, z, a and d represent.

Response:

Thanks for this comment. We will remove this part from the revised manuscript.

21) P14, table 3: state nr. of data pairs (optical shoreline position + altimetric lake level) rather than optical data points

Response:

Yes, they have been added.

Validation

1) P16, L5: unclear sentence

Response:

The sentence has been reorganized.

Modifications: Part of the 1st paragraph of Section 4.1 was reorganized to show that the focus is the uncertainty of optical water levels.

2) P16, L25: the drone GPS tracker alone might not be very accurate, you may easily get a skewed/stretched image composite. Did you use ground control points?

Response:

We did not get ground control points. It is true that there may be skewing or stretching distortions in UAV images. So, we redid the experiment with some commercial high-resolution data such as GF-2 (China's High Resolution Satellite, GF-2, with a panchromatic resolution of 0.8 m), which has larger coverage and more ground features for co-registration with Landsat OLI image.

Modifications: The UAV image-based validation was removed and replaced by GF-2 image-based validation, the later has undergone co-registration.

3) P16, L27: This seems a rather dodgy way to determine the resolution of your image composite.

Response:

Yes, it is not very rigorous, and we have abandoned it.

4) P17, figure7: which lake? images a) and e) should have the same size/spatial resolution. An overview map would be useful.

Response:

Thanks for this comment. We performed UAV scanning and water level sensor installation in both lakes. However, the water level sensor in Nam Co was broken down soon after installation and did not provide much information. Figure 7 shows pictures acquired at the Nam Co experiment spot. An overview map has been added into the study area section (section 2.1), as the referee suggested before. We decided not to use the UAV image as a validation basis, but we keep it here to show the environment at the experimental spot. In addition, the up-left image from Landsat 8 has been changed into an overview map of Nam Co and the experiment location.

Modifications: An overview map was added to Figure 7.

5) P17, L9ff: extrapolated or interpolated? Provide the parameters and statistical relationship here, maybe even in an additional figure.

Response:

The optical water levels of Yamzhog Yumco used for validation are interpolated. The statistics of regression are already shown in Figure 3.

6) P18, figure 8b: add 1:1 line and error bars for the data points

Response:

Yes, they have been added as shown in the figure below.

[Figure]

7) P18, 19: How did you coregistrate the UAV image composite and Landsat image? It seems a spatial shift will completely alter the (relative) shoreline position and thus the basis for your entire analysis: In Figure 9, shifting the shoreline only slightly in e.g. north-south direction will greatly change water/land (sub) pixel counts and thus the basis for the relationship in (b). In my opinion, an error analysis would require several image pairs (UAV and satellite-borne) and a solid coregistration basis, e.g. river/road crossings as clear tie points, or at least a round lake or elongated peninsula rather than a straight shore line.

Response:

This is a very constructive comment. We agree that there might be a spatial shift in the UAV image. Therefore, we no longer use the UAV image because there are very few ground features for image co-registration. Instead, we purchased some high-resolution commercial images obtained by GF-2 (0.8 m resolution at the panchromatic band) to repeat the analysis. The GF-2 images cover a much larger area and more diverse ground features, making it easier for image co-registration. The following figure shows control points that we selected for one of the GF-2 images. The co-registration error is 1.2 GF-2 pixels, say ~1 m. The other two GF-2 images have a co-registration error of 2.45 pixels and 2.72 pixels, respectively, corresponding to ~2 m.

[Figure]

Modifications: Co-registration Information of GF-2 images and Landsat images was added to the study data section and validation section.

8) P18, L7: what do you mean with "concurrent"? What dates?

Response:

It means the "same period" image. We have changed this expression.

9) P22, L1ff: Do not forget the local conditions: ice, snow, wet, dry, muddy shore conditions or also waves greatly affect the water classification result.

Response:

Thanks for this comment. Yes, the local condition is an important factor affecting the water area classification accuracy. Therefore, we chose three high resolution images acquired in different seasons and different places representing typical local conditions around the TP lakes, covering turgid water (wet season), lake ice, and dry season. As for vegetation, most of the TP lakes do not have much vegetation on the lake bank, with the Landsat images unable to detect information on vegetation.

Applications

1) P22, L10f: Are these your own numbers? How do they compare to previous estimates?

Response:

Yes, they are results generated from our product. There has not been any published study that has exactly the same study period or lakes as what we did. But for the overlap periods and lakes, the results are similar. We have made many comparisons with published studies or open source data, including the comparison between our product and Hydroweb data in Figure 14 in the original manuscript (now Figure 11).

2) P22, L14ff: mark all lakes mentioned in the text in the map. If they are very close

to each other, an extra zoom-in map might be useful.

Response:

Yes, this has been done, as shown in the figure below:

[Figure]

3) P23f: restructure section 5 to avoid splitting the Selin Co basin analysis in two sections (5.1 and 5.3). How much of this is new, i.e. has not been published before? How does your dataset make a difference?

Response:

Thanks for this comment. We have only talked about Lake Kusai in section 5.3. All discussion about Selin Co is shown in section 5.1. There are some published studies that report the unusual spatial pattern of lake area/water level/storage changes in the Selin Co basin. However, there is no discussion about the reason. We proposed a possible explanation. On the other hand, given the complexity of modeling a multi-lake endorheic basin (Zhou et al., 2015), our product does provide a chance for investigating the structure of such a endorheic basin with complicated lake-river systems. For instance, the height of outlet of three upstream lakes in the Selin Co basin may be inferred from the dense time series from our product with the help of a hydrologic/hydrodynamic model.

4) P23, L5ff: You mention only the study of Yao et al. (2018). How about other publications? Also, from figure 12 it seems quite clear that the Yao data contains two outliers. Consider using a robust fitting method rather than regular linear regression.

Response:

Thanks for this comment. Song et al. (2013) notice the decreasing trend of the three lakes in the upstream of Selin Co during 2003 to 2009 when ICESat data are available,

but there are no comments or discussion about the reason. We found that Hydroweb data do not catch the decreasing trend of Urru Co after 2000. Jiang et al. (2017) did not investigate the decreasing trend of Urru Co from 2003 to 2015 as their altimetry data from ICESat and CryoSat were not linked together but separately discussed instead. Hwang et al. (2019) reported a similar problem as Jiang et al. (2017). Other studies do not present specific statistics for the comparison nor do they cover those lakes.

With a robust linear fit method (Theil-Sen estimator), the result from Yao et al. (2018b) did show a decreasing trend, consistent with our result. But they clearly did not use a robust fitting in their published paper/dataset.

[Figure]

5) P24, L10: Depicting intra-annual variation is a strength of your dataset that you might want to emphasize more.

Response:

Yes, we did describe the intra-annual variation in the lakes we studied.

6) P24, 5.2: Rather than treating the comparison to the LEGOS Hydroweb data as an application case this should be part of the validation section!

Response:

Yes, we have moved part of section 5.2 to the validation section (section 4.3).

7) P25, L8: "some kind of" bias removal: be more specific. The magnitude of the vertical shift between the two datasets fits to e.g. geoid/ellipsoid height confusion, but the temporal variability of the shift is worrying. Rather than speculating about the cause and assuming that the Hydroweb data is wrong you ought to find the reason for the differences - which may lie in your data processing/method.

Response:

Thanks for raising this insightful comment. The reason for the vertical shift between our product and Hydroweb data possibly lies in different geoids/reference ellipsoids, as

illustrated in our response to referee comment 1 (General comment 3). However, we respectfully disagree on the point of the temporal variability in the vertical shift.

In the manuscript, we just indicate that partial Hydroweb data are not quite consistent with the optical water levels (e.g., in the three lakes shown in Figure 15), which are able to provide a straightforward answer to "in which period the lake has higher water level". As we have clarified in the revised manuscript, such a relationship on relative magnitudes reflected by the optical water levels does not change with the linear fitting parameters (unless using a negative slope, which is impossible) and that is why we regard it as robust. What we did was merging different altimetry data sources based on the reference provided by the optical water levels. Therefore, it is not likely to be a problem for this straightforward and robust scheme for merging altimetry data.

8) P25, L13ff: "reverse relationship" and the conclusion you draw (Hydroweb may "underestimate decreasing trends"): unclear what you mean

Response:

Thanks for raising this comment. We apologize for making a wrong expression in the original manuscript: the conclusion should be "…there is a possibility that Hydroweb data overestimate the increasing trend of water levels in Taro Co from 2003 to 2015".

As shown in Figure 15 (a), the last two measurements from ICESat should equal or be even larger than the first two/three measurements from CryoSat/Saral based on optical water levels, but the Hydroweb data show a reverse relationship that the last two ICESat measurements is 0.3~0.4 m smaller than the first two CryoSat/Saral measurements. This phenomenon suggests that ICESat water levels of Taro Co from Hydroweb is 0.3~0.4 m lower than the expected (in other words, CryoSat/Saral time series from Hydroweb is 0.3~0.4 m higher than the expected). It would therefore result in an overestimation of increases in lake water levels in Taro Co during the time window. In addition, the optical water levels in Taro Co were interpolated with the developed statistical relationship. Therefore, the discrepancy between Hydroweb and our product is not attributed to the extrapolation of the optical water levels.

9) P25, figure 15: What are the red/blue shaded areas? (a) compare the series after removing the shift. Sadly, the series (b) and (c) are no where discussed. The temporally varying offsets between the series from different data sources should be analysed and removed, or at least explained.

Response:

Thanks for this comment. The red and blue areas were meant for highlighting/comparing the periods when an obvious discrepancy between Hydroweb data and optical water levels from our product occurs. As for Figure 15 (a), we have removed the systematic vertical offset between our dataset and Hydroweb data of Taro Co, which is shown in the figure below:

[Figure]

As we suggest earlier in the response letter, there might be a remaining systematic bias between ICESat and CryoSat/Saral data from Hydroweb. Based on optical water levels, the peak water level of 2009 shall be higher than that of 2010 (again, such a relationship does not change regardless of the uncertainty in the linear fitting parameters during the generation of optical water levels), which means that the last two ICESat measurements are supposed to be higher or equal the first a few CryoSat/Saral measurements. However, this is not seen in the Hydroweb data for this specific lake.

As for Figure 15 (b) and (c), they show other examples of possible remaining systematic biases in Hydroweb data. The explanation is exactly the same as that of Figure 15 (a) and we did explain the discrepancy in the manuscript for Figure 15 (a). Thanks for your kind attention to this.

10) P26, figure 16: again, there seems to be some time-dependent offset between the optical and altimetry-based lake levels, e.g. optical levels are too high around 2005 in the top left panel, and too low around 2005 vs. too high from ca. 2013 in the middle right panel. Can you explain this?

Response:

Thanks for this insightful comment. Though there seems to be an offset at around 2005, the actual deviation between the optical and altimetry water levels here (ICESat data) is about 0.2~0.3 m, which is within the uncertainty range of altimetry measurements for inland water bodies. Instead of a time-dependent offset, we think it is more like a random error, which can be caused by the loss of valid altimeter footprints of that cycle, e.g., a random shift of ground tracks resulting in a smaller cross section and fewer footprints on the lake. It is also suggested that optical water levels may be more robust and less noisy than altimetry data. This is the same for the middle right panel. It should be noted that in the middle right panel, Envisat/RA-2 was used, which has a larger uncertainty than ICESat. Therefore, it is not surprising that altimetry dots seem to be more randomly distributed.

11) P28, figure 17: What does the blue shaded area show? What data are you showing in these time series? Is the right panel a zoom-in of the left panel?

Response:

The blue shade shows the period when an outburst happens. The data we show in Figure 17 are lake water storage changes for relevant lakes during the outburst event. Their locations are shown in Figure 18 (b) and (c). And yes, the right panel is a magnified plot of the blue shade in the left panel.

12) P28, L17: "Team, 2017": check author name

Response:

Yes, we have checked the reference. It has been cited dozens of times in other journal papers according to the Google Scholar.

[CITATION] **Planet Application Program Interface**: In Space for Life on Earth
P Team - San Francisco, CA, 2017
☆ 〝 Cited by 69  Related articles

13) P29, figure 18: acquisition dates of the images in b) and c)?

Response:

Figure 18 (b) was acquired in December, 2010. Figure 18 (c) was acquired in December, 2013. The outburst took place at the end of 2011. These are images from the Google Earth (i.e., the image source is Landsat but experienced merging processes, e.g., merging of images acquired from the same month) and we do not know the exact acquisition date.

Modifications: The acquisition time was added in the annotation of Figure 18.

14) P29ff: The entire overflow analysis (lots of new methods introduced) seems to be a study on its own and somewhat out of place in the applications (results) section of this paper.

Response:

Thanks for this comment. We have shortened this section and moved some of the analyses into the supplementary file. But we would like to keep this part, because some information (e.g., height and width of the outlet) of the overflow lake, Lake Kusai, is critical to downstream residents and emergency administrations, given that there are reports showing high overflow/outburst risks of Lake Salt in the near future.

Modifications: Part of the overflow modeling was moved to the supplementary file.

Conclusions

1) A short summary of your methods should be provided, in particular the novelty of using shoreline positions from optical data to interpolate between available lake level measurements.

Response:

Yes, this has been added.

2) P31, L7: rephrase the sentence to avoid brackets.

Response:

Done.

3) P31, L10f: Unclear what you mean. From the comparison you provide currently, I am not yet convinced that your dataset is more correct than the Hydroweb data.

Response:

Thank you for this comment. We have put more detailed explanations (most of them are already discussed in this response letter) in the second paragraph of section 5.2 and hopefully this would convince the reviewers and readers. Based on the overall comparison shown in previous Figure 14 (now Figure 11), our product is generally consistent with Hydroweb data, and has a higher temporal resolution.

But there are indeed some discrepancies between the two products over some lakes during some time windows as what we illustrated earlier. Hydroweb is a decent global dataset whereas our dataset is more a regionally based product. It is not uncommon in the remote sensing community that a regionally based dataset may have some advantages than a global dataset in some aspects due to the improvement of the algorithm for the data generation and use of more detailed (a priori) information derived from optical images to densify the spaceborne altimetry water levels with systematic errors being well removed. The developed method we present has potential to improve lake water level and storage changes in different regions globally at large.

4) P31, L18: "rigorous uncertainty analysis": As mentioned above, I am not convinced about the theoretical uncertainty exercise you provide.

Response:

Thanks for this comment. We have redone the uncertainty analysis with more high-resolution images and corresponding Landsat images. We have also provided co-registration accuracy and considered different seasons and locations as the reviewer suggested. This part should now be convincing the reviewer and general readers.

5) P31, L25ff: These insights about extrapolating using the derived statistical relationship are very important, but currently not quantified, mentioned or discussed anywhere else in the paper.

Response:

Thanks for this constructive comment. We will put the discussion of extrapolation we made in this response letter (specific comment 8 in the method section) into the supplementary file. Clearly, our discussion mainly focuses on the period during

which various altimetry data sources are merged, but does not include the period before 2002 when little altimetry information is available and DEM is too course (for instance, SRTM DEM has a 1 m vertical resolution with more than 10 m vertical uncertainty according to Mukherjee et al. (2013)) to provide a detailed description on the lake shore micro topography. Therefore, we do not have much information and materials to discuss about the extrapolation before 2002.

We have informed readers in the manuscript that this is a possible issue but it may only exist in the first 2–3 years of the dataset for lakes with strong signal from multiyear trends as opposed to seasonal variations. After all, compared with the 18-year study period, the impact of extrapolation of the optical water levels during 2000–2002 would be quite limited.

**A list of modifications in the manuscript**

Modification position is referred to the marked-up manuscript.

| Comment | Modification | Modification position |
|---|---|---|
| Modifications based on SC1 | | |
| SC1 General comment | None | None |
| SC1 Specific (1) | A separate paragraph summarizing the altimetry data merging process was added to the end of section 3.2. | P20, L4-L14 |
| SC1 Specific (2) | None | None |
| Modifications based on SC2 | | |
| SC2 General comment | None | None |
| SC2 Specific (1) | A separate paragraph was inserted following the 3$^{rd}$ paragraph of section 3.1 to briefly introduce the threshold waveform retracking scheme | P11, L21-L25 |
| SC2 Specific (2) | An explanation was inserted to the 1st paragraph of section 4.2. | P26, L7-L8 |
| SC2 Specific (3) | An explanation was inserted to the abstract to clarify the lake area. | P1, L18 |
| SC2 Specific (4) | Unit and scale were added to Figure 12 | P36, L4 |
| SC2 Specific (5) | Unit of y axis was added to Figure 16 | P42, L7 |
| Modifications based on RC1 | | |
| RC1 General (1) | Uncertainties were added for every lake volume/water level number, most of them appear in the Application section | everywhere |
| RC1 General (2) | None | None |
| RC1 General (3) | "…all water levels were with respect to EGM96…" was removed from the manuscript; a table containing reference ellipsoid and geoid for each lake was provided in the supplementary file | P12, L21; Supplementary file part 1 |
| RC1 General (4) | None | None |
| RC1 General (5) | Same as SC2 Specific (2) | Same as SC2 Specific (2) |
| | | |
| RC1 Specific (1) | The reference (Zhang, G. et al., 2014) was added in the 1$^{st}$ paragraph of introduction and "ETM$^{+}$" was change into "ETM+" everywhere | P2, L7 everywhere |
| RC1 Specific (2) | "long-term" was changed into | everywhere |

| | "multiyear" everywhere | |
|---|---|---|
| RC1 Specific (3) | The two references (Wan, W. et al., 2016, Zhang, G. et al., 2013) were added in the 1st paragraph of section 2.1. | P6, L7 |
| RC1 Specific (4) | "Lake Selin Co" was changed into "Selin Co"; "Lake Nam Co" was changed into "Nam Co": "Lake Zhari Namco" was changed into "Zhari Namco"; "Lake Goren Co" was changed into "Goren Co"; "Lake Urru Co" was changed into "Urru Co"; "Lake Yamzhog Yumco" was changed into "Yamzhog Yumco" | everywhere |
| RC1 Specific (5) | None | None |
| RC1 Specific (6) | None | None |
| RC1 Specific (7) | None | None |
| RC1 Specific (8) | Units of Table 2 were moved to the first row | P9, L26 |
| RC1 Specific (9) | None | None |
| RC1 Specific (10) | None | None |
| RC1 Specific (11) | None | None |
| RC1 Specific (12) | the background of Figure 12 has been changed into green | P36, L4 |
| RC1 Specific (13) | Figure 13 was revised with outliers removed | P37, L5 |
| RC1 Specific (14) | None | None |
| RC1 Specific (15) | None | None |
| RC1 Specific (16) | The title has been changed with "overflow" removed and part of this section has been moved into the supplementary file | P1, L1-L7 Supplementary file Part 4 |
| RC1 Specific (17) | (Xiaojun et al., 2012) was changed into (Yao et al., 2012) | everywhere |
| RC1 Specific (18) | This part was removed from the 2nd paragraph of conclusion | P49, L9-L13 |
| RC1 Specific (19) | Part of the content in section 4 was moved into the supplementary file | P30 L19-P31 L4 Supplementary file part 2 |
| Modifications based on RC2 | | |
| RC2 General (1) | See details in the modified abstract, introduction and conclusion sections | P1, L23-P2-L2 |
| RC2 General (2) | See details in the modified method section, validation section and | P19-P20; P25, P29, P33; |

| | supplementary file | Supplementary file Part 3 |
|---|---|---|
| RC2 General (3) | The description of the dataset is combined with the data availability following the validation part | P35, L4-L20 |
| RC2 General (4) (a) | An analysis of how uncertainty in altimetry data evolve into optical data is provided in section 4.2 | P33 |
| RC2 General (4) (b) | See details in the modified section 4.2 | P25, P29 |
| RC2 General (4) (c) | Section 4.3 was added | P34 L10-L15 |
| RC2 Specific comments | | |
| Abstract (1) | Numbers of lake volume changes were removed from the abstract; validation of the optical water level was emphasized | P1, L23-P2-L2 |
| Abstract (2) | None | None |
| Abstract (3) | Brackets have been removed and a brief explanation to optical water levels has been added | P1, L21 |
| Abstract (4) | "densified" has been replaced with "merged" | everywhere |
| Abstract (5) | Numbers of lake volume changes were removed | P1, L30-P2, L2 |
| | | |
| Introduction (1) | More information was added to the 1$^{st}$ paragraph of introduction to clearly state the motivation of TP lake studies | P2, L12-L14 |
| Introduction (2) | None | None |
| Introduction (3) | References was changed into (Frappart et al. 2005) | P2, L21 |
| Introduction (4) | "ice berg" was changed into "ice sheet/sea ice freeboard" | P2, L26 |
| Introduction (5) | "ICESat-1" was changed into "ICESat" | everywhere |
| Introduction (6) | None | None |
| Introduction (7) | None | None |
| Introduction (8) | Reference for Otsu method was added | P3, L24 |
| Introduction (9) | None | None |
| Introduction (10) | (Wan et al., 2016) and (Yang., 2017) were removed from Table 1 | P4, Table 1 |
| Introduction (11) | An explanation in brackets was added following "hypsometric curve" | P5, L5 |
| | | |
| Study area and data (1) | Part of 4$^{th}$ paragraph was moved to the | P9, L7-L10 |

| | second paragraph of section 3.2. | P14, L14-L16 |
|---|---|---|
| Study area and data (2) | None | None |
| Study area and data (3) | An overview map was added | P7, Figure 2 |
| Study area and data (4) | "moderate set of orbital parameters" was removed and replaced by "…a lower orbit than Jason-1/2/3 but higher than ICESat…" | P7, L14 |
| Study area and data (5) | None | None |
| Study area and data (6) | None | None |
| Study area and data (7) | Information of sensor name and type was added into Table 2, as well as data record name; A separate paragraph was added following 1st paragraph of section 2.2 to clarify the difference between altimeters | P9, Table 2 P8, L9-P9, L3 |
| | | |
| Methods (1) | 1st paragraph was moved to the Introduction part | P10, L2-L15 P5, L11-L18 |
| Methods (2) | Same as SC1 Specific (1) | Same as SC1 Specific (1) |
| Methods (3) | Reference to Figure 1 was added several places; More information about the second regression was added | P5, L7, L10; P7, L7; P12, L8, L19 P19, L11-P20, L3 |
| Methods (4) | "Envisat" was changed into "Envisat/RA-2" | P11, L17 |
| Methods (5) | "bucket" was changed into "histogram bin" | P11, L29, L30 |
| Methods (6) | None | None |
| Methods (7) | Same as SC1 Specific (1) | Same as SC1 Specific (1) |
| Methods (8) | References to Figure 3 and Figure 4 were added in the manuscript and part of this paragraph was moved to the end of the optical water levels section (section 3.2). More information about the second regression was added. Discussion about extrapolation issue was added into the supplementary file | P12, L17-L19 P19, L8-L10 P19, L11-P20, L3 Supplementary file part 3 |
| Methods (9) | References to Figure 3 were added | P7, L2 |
| Methods (10) | The sequence has been changed as this paragraph is moved to the end of section 3.2 | Same as Methods (8) |
| Methods (11) | None | None |
| Methods (12) | Reference for Otsu method was added | P17, L11 |
| Methods (13) | None | None |
| Methods (14) | None | None |

[revised manuscript text omitted]

---

## Author Response (AR2)

Response Letter to the Topical Editor Report

Overall comment:

thanks for the authors for the replies to the reviews of your paper and for the revision. The overall tendency of the reviews of your paper was positive and your suggestions for the revision are convincing.

Response:

Thanks for this encouraging comment. The Topical Editor's reports have been addressed point-by-point as follows.

Specific Comment:

1) As indicated by the reviewers and added by the authors not all lake level data sets refer to the same reference.

   The authors state: ... For the 12 lakes with Jason data, all water levels were converted into T/P, EGM96, because the Jason-1/2/3 data were used as the baseline … for lakes without Jason data but having Envisat data, the water levels were converted into WGS84, EGM2008 … For lakes without either Jason or Envisat data, Cryosat-2 data were used as the baseline, so water levels for these lakes were converted to WGS84, EGM96 …

   For publication in ESSD, the authors need to publish a 2nd version of the data set Li et al 2019 in PANGAEA with corrected meta data related to the Geoid.

   In addition, the data set could be further optimized by changing 0 = optical to 0= Landsat + DEM in the meta data, however, this change in the metadata of 0 = optical to 0= Landsat + DEM is not a requirement.

Response:

Thanks for this comment. The data set have been updated with corrected meta data, which is available at https://doi.org/10.1594/PANGAEA.898411. As suggested by the editor, we replaced '0 – Optical' with '0 – Landsat' in the meta data to make it consistent with the rest labels, e.g., '1 – Envisat'. The reason for not using '0 – Landsat + DEM' is because DEM was not directly used in generating the Landsat-derived water levels (what we used to call optical water levels). Instead, altimetric information was incorporated in the generation of Landsat-derived water levels as illustrated in the manuscript. Therefore, to be concise and avoid confusion, we used '0 – Landsat' instead of '0 – Landsat + DEM'.

2) The term 'optical water level' for the Landsat-derived lake area cross referenced with a DEM is not optimally formulated (as data set, in the metadata, subtitles and throughout the text of the manuscript). Already existing formulations of 'Optical water level' are referring to in situ optical sensors measurements or remote sensing laser-derived water levels. The authors' formulation in the title as Landsat derived lake shoreline position is a well-chosen formulation. I recommend of exchanging

'optical water level throughout the manuscript text, subtitles and graphs with Landsat lake shoreline position, or for a shorter naming in figures with Landsat instead of optical.

Response:

Thanks for this comment. As indicated by the editor, the term 'optical water level' might be misleading because there were existing formulations with different meanings. Therefore, we have changed the term 'optical water level' to 'Landsat-derived water level' or in short 'Landsat-derived' (used in figures) throughout the manuscript. The 'Landsat-derived lake shoreline position' is an intermediate variable in the generation of the water level and thus not accurate enough as a denotation of this newly-proposed water level.

3) The title could be slightly shortened to" high temporal resolution water level and storage change data sets for lakes on the Tibetan Plateau during 2000–2017 using multiple altimetric missions and Landsat-derived lake shoreline positions "

Response

Thanks for this comment. Yes, it has been shortened as suggested by the editor.

4) Please provide more details on the sources, here named aviso, NASA, ESA. Table 2 is a good opportunity to do this, e.g. add in the table caption or below the link and the date of data access. Please provide in the manuscript text the source of the used DEM.

Response:

Thanks for this comment. Yes, details have been added for the source data. SRTM DEM data were used but only in graphing. Since DEM information was not used to generate the Landsat-derived water levels nor lake storage changes, we thought it is not necessary to list it in section 2.2, instead we cited the DEM source in the caption of Figure 4.

5) Be consistent in naming a specific Landsat mission together with the sensor. e.g., p.4 last paragraph: Landsat 7 add the sensor: Landsat 7 ETM+, Figure 4 Landsat ETM+ add the mission: Landsat 7 ETM+; Figure 5, 9 Landsat OLI add the mission: Landsat 8 OLI, etc.

Response:

Thanks for this comment. The naming of Landsat data has now been unified as 'mission + sensor', i.e., 'Landsat 5 TM', 'Landsat 7 ETM+', and 'Landsat 8 OLI'.

6) Write out all abbreviations when they appear 1rst time; e.g. p.2. all sensor abbreviation, etc. p.3 LEGOS, also all space programs, ESA, NASA,…; later in the manuscript new sensors appear ,e.g. p.9 write out UAV and GF-2, e.g. Gaofen-2 (GF-2), China High Resolution Earth Observation System (CHEOS) mission, if you use abbreviations in a table. e.g. table 2 you can introduce them in the table captions e.g. GDR, same for abbreviations of methods, please write them out if you name

them for the 1rst time, e.g. API.

Response:

Thanks for this comment. All abbreviations have now been written out at the first time when they appear in the manuscript.

List of Modifications:

1. Meta data of the online data set have been corrected.

2. All figures have been updated in aspect of color, symbol, font size or style to make them more readable.

3. Page 1 - Title has been shortened as suggested by the editor.

4. 'Optical water level' has been replaced with 'Landsat-derived water level' throughout the manuscript including figures.

5. Data source information including website link and last access have been provided in the caption of Table 2.

6. Write out all abbreviations: Page 2 – ICESat, ERS, T/P; Page 3 – SAR, LEGOS; Page 7 – NASA, ESA, CNES; Page 8 – InSAR, GLAS; Page 9 – GF-2, GDR, S-GDR, Aviso, GLAH14; Page 10 – LSH; Page 11 – API. 'UAV' has been written out in Page 9 in the previous version.

7. Naming of Landsat products have been unified throughout the manuscript.

8. Part of Section 4.3 was slightly reorganized.

[revised manuscript text omitted]

---

## Author Response (AR3)

Response Letter to Editor Decision

Thank you for your revision of the manuscript and the data set published on PANGAEA. There are some minor issues left for revision.
Response:
Thanks for this positive evaluation. We are glad to have the manuscript further improved according to the editor's comments. Our point-by-point responses are given as follows.

General Comment: throughout the text
1) avoid the term 'study lakes', you could, e.g., use 'lakes in this study' or also often 'lakes' would be adequate, for example table 1: N of lakes, instead of N of study lakes
Response:
Thanks for this comment. Yes, it is clear enough to use 'lakes' in most cases. We have replaced all 'study lakes' with either 'lakes' or 'lakes in this study'.

2) avoid the term 'rest lakes' (e.g. table 1), you could either name them as 'group of lakes related /derived from xx sensor' or 'group of lakes not related to xx sensor'
Response:
Thanks for this comment. Yes, we have replaced all 'rest lakes' as suggested.

3) Figure and table captions: please spell out all abbreviations you use in tables and figures in the respective captions
Response:
Thanks for this comment. Yes, all abbreviations in tables and figures are spelled out in captions.

Specific Comments:

1) Abstract
   a) recommendation: when you for the first time use Landsat expand it to Landsat satellite data
Response:
Thanks for this comment. It has been modified as suggested.

   b) typo: The resulting merged Landsat-derivedl
Response:
Thanks for this comment. It has been corrected.

   c) This sentence remains unclear: 'consistent with the theoretical analysis' – for the abstract in general: as your conclusions are strong and well formulated, please check if eventually you could use more of the content in your conclusions for the abstract

Response:

Thanks for this comment. The magnitude of the theoretically analyzed uncertainty has been added to the abstract to make it clear. In section 4, we pointed out that for a typical lake, Yamzhog Yumco, the uncertainty of the Landsat-derived water level could be ~0.12 m if we neglect the regression error, and will not exceed 0.22 m if the regression error is considered. Therefore, we think it is proper to say that the theoretically analyzed uncertainty for Landsat-derived water level is 0.1–0.2 m.

2) Table 3 change figure caption to Summary of regression analyses and hypsometric function by lake

Response:

Thanks for this comment. It has been done.

3) Figure 4) (b) ROI (yellow area) selected from a Landsat 7 ETM+ image for detecting changes in lake shoreline and the gaps (black areas) – this statement remains unclear – how can you detect change in the areas of data gaps?

Response:

Thanks for this comment. Yes, this statement is unclear. We have changed it into 'region of interest (ROI, yellow area) selected from a Landsat 7 ETM+ image for detecting changes in lake shoreline (black areas represent gaps in the image)'. There shall be no confusion now.

4) PANGAEA data publication
   We ask for the accuracy of decimal degree coordinates with at least 4 digits. Your published data set has an accuracy of decimal degree coordinates with 2 digits only. Please change your data publication accordingly.

Response:

Thanks for this comment. It has been done.

List of Modifications

1. Abstract: 'Landsat archive' was changed into 'Landsat satellite data'; magnitude of theoretical-analyzed uncertainty was added; Typos 'Landsat-derivedl' was corrected.
2. Throughout the manuscript: 'study lakes' was changed into 'lakes' or 'lakes in this study'.
3. Table 1: 'rest lakes' was changed into 'lakes without xxx data'.
4. Figure 3: abbreviation 'LSH' was spelled out in the caption.
5. Table 2: abbreviation 'CNES' was spelled out in the caption.
6. Figure 4: unclear sentence 'ROI (yellow area) selected from a Landsat 7 ETM+ image for detecting changes in lake shoreline and the gaps (black areas)' was changed into 'region of interest (ROI, (yellow area) selected from a Landsat 7 ETM+ image for detecting changes in lake shoreline and the gaps (black areas represent gaps in the image)'.
7. Figure 5: 'NDWI' and 'MNDWI' was spelled out in the caption.
8. Figure 6: 'GEE' and 'ROI' was spelled out in the caption.
9. Table 3: title of Table 3 was changed as suggested by the editor.
10. Figure 8: y-axis label of Figure 8 (b) was corrected.
11. Figure 15 and 16: 'LSH' was spelled out in the caption.

[revised manuscript text omitted]